# Bacteriophages avoid autoimmunity from cognate immune systems as an intrinsic part of their life cycles

**Jakob T. Rostøl** [1,4] ✉, **Nuria Quiles-Puchalt** [1,2,4], **Pablo Iturbe-Sanz**[3], **Íñigo Lasa** [3] & **José R. Penadés** [1] ✉

Dormant prophages protect lysogenic cells by expressing diverse immune systems, which must avoid targeting their cognate prophages upon activation. Here we report that multiple *Staphylococcus aureus* prophages encode Tha (tail-activated, HEPN (higher eukaryotes and prokaryotes nucleotide-binding) domain-containing anti-phage system), a defence system activated by structural tail proteins of incoming phages. We demonstrate the function of two Tha systems, Tha-1 and Tha-2, activated by distinct tail proteins. Interestingly, Tha systems can also block reproduction of the induced *tha*-positive prophages. To prevent autoimmunity after prophage induction, these systems are inhibited by the product of a small overlapping antisense gene previously believed to encode an excisionase. This genetic organization, conserved in *S. aureus* prophages, allows Tha systems to protect prophages and their bacterial hosts against phage predation and to be turned off during prophage induction, balancing immunity and autoimmunity. Our results show that the fine regulation of these processes is essential for the correct development of prophages' life cycle.

Outnumbered by bacteriophages (phages), numerous defence systems are expressed in bacteria to protect against phage infection[1–4]. Recent breakthroughs in bioinformatics have led to a wealth of systems being discovered[4–8], with concomitant development of biotechnological tools. Interestingly, such defence systems are commonly encoded by mobile genetic elements (MGEs), such as temperate phages[9,10], plasmids[11] and phage satellites[12–14]. This protects both the bacterial host and the MGE from external threats. Yet, these systems may also target the MGE they are carried on. To prevent autoimmunity, the MGE must either avoid recognition or use mechanisms to evade these immune systems, although how this occurs is often unclear.

Due to their limited genome sizes, bacteria and viruses often rely on operons with a single promoter to encode functionally related genes. Genes can also be transcribed in opposite directions, being

reciprocally regulated through RNA polymerase collisions and double-stranded RNA degradation. For example, in *Staphylococcus aureus*, 'noncontiguous operons' have been defined[15], where, nested within a standard operon, an antisense gene(s) undergoes transcription under its own promoter (Extended Data Fig. 1a). A similar genetic architecture has been observed in eukaryotic viruses, referred to as 'complex transcripts'[16,17]. Nevertheless, the physiological implications of such overlapping transcripts remain unclear.

In this Article, we describe Tha (tail-activated, HEPN (higher eukaryotes and prokaryotes nucleotide-binding) domain-containing anti-phage system), a phage-encoded anti-phage system widespread in *S. aureus*. Tha is a non-specific RNase that is activated by conserved minor tail proteins from temperate phages, analogous to pattern recognition in eukaryotic immunity[18]. In phage 80α, Tha can also be activated

[1]Centre for Bacterial Resistance Biology, Imperial College London, London, UK. [2]School of Health Sciences, Universidad CEU Cardenal Herrera, CEU Universities, Alfara del Patriarca, Spain. [3]Laboratory of Microbial Pathogenesis. Navarrabiomed, Hospital Universitario de Navarra (HUN), Universidad Pública de Navarra (UPNA), IdiSNA, Pamplona, Spain. [4]These authors contributed equally: Jakob T. Rostøl, Nuria Quiles-Puchalt. ✉e-mail: j.rostoel@imperial.ac.uk; j.penades@imperial.ac.uk

by the 80α minor tail protein, and to prevent autoimmunity, Tha is encoded in a noncontiguous operon along with its inhibitor, a gene previously annotated as an excisionase and renamed *ith-1* (inhibitor of Tha-1). This genetic organization allows protection against incoming phages while avoiding being triggered by the cognate phage during prophage induction. Overall, our research highlights the importance of phage-carried accessory genes for MGE defence and how the regulation of such systems can protect by recognizing conserved phage components while avoiding autoimmunity.

## Results

### The 80α *xis* gene does not encode an excisionase

In lambdoid-like phages, the integrase (*int*) is required for phage integration, while both the *int* and the nearby excisionase (*xis*) are required for prophage excision. In the *S. aureus* phage L54a, *orf2* (Extended Data Fig. 1b) was reported to act as an excisionase[19], with the equivalent gene being annotated as *xis* in many staphylococcal phages. In the current study, however, we show that Xis is not an excisionase, the gene being renamed *ith-1* (for details, see 'Tha-1 and Tha-2 are inhibited by their cognate Ith proteins').

First, phages 80α and Φ11 encode different integrases, with different chromosomal attachment sites, while having identical *ith-1* genes (Extended Data Fig. 1c). A higher level of sequence similarity between their integrases would be expected if they interact with the same excisionase (Ith-1). Second, in an 80α *ith-1* deletion phage (80α$^{\Delta ith-1}$), we observed a severe drop in the phage titres upon prophage induction (3–4 logs), as expected for an excisionase mutant. However, when using 80α$^{\Delta ith-1}$ to infect susceptible *S. aureus* cells, the plaque size was markedly reduced (Fig. 1b). If Ith-1 was an excisionase, the plaque size should be unaffected as the phage does not require Xis or Int during the lytic cycle. Indeed, the plaque size of the integrase mutant (80α$^{\Delta int}$) was similar to that in wild-type phage (Fig. 1b). Next, to track DNA excision, we performed quantitative polymerase chain reactions (qPCR) to analyse the excision of the wild-type 80α, 80α$^{\Delta int}$ and 80α$^{\Delta ith-1}$ prophages upon induction. Upon mitomycin C induction, both 80α and 80α$^{\Delta ith-1}$ increased in empty chromosome abundance, while 80α$^{\Delta int}$ remained unaffected, suggesting that the 80α$^{\Delta ith-1}$ mutant can excise normally from the chromosome, unlike 80α$^{\Delta int}$ (Fig. 1c). This was corroborated using non-quantitative PCRs with 80α and Φ11 mutants (Extended Data Fig. 1d–g). Altogether, these data suggest that the genes labelled *xis* in 80α and Φ11 serve a different role than as an excisionase.

To investigate the role of *ith-1*, we evolved the 80α$^{\Delta ith-1}$ mutant phage until we obtained plaques identical to those produced by the wild-type phage (Fig. 1d), which are expected to be insensitive to the *ith-1* deletion through genetic changes. Sequencing of five parallel evolutions revealed that all mutants had disruptions in the *orf3* gene (Supplementary Data 1a), suggesting that *orf3* causes the defects observed in the *ith-1* mutants. To validate this, we generated a mutant 80α with an *ith-1/orf3* double deletion (80α$^{\Delta(ith-1+tha-1)}$), and the resulting prophage produced the same titres as wild-type 80α upon mitomycin C induction (Fig. 1e), confirming that Ith-1 is not required for phage excision and that the 80α$^{\Delta ith-1}$ phenotype is mediated via ORF3. From here on, ORF3 is referred to as Tha (see 'Tha is a *bona fide* anti-phage system in *S. aureus* phages') and the 80α/Φ11 variant as Tha-1.

### Tha is a *bona fide* anti-phage system in *S. aureus* phages

One explanation for why Tha-1 causes lower 80α induction titres without *ith-1* could be that Tha-1 is an anti-phage system that causes autoimmunity when improperly regulated. To test the anti-phage properties of Tha-1, we cloned the *tha-1* gene from 80α onto a plasmid (pTha-1$^{80α}$) and challenged *S. aureus* strain RN4220 cells harbouring this plasmid with a panel of phages in spot assays. These data are summarized in the heat map in Fig. 2a, where the fold protection relative to an empty plasmid is shown. This revealed that pTha-1$^{80α}$ provides strong anti-phage resistance, targeting similar temperate Siphoviridae, but not more

distantly related lytic Myoviridae (Supplementary Data 1b). To confirm that Tha-1 also protects cells when expressed from a prophage, we also challenged lysogens with wild-type 80α or 80α with Tha-1 deleted (80α$^{\Delta tha-1}$) (Fig. 2a). We observed that wild-type 80α blocked 16 of the 22 phages tested, while 9 were also blocked by plasmid pTha-1$^{80α}$. For some phages (for example, Φ7206), Tha-1 accounts for all the interference observed by 80α, while for others (for example, K), Tha-1 plays no role in defence, with the observed interference likely being mediated by other 80α-encoded immune system(s), such as the recently described Pdp$_{Sau}$[20].

We next analysed the distribution of the *ith* and *tha* genes in *S. aureus* phages. Multiple phages contain an identical genetic architecture to 80α in their lysogeny regions, with *ith-1/tha-1*, including Φ11 and Φ69. A second group of phages, including ΦNM2 (Extended Data Fig. 2a) and ΦNM1, have similar genetic structures, but the genes vary slightly in their sequences from the first group (Extended Data Fig. 2b), containing *ith-2* and *tha-2*, which are even more abundant in the databases. Reflecting this, of the phages used in Fig. 2a, two have *tha-1/ith-1* and seven have *tha-2/ith-2* (Supplementary Data 1b).

To test whether the second group of phages also provide Tha-mediated protection, we cloned Tha-2 of ΦNM2 into a plasmid (pTha-2$^{ΦNM2}$) and assayed for immunity. pTha-2$^{ΦNM2}$ protected against the multiple phages tested (Fig. 2a), and pTha-1$^{80α}$ and pTha-2$^{ΦNM2}$ have different specificities, suggesting they are activated or inhibited by different proteins or processes. We confirmed that the protection observed was not caused by general cellular toxicity (Extended Data Fig. 2c) and that the defence was not due to an increase in lysogenization of the incoming phage (Extended Data Fig. 2d,e). Using the ΦNM2 lysogen, we saw robust protection against eight phages, with pTha-2$^{ΦNM2}$ showing strong immunity against six phages (Fig. 2a). Overall, these results highlight the broad immune potential of phage accessory genes, with Tha-1 and Tha-2 playing a major role against some phages.

To further elucidate the 80α Tha-1-mediated anti-phage protection, we infected lysogens containing 80α or 80α$^{\Delta(ith-1+tha-1)}$ with Φ7206 at multiplicities of infection (MOIs) of 0.1 and 5 (Fig. 2b). At both MOIs, bacteria lysogenic with 80α resist infection, while in the absence of Tha-1 (80α$^{\Delta(ith-1+tha-1)}$), Φ7206-mediated lysis was observed. Enumerating plaque-forming units (p.f.u.) after the experiment shows that Tha-1 also reduced the number of phages produced by infection (Fig. 2c). It is possible that Tha mediates abortive infection (Abi)-like immunity[3], where the infected cells stop growing and do not lyse, preventing phage replication and protecting the uninfected population. Consistent with this, the 80α cells recovered slowly at MOI 5, likely because most of the bacterial population was initially infected and did not survive (Fig. 2b), while they grow well at MOI 0.1, where most cells were not infected.

### Tha recognizes and is activated by phage tail components

In the absence of *ith-1*, Tha-1$^{80α}$ inhibits 80α reproduction, indicating that the 80α lytic cycle activates processes or produces components recognized by Tha-1. To determine the activator of Tha-1, we passaged 80α to select for phage mutants insensitive to Tha-1 inhibition (Methods). After several rounds of infection, we obtained one plaque for which titre and plaque size were partially restored. Sequencing revealed a F13C mutation in *orf61* (ORF61) (Fig. 1a), a gene in the 80α packaging region encoding a minor tail protein[21]. In parallel, when we performed the same experiment with Tha-2$^{ΦNM2}$ against Φ11, we acquired an escaper phage where Φ11 had a ΔT248-G396 deletion in *orf54* (ORF54$^{\Delta T248-G396}$). Φ11 ORF54 is another minor tail protein[22], homologous to 80α ORF62. Tha-1$^{80α}$ and Tha-2$^{ΦNM2}$ selected for mutants in different minor tail genes, suggesting that Tha-1 and Tha-2 are activated by separate tail proteins, possibly explaining the differences in interference observed earlier (Fig. 2a). These escaper mutants showed reduced fitness, with 80α ORF61$^{F13C}$ producing below 50% of wild-type titres and Φ11 ORF54$^{\Delta T248-G396}$ producing 100- to 1,000-fold lower titres compared to wild type upon prophage induction (Extended Data Fig. 3a,b).

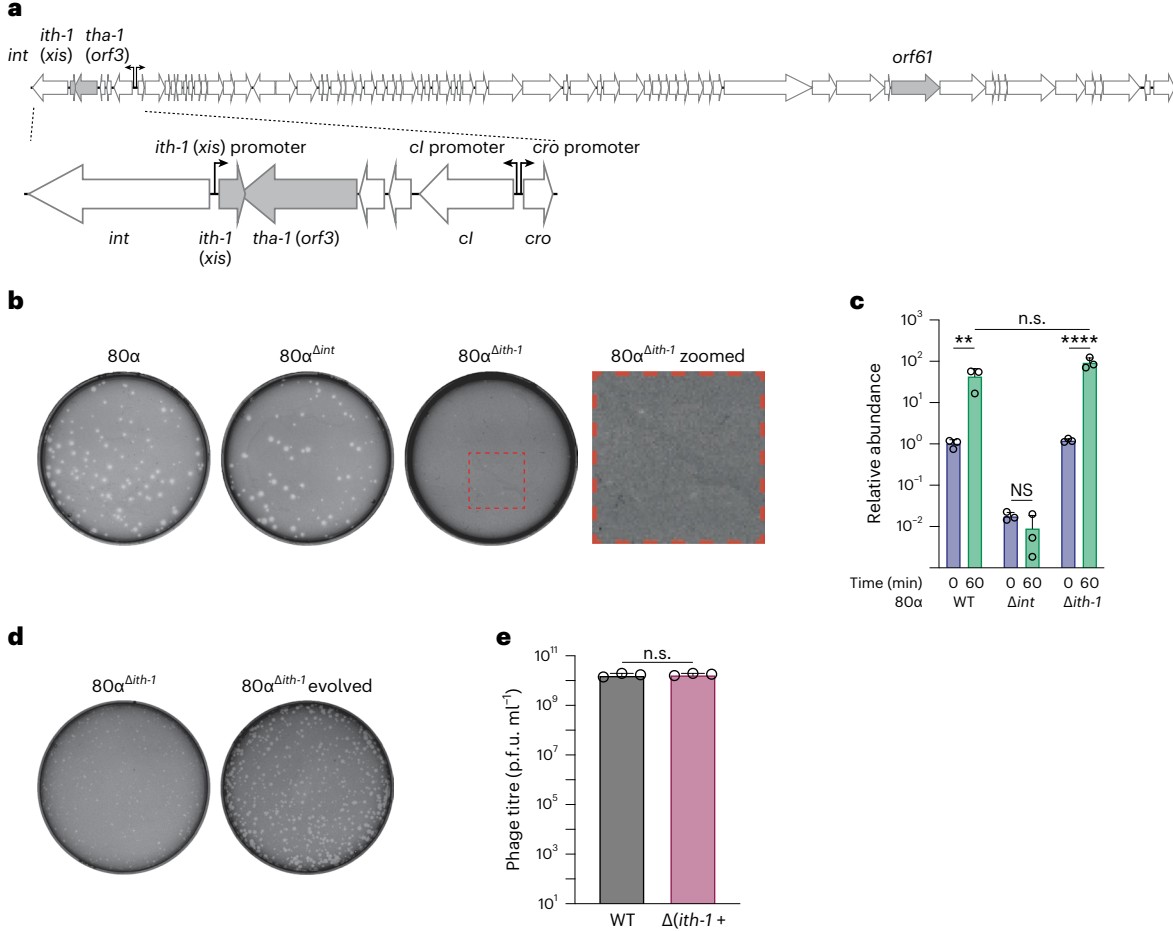

**Fig. 1 | The ith-1 (xis) gene does not act as an excisionase during 80α prophage induction. a**, A map of the phage 80α genome, with salient features highlighted. **b**, Representative plaques on *S. aureus* RN4220 from prophage induction of wild-type 80α, 80α$^{\Delta int}$ and 80α$^{\Delta ith-1}$. A zoomed in picture is provided for 80α$^{\Delta ith-1}$ to show the small plaque phenotype. **c**, Quantification of empty chromosomal *attB* sites, indicative of prophage excision, before and after phage induction by mitomycin C. WT, wild type. **d**, Representative plaques of 80α$^{\Delta ith-1}$ and 80α$^{\Delta ith-1}$ evolved, showing restored plaque sizes following passaging of the 80α$^{\Delta ith-1}$ mutant. **e**, Phage titres following induction of 80α and 80α$^{\Delta(ith-1+tha-1)}$. For **c** and **e**, each bar represents the mean of three biological replicates, and error bars represent s.d. Unpaired, two-tailed *t*-tests are used to determine significance. \*\**P* < 0.01; \*\*\*\**P* < 0.0001; n.s., *P* > 0.05. n.s., not significant. In **c**, *P* = 0.0011, *P* = 0.1926 and *P* < 0.0001 for wild type, Δ*int* and Δ*ith-1*, respectively.

To validate Tha-1 activation by the minor tail protein ORF61, we co-expressed the 80α Tha-1 and ORF61 from separate plasmids in *S. aureus* cells in the absence of phage infection. pTha-1$^{80α}$ or pTha-1$^{80α(Tha-1 H270A)}$ (encoding an inactive derivative mutant; see 'Tha-1 has non-specific RNase activity') were used to constitutively express the wild-type or Tha-1 H270A variants, while a pE194-based plasmid with a tight, anhydrous tetracycline (aTc)-inducible promoter[23] was used to inducibly express ORF61 or ORF61$^{F13C}$. Without aTc, the plasmids could co-exist in the same cell, resulting in similar colony-forming unit (c.f.u.) counts on agar (Fig. 2d). However, when plated on agar containing aTc, ORF61 or ORF61$^{F13C}$ expression was induced, and a severe reduction in c.f.u. was observed in the cultures expressing Tha-1 and ORF61 (Fig. 2d). This decrease was not observed with the Tha-1 catalytic mutant (Tha-1$^{H270A}$), while the Tha-1 with ORF61$^{F13C}$ resulted in a moderate c.f.u. decrease, possibly due to partial Tha-1 activation. To ensure that Tha-1 was activated by ORF61 protein as opposed to *orf61* RNA, we also tested two stop codon mutants (ORF61$^{D2*}$, ORF61$^{H50*}$), where ORF61 protein is not produced but the RNA remains largely unchanged (Extended Data Fig. 4a).

## Tha contains a HEPN domain

Analysis of the 301-amino-acid Tha-1$^{80α}$ coding sequence using HHpred[24] revealed that Tha-1 has no discernible N-terminal domain but

contains a C-terminal HEPN superfamily domain[25] (residues 184–296). HEPN domains are typically non-specific RNases and are anti-phage effectors in type III-A CRISPR–Cas (clustered regularly interspaced short palindromic repeats and CRISPR-associated protein) systems[23] and the HEPN-transmembrane (TM) system[12]. HEPN domains usually have an ArgX$_4$His catalytic motif[25], which is present as R$_{265}$NSIMH$_{270}$ in Tha-1$^{80α}$. A similar domain structure was present in Tha-2$^{ΦNM2}$ (Extended Data Fig. 2b). To determine whether this motif is involved in protection in Tha-1, we challenged cells expressing either a wild-type or a H270A mutant version of Tha-1 with Φ85. As expected, while both proteins were expressed at similar levels (Extended Data Fig. 4b), the catalytic mutant abrogated the anti-phage activity of Tha-1$^{80α}$ (Fig. 3a).

As HEPN domains typically function as dimers, we predicted the structure of the Tha-1$^{80α}$ dimer using AlphaFold2[26,27] (Fig. 3b). This predicted the HEPN region of the dimers with high probability (predicted local distance difference test (pLDDT) > 90), while the N-terminal part of each monomer (residues 1–167) returned low confidence scores (pLDDT < 50). A DALI[28] search of the Tha-1 HEPN dimer (residues 164–301) showed a structural resemblance to known HEPN RNase toxins, including the HEPN-minimal nucleotidyltransferase (MNT) HEPN toxin[29] (PDB accession 5yep, DALI *Z* score 9.9) and *cas* subtype *Mycobacterium tuberculosis* 6 (Csm6) (PDB accession 5yjc, DALI *Z*

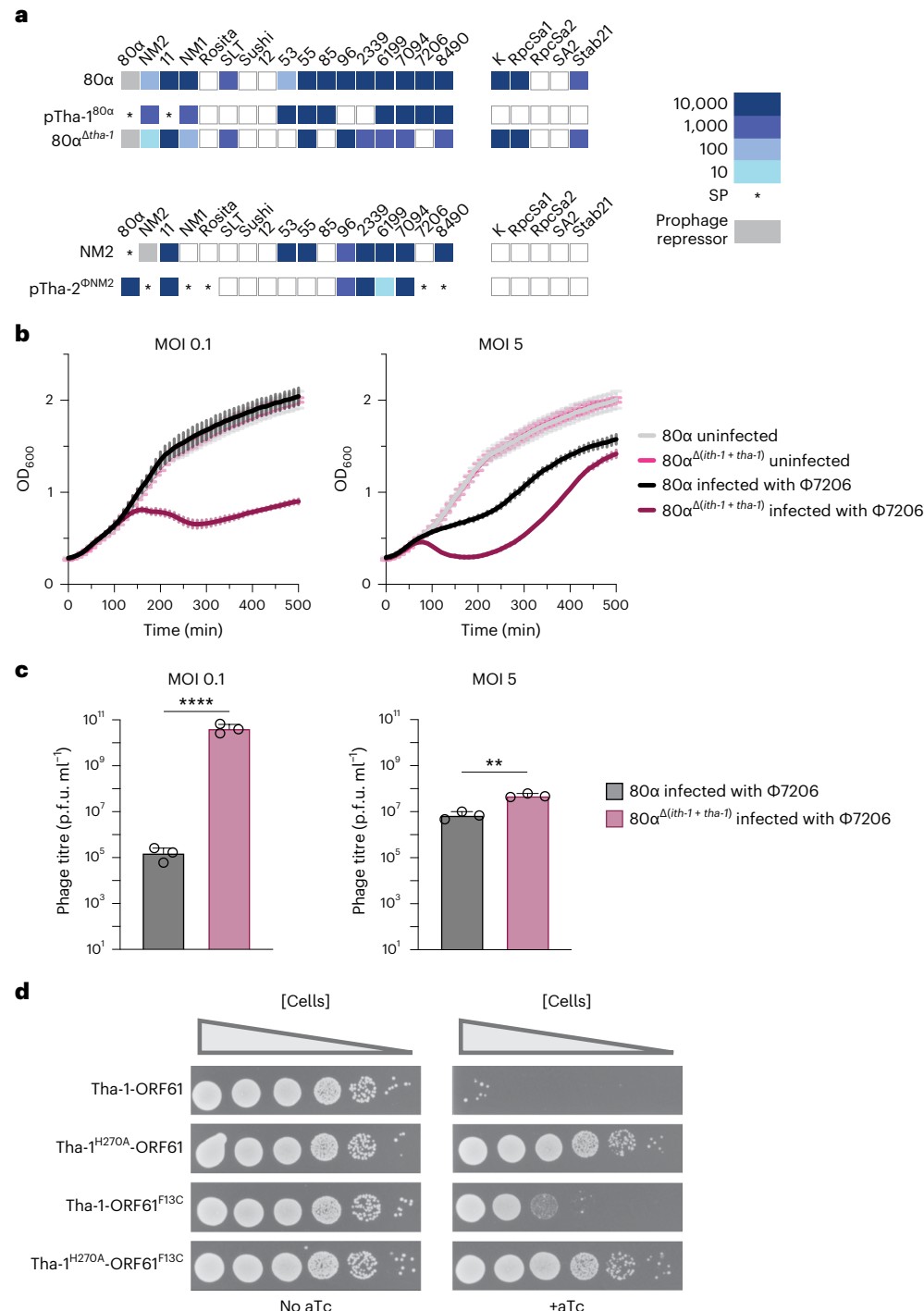

**Fig. 2 | Tha is a bona fide anti-phage system activated by minor tail proteins.**
**a**, A heat map summarizing spot assay experiments on *S. aureus* RN4220 cells using either plasmid-carried Tha (pTha-1$^{80\alpha}$, pTha-2$^{\Phi NM2}$) or lysogens (80α, 80α$^{\Delta tha-1}$, ΦNM2). The tenfold reductions in plaquing efficiency of temperate phages (Siphoviridae, left phage group) or lytic phages (Myoviridae, right phage group) are shown from at least three independent experiments. Where plaques could not be seen, the highest dilution where clearing of the bacterial lawn observed was used. White squares signify no immunity. Prophage repressor indicates repression of the incoming phage by its cognate phage repressor. SP, small (but similar number of) plaques. **b**, Growth curves following infection of

*S. aureus* RN4220 containing prophage 80α or 80α$^{\Delta(ith-1+tha-1)}$ by Φ7206 at a MOI of 0.1 or 5. The same uninfected curves are shown in both graphs. Each datapoint represents the mean of three biological replicates ±s.d. **c**, The p.f.u. retrieved from the cultures following the experiments in **b**. Unpaired, two-tailed *t*-tests are used to determine significance. **P < 0.01; ****P < 0.0001. P < 0.0001 and P = 0.0014 for MOIs 0.1 and 5, respectively. **d**, Tenfold dilutions of *S. aureus* RN4220 cells harbouring a plasmid constitutively expressing either Tha-1 or Tha-1$^{H270A}$, and an aTc-inducible plasmid expressing ORF61 or ORF61$^{F13C}$, plated on agar without (left) or with (right) aTc. A reduction in c.f.u. is indicative of toxicity due to Tha-1-ORF61(/ORF61$^{F13C}$) co-expression.

score 9.0). The structure of the Tha-1 HEPN dimers revealed a predicted dimer with a positively charged cleft between the subunits (Extended Data Fig. 4d), likely for binding negatively charged RNA substrates. The

putative catalytic residues R265 and H270 extend into the cleft (Fig. 3b and Extended Data Fig. 4c), consistent with their roles in catalysis. The failure to predict the N-terminal domain of the Tha-1 dimer suggests

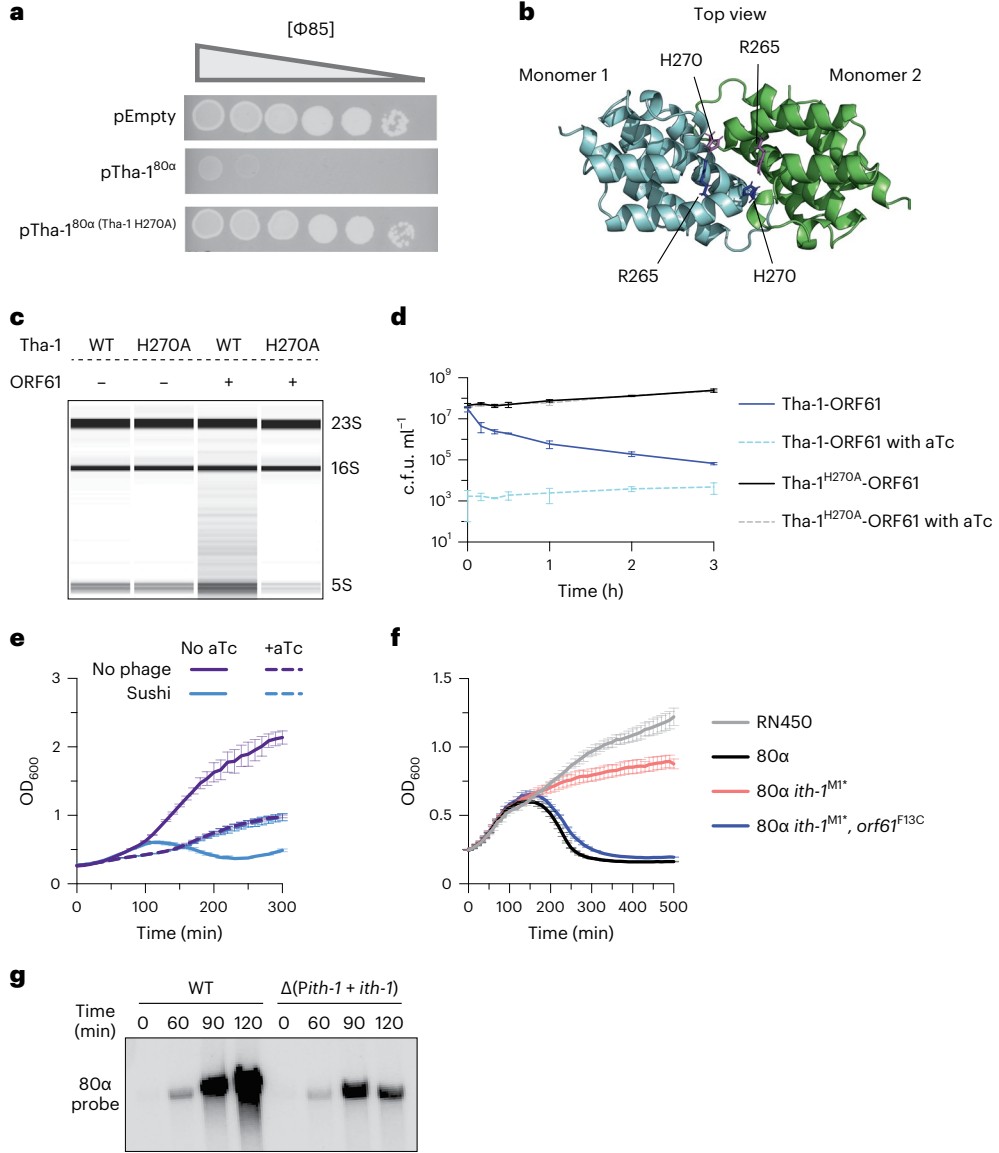

**Fig. 3 | Tha-1 is a HEPN domain-containing RNase that inhibits host cell and phage replication upon activation. a**, Spot assay of *S. aureus* RN4220 harbouring either an empty plasmid (pEmpty), a plasmid encoding 80α Tha-1 (pTha-1$^{80α}$) or a plasmid encoding the catalytic H270A point mutant (pTha-1$^{80α(Tha-1 H270A)}$), being infected with Φ85. Tenfold phage dilutions were plated, with the triangle representing decreasing phage titre. **b**, A top view of the 80α Tha-1 HEPN domain homodimer (residues 164–301), coloured according to chain, with the likely catalytic site in the central cleft. Predicted catalytic residue side chains of R265 and H270 of each monomer are shown. The structure is predicted by AlphaFold2. For more images, see Extended Data Fig. 4c,d. **c**, An Agilent Bioanalyzer image of total RNA isolated from staphylococci expressing wild-type or H270 versions of Tha-1, before and after induction of ORF61 expression with aTc. rRNA bands are labelled. This experiment was performed once. **d**, Enumeration of c.f.u. from cells harbouring plasmids expressing Tha-1/Tha-1$^{H270A}$ and ORF61, before and at time points after inducing ORF61 expression with aTc. Cells are plated on agar to determine the number of viable cells. To obtain the number of escaper mutants, aliquots were also plated on agar containing aTc ('with aTc' cells). **e**, Optical density at 600 nm of RN4220 cells harbouring plasmids expressing constitutive Tha-1, with and without aTc-inducible ORF61 expression initiated at time 0, upon infection by phage Sushi, a phage that does not trigger Tha-1, at an MOI of 1. **f**, Growth curves of *S. aureus* RN450 cells without or with various 80α lysogen variants upon prophage induction by mitomycin C. In **d**–**f**, each datapoint represents the mean of three biological replicates ±s.d. **g**, A southern blot of lysogens with either an 80α prophage or an 80α mutant where Tha-1 triggers autoimmunity (Δ(P*ith-1*+*ith-1*)) before and after induction with mitomycin C, using a probe against 80α DNA. This experiment was performed once.

that Tha-1 may form higher-order oligomers or that this region is disordered in the absence of a binding partner.

## Tha-1 has non-specific RNase activity

Having identified the activator of the system, we sought to validate the RNase activity of Tha-1. Using cells carrying the same two-plasmid system as above, we isolated total RNA before and after activating ORF61 expression with aTc and ran the samples on an Agilent Bioanalyzer. This revealed non-specific degradation of rRNAs upon co-expression

of wild-type Tha-1 and ORF61 (Fig. 3c), confirming that Tha-1 is a non-specific RNase.

To investigate whether the toxicity of the Tha-1$^{80α}$ RNase activity was reversible, we used the two-plasmid system and induced ORF61 expression with aTc at time 0, removing aliquots over time to enumerate remaining viable c.f.u. When Tha-1 was activated, there was a rapid loss in viability after 10 min, with a further gradual decline over 3 h in cells that could be resuscitated following Tha-1 activation (Fig. 3d). Tha-1 is therefore rather toxic, consistent with an Abi mechanism,

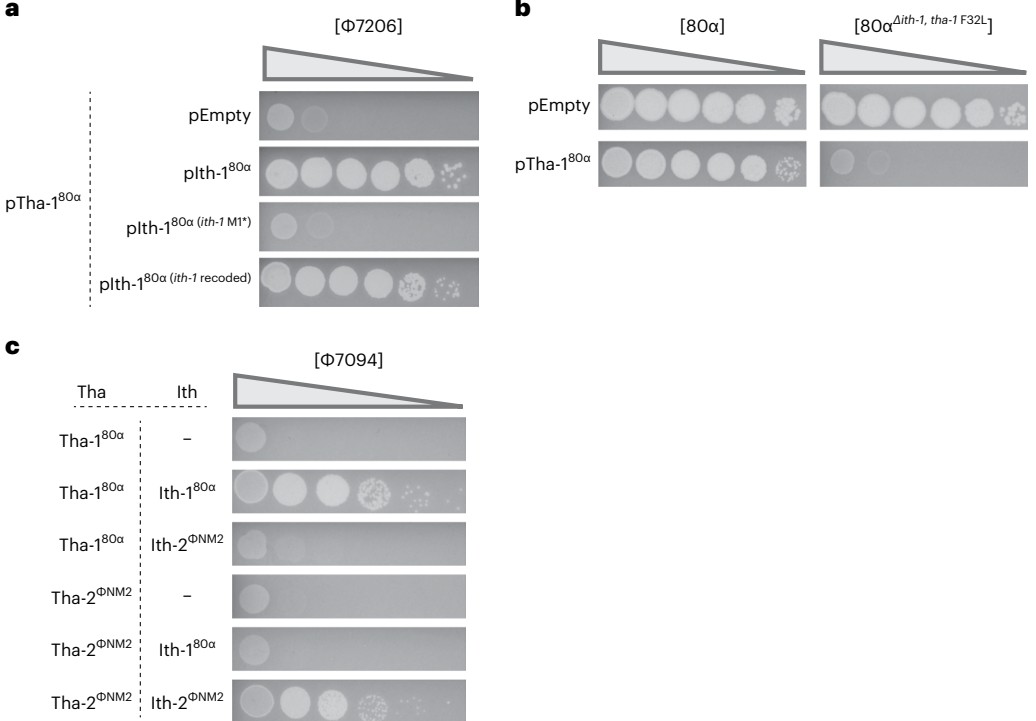

**Fig. 4 | Tha is inhibited by its cognate Ith protein. a**, RN4220 cells harbouring a plasmid constitutively expressing *tha-1*$^{80\alpha}$, and a plasmid that is either empty (pEmpty) or expresses different versions of *ith-1* genes under their native P*ith-1* promoter, are infected on soft agar plates by Φ7206. pIth-1$^{80\alpha}$ encodes an *ith-1* gene with 90/201 nucleotide substitutions but the same amino acid sequence. Tenfold phage dilutions are spotted. **b**, Cells with either an empty plasmid or a plasmid expressing *tha-1*$^{80\alpha}$ are infected by tenfold dilutions of either 80α or 80α$^{\Delta ith-1, tha-1\,F32L}$, a mutant with an *ith-1* deletion and a non-functional version of Tha-1. **c**, Like **a**, but with one plasmid expressing Tha-1$^{80\alpha}$ or Tha-2$^{\Phi NM2}$, and another plasmid being empty, or expressing Ith-1$^{80\alpha}$ or Ith-2$^{\Phi NM2}$. Cells are infected by tenfold dilutions of phage Φ7094.

although we were unable to validate this at physiological protein levels with prophage induction/infection due to the expression of other toxic phage genes.

To test whether co-expression of Tha-1/ORF61 was sufficient to prevent phage lysis, we used the two-plasmid system with phage Sushi, a phage insensitive to pTha-1$^{80\alpha}$ (Fig. 2a). In a liquid growth assay, we saw phage lysis caused by phage Sushi in the absence of ORF61 expression, but with aTc, the Tha-1-ORF61 co-expression prevented phage lysis (Fig. 3e). Overall, these experiments suggest that minor tail protein expression during the lytic cycle either directly, through protein–protein interactions, or indirectly, via host factors, activates the anti-phage activity of Tha-1.

In our previous experiments, we noticed that for 80α$^{\Delta ith-1}$, lysis was incomplete upon prophage induction. To corroborate ORF61 being the Tha-1 activator, we performed growth curves with different 80α lysogens after the addition of mitomycin C to initiate prophage induction. In contrast to wild-type 80α, 80α with an *ith-1* stop codon (*ith-1*$^{M1*}$) failed to lyse (Fig. 3f), implying a growth arrest was caused by Tha-1 activation without Ith-1. Lysis was restored in the double mutant where ORF61$^{F13C}$ was also present (*ith-1*$^{M1*}$, *orf61*$^{F13C}$), aligning with the ability of 80α ORF61$^{F13C}$ to evade Tha-1 targeting.

To determine how Tha-1 impacts phage replication, we performed a southern blot on wild-type 80α and an 80α mutant where Tha-1 causes autoimmunity due to a deletion of *ith-1* and its promoter (P*ith-1*) (Δ(P*ith-1*+*ith-1*)). Early replication is largely unaffected by Tha-1, but there is markedly less 80α phage DNA present at later time points in the Δ(P*ith-1*+*ith-1*) condition (Fig. 3g). This is consistent with Tha-1 being activated by the relatively late-expressed ORF61 minor tail protein, with a concomitant decrease in phage replication due to a general growth arrest.

## Tha-1 and Tha-2 are inhibited by their cognate Ith proteins
Based on our results, we suspected that the 80α Ith-1 protein could inhibit its cognate Tha-1. To test this, we used phage Φ7206 to infect RN4220 cells harbouring plasmid pTha-1$^{80\alpha}$ and a second plasmid, which was either empty (pEmpty), expressed 80α *ith-1* (pIth-1$^{80\alpha}$), *ith-1* with a stop codon mutation (pIth-1$^{80\alpha\,ith-1\,M1*}$) or a recoded *ith-1* (pIth-1$^{80\alpha\,ith-1\,recoded}$). Protection was abrogated in the presence of wild-type and recoded Ith-1, but *ith-1*$^{M1*}$ did not inhibit Tha-1 (Fig. 4a), confirming that the Ith-1 protein can inhibit its cognate Tha-1.

In transcriptomic analyses of 80α during infection, we noticed that the *ith-1* gene is expressed shortly after infection (Extended Data Fig. 5). We wondered whether this *ith-1* expression could protect the infecting phage from Tha-1 immunity. When infecting cells harbouring a plasmid-expressed Tha-1 (pTha-1$^{80\alpha}$) with wild-type 80α, plaques are formed (Fig. 4b). In an 80α mutant lacking *ith-1*, however, pTha-1$^{80\alpha}$ blocked infection. Early expression of *ith-1* by the incoming wild-type 80α is therefore sufficient to circumvent Tha-1 immunity, making Ith-1 a bona fide immune evasion protein.

As 80α and ΦNM2 encode *tha-1* and *tha-2*, respectively, activated by different minor tail proteins, we wondered whether their respective *ith-1* and *ith-2* genes have different immune suppression specificities. To test this, we infected staphylococci with Φ7094, a phage sensitive to both Tha-1 and Tha-2 (Fig. 2a). These cells harboured two plasmids, one encoding either Tha-1$^{80\alpha}$ or Tha-2$^{\Phi NM2}$ and the second expressing either Ith-1$^{80\alpha}$ or Ith-2$^{\Phi NM2}$. As expected, the Tha protein of each system is inhibited only by its cognate Ith protein and not by the non-cognate Ith from the different system (Fig. 4c).

Next, we attempted to understand the targeting specificities observed in Fig. 2a. Of the 17 temperate phages, 15 encode 80α ORF61 (recognized by Tha-1) and Φ11 ORF54 (recognized by Tha-2) homologues (Supplementary Data 1b). Yet, sequence alignments and phylogenetic

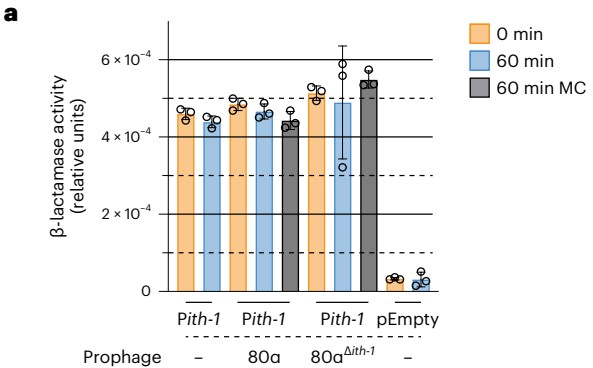

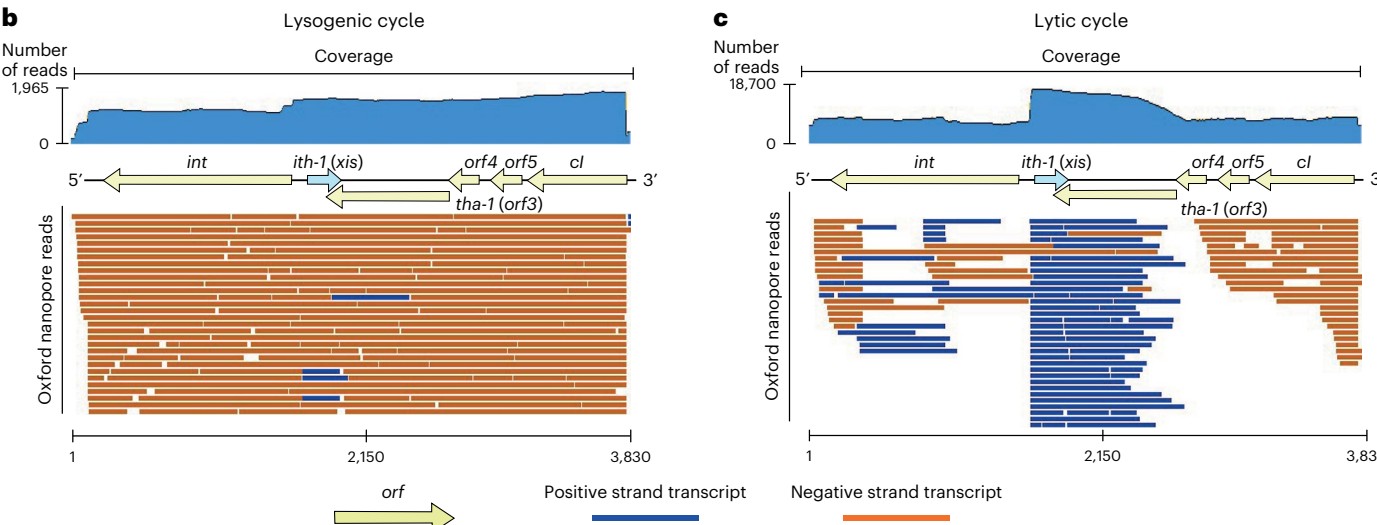

**Fig. 5 | tha-1 and ith-1 are parts of two antisense, overlapping operons.**
**a**, A plasmid reporter assay with *S. aureus* cells harbouring a plasmid with the *ith-1* promoter (P*ith-1*) in front of a *blaZ* reporter gene, or a plasmid without P*ith-1* in front of *blaZ* (pEmpty). Where indicated, 80α or 80α$^{\Delta ith-1}$ prophages are also present. Samples were taken at 0 min, 60 min or 60 min after prophage induction by mitomycin C. Promoter activity is measured in relative units and revealed no visible change in P*ith-1* activity with prophage induction. Each bar represents the mean of three biological replicates ±s.d. MC, mitomycin C. **b**,**c**, The transcriptome of the lysogenic region of 80α. Phage 80α RNA was purified and sequenced from S. aureus RN450 harbouring 80α before (**b**) and after (**c**) induction with mitomycin C, representing the lysogenic and lytic phage life cycles, respectively. Going from the lysogenic to the lytic cycle, the abundance of *tha-1* transcripts (from the *cI* promoter) decreases, with a concomitant increase in *ith-1* transcripts (from the P*ith-1* promoter). The read depth is indicated on top, on a logarithmic scale, and each read mapped to the phage genome is shown as a line. Blue arrows represent antisense reads, which is transcribed in the opposite direction to the operon (yellow open reading frames (*orf*)). The genomic positions are shown along the *x*-axis. See also Extended Data Figs. 7 and 8.

trees of these proteins revealed no strong correlation with Tha-1 and Tha-2 sensitivity (Extended Data Fig. 6a,b). However, as the *ith-1* of 80α can protect from Tha-1-mediated interference (Fig. 4b), we wondered whether other phages might also be protected by their *ith* genes. Indeed, more phages contained *tha-2/ith-2* (Extended Data Fig. 6c,d), and none of the *ith-1* phages were strongly targeted by Tha-1, while none of the *ith-2* phages were strongly targeted by Tha-2 (Fig. 2a). Overall, this analysis suggests that sensitivity to a Tha system depends on both the *orf54/orf61* homologous genes of a phage and whether it harbours an *ith* gene.

### *Ith-1/tha-1* form an antisense transcriptional unit

Transcriptomic analyses[30,31] and our previous results indicated that Tha-1 is highly expressed in the integrated 80α prophage, and upon prophage induction, Ith-1 is expressed to inhibit Tha-1. As *ith-1* expression must be finely tuned for proper regulation of immunity, we hypothesized that the *ith-1* promoter (P*ith-1*) activity is regulated by 80α, as *ith-1* transcript levels increase after prophage induction[30]. Nevertheless, using a β-lactamase *blaZ* reporter assay (Methods), we concluded that P*ith-1* is constitutive, regardless of phage induction by mitomycin C (Fig. 5a). Instead, *ith-1* levels are likely controlled by the localization of *ith-1* within the phage genome.

As displayed in Fig. 1a, all the genes in this region are controlled by the *cI* promoter, except for *ith-1*, which is expressed in the antisense direction under its own constitutive P*ith-1* promoter, a genetic architecture that can be categorized as a noncontiguous operon[15] (Extended Data Fig. 1a). To characterize this region, we performed direct RNA sequencing (RNA-seq) of 80α using Oxford Nanopore technology and observed a dramatic transcriptome shift between the lysogenic and lytic cycles (Fig. 5b,c and Extended Data Fig. 7a,b). In the lysogenic phase, the lysogeny module genes (starting from *cI*) were expressed as a large transcript that overlaps with *ith-1*, confirming the existence of the noncontiguous operon, with weak *ith-1* expression (Fig. 5b). During the lytic cycle, however, *ith-1* expression increased, while the *cI* operon, including *tha-1*, had reduced transcript abundances (Fig. 5c). The noncontiguous operon architecture facilitates switching between *cI*/*tha-1* expression and *ith-1* expression, making the two expression states mutually exclusive. During lysogeny, the strong *cI* promoter interferes with the weaker P*ith-1* through antisense transcription, favouring Tha-1 immunity, whereas during the lytic cycle, the *cI* promoter is turned off, leading to increased P*ith-1* activity and Tha-1 immunity being turned off. Moreover, we noticed three other regions where long transcripts overlap with genes transcribed in the opposite direction (Extended

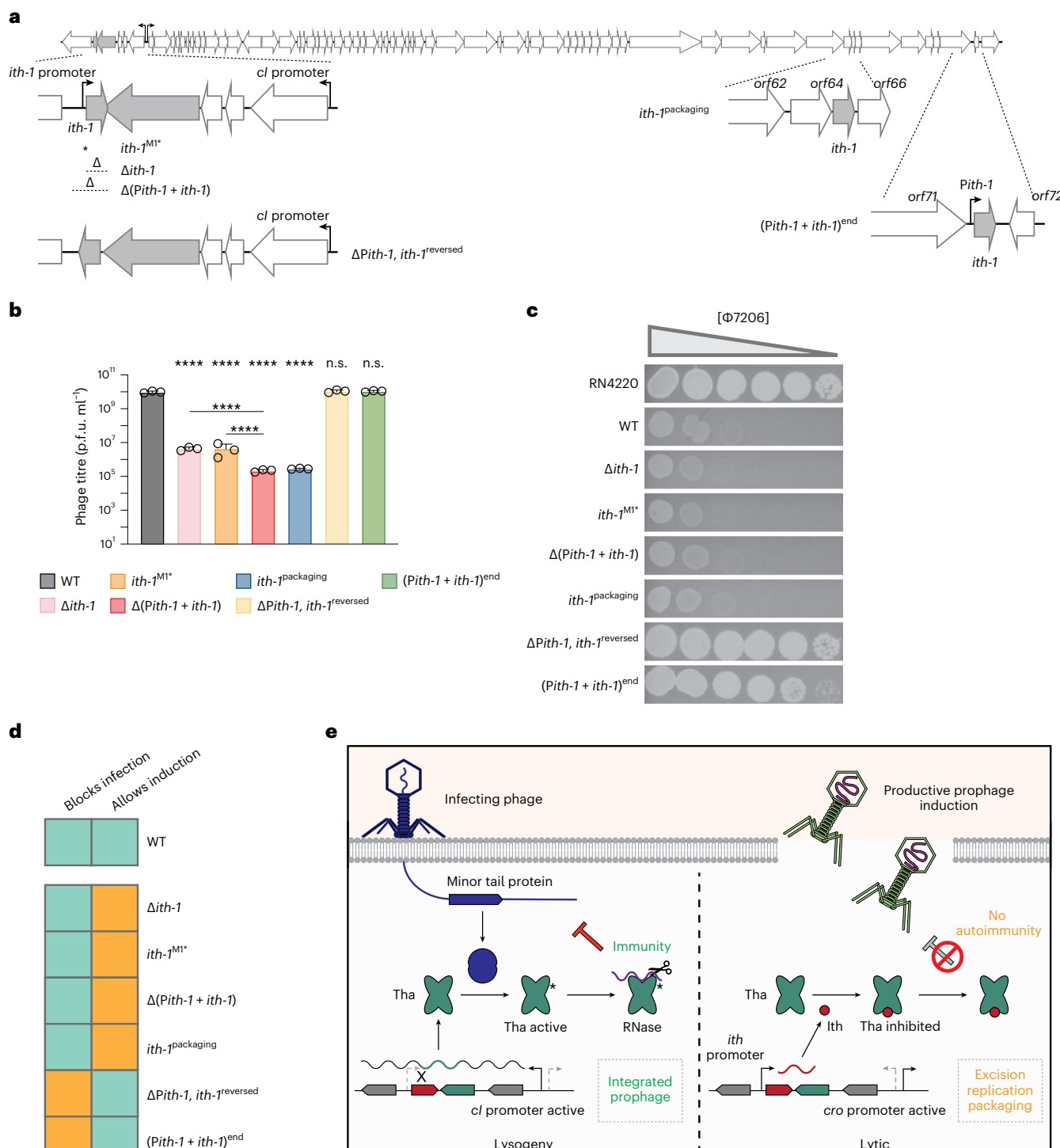

**Fig. 6 | The reciprocal regulation of the genes in the ith/tha region allows both productive cognate prophage induction, and immunity against infecting phage. a**, An overview of the 80α genome, showing the positions and orientations of the mutations used to dissect the operonic structure. See text for details. **b**, Resulting phage titres following induction of *S. aureus* RN4220 cells harbouring wild-type 80α or various 80α mutants from **a**. Each bar represents the mean of three biological replicates, error bars represent s.d. One-way analysis of variance with multiple comparisons was used to compare each mutant against wild-type 80α and for comparing Δ*ith-1*/*ith-1*M1* and Δ(P*ith-1*+*ith-1*). ****$P < 0.0001$; NS, $P > 0.05$. NS, not significant. **c**, Tenfold dilutions of infecting Φ7206 spotted on bacterial lawns of the same strains as in **b**, with the triangle representing decreasing phage titre. **d**, A summary of the ability of wild-type 80α and operonic mutant phages to both block phage infection (immunity) and allow

prophage induction (avoid autoimmunity), with green indicating positive and orange indicating negative for each ability. Only in the wild-type, noncontiguous operon configuration (Extended Data Fig. 1a) of 80α are both functions preserved. **e**, A model of the regulation of the Tha/Ith system during the lysogenic and lytic cycles. On the left, the strong *cI* promoter expresses *tha*, while also reducing the expression of *ith* (expressed from the P*ith* promoter) through RNA polymerase collisions (shown as an X). Upon phage infection, the phage produces minor tail proteins, which activate the RNase activity of Tha, blocking phage and host cell replication. On the right, the prophage enters the lytic cycle. The *cI* promoter is turned off, with less Tha expressed, and the *ith* promoter produces Ith protein to inhibit Tha. The phage life cycle can then proceed like normal (driven by the *cro* promoter), producing mature phage particles.

Data Fig. 8a–c). This highlights how *S. aureus* phages rely on antisense noncontiguous operons to switch from lysogeny to lysis.

## The noncontiguous operon balances immunity and autoimmunity

For Tha to be a successful phage-carried anti-phage system, it must (1) block incoming phages during lysogeny and (2) be switched off when its cognate prophage enters the lytic cycle to avoid autoimmunity. We hypothesized that the noncontiguous operon structure around *tha-1/ith-1* in 80α was the genetic architecture that allowed this switch, coupling Tha-1 activity to the phage life cycle. Therefore, the disruption of the genetic configuration should either abrogate protection against phage infection or impede cognate prophage induction. To dissect the noncontiguous operon, we generated 80α mutants with altered genetic architecture (Fig. 6a). One group of mutants was expected to not properly express *ith-1*, leading to Tha-1 being constitutively on, resulting in protection from external phages but improper 80α induction. This group includes *ith-1*[M1*] (*ith*-1 gene with a stop codon), Δ*ith-1* (*ith-1* deleted), Δ(P*ith-1*+*ith-1*) (*ith-1* promoter and *ith-1* deleted) and *ith-1*[packaging] (*ith-1* promoter and *ith-1* deleted, and *ith-1* gene inserted in the 80α packaging region, expected to be expressed late in the phage life cycle). The second group of mutants expresses *ith-1* outside of the antisense context of the noncontiguous operon and includes Δ*Pith-1*, *ith-1*[reversed] (where the *ith-1* promoter is deleted, and *ith-1* is reversed, being expressed from the *cI* promoter) and (P*ith-1*+*ith-1*)[end] (where the *ith-1* promoter and *ith-1* are moved to the end of the phage). These mutants are expected to have elevated *ith-1* expression, leading to Tha-1 being inactive.

First, we tested the ability of the 80α mutants to undergo successful prophage induction (Fig. 6b), which requires that Tha-1 is turned off to prevent autoimmunity. With one group, including Δ*Pith-1*, *ith-1*[reversed] and (P*ith-1*+*ith-1*)[end], the phage titres were indistinguishable from wild-type 80α, suggesting that sufficient Ith-1 is produced to inhibit Tha-1 activity. By contrast, for the other variants, prophage induction was severely impaired. Notably, and in agreement with previous data (Fig. 4a), the Ith-1 protein is the major Tha-1 inhibitor, although P*ith-1* also plays a minor role, presumably through RNA polymerase head-on collisions with transcription from the *cI* promoter (Extended Data Fig. 9a). Moreover, Ith-1 must be expressed by P*ith-1*, as when controlled by the late lytic promoter (*ith-1*[packaging]), Tha-1 still impedes prophage induction. These phage titres were consistent with the resulting phage plaque sizes observed for each mutant (Extended Data Fig. 9b).

Next, we investigated the ability of the 80α mutants to block incoming phage infection by Φ7206 (Fig. 6c). For successful immunity, Tha-1 must be turned on and not be inhibited by Ith-1 and/or P*ith-1* transcription. As expected, for the group of variants with deleted or late-expressed *ith-1*, Tha-1 immunity was active and blocked Φ7206 plaque formation. For the other group of mutants, the Δ*Pith-1*, *ith-1*[reversed] mutant showed no phage immunity, likely because Ith-1 expression is driven by the strong *cI* promoter. In the case of (P*ith-1*+*ith-1*)[end], there was mild blocking (reduced Φ7206 plaque size). Here Ith-1 is made from the P*ith-1* promoter, but there is no transcription from P*ith-1* suppressing *tha-1* transcription such as there is in the native noncontiguous operon. Therefore, P*ith-1*-driven Ith-1 production alone is not sufficient to completely block Tha-1, and P*ith-1* antisense transcription in the noncontiguous operon contributes with mild suppression of the *tha-1* transcript.

Overall, we conclude that the antisense genetic architecture around *ith-1/tha-1* is essential for proper Tha-1 function. In other configurations, Tha-1 can be turned off during induction (Fig. 6b) and be constitutively on during lysogeny (Fig. 6c). However, only when the wild-type noncontiguous operon is intact can both activities be balanced for the benefit of the (pro)phage (Fig. 6d,e). Moreover, this inhibition is largely based on Ith-1 protein being produced, although P*ith-1* transcription also plays a minor role in Tha-1 suppression (Fig. 6b,c).

## Discussion

To enhance the fitness of their bacterial hosts, prophages often express 'moron' genes during lysogeny[32]. These include anti-phage genes, which protect the prophage and the host from external phage threats[33,34]. Here we describe Tha, a defence system encoded by many *S. aureus* phages, with variants Tha-1 (for example, in 80α) and Tha-2 (for example, in ΦNM2). In response to phage, the infected cell likely dies through a combination of non-specific RNase activity by the Tha HEPN domain and toxic phage gene products, thereby protecting the cell population through an Abi-like mechanism. Tha is directly or indirectly triggered by minor tail proteins expressed by the infecting phage, ORF61[80α] homologues for Tha-1 and ORF54[Φ11] homologues for Tha-2. These proteins are highly conserved in many *S. aureus* temperate phages. Tha activation therefore mirrors other recently described bacterial defence systems[35–37] and eukaryotic pattern recognition receptors[18,38]. Recognition of conserved signatures of infection confers broad targeting specificities to the systems[18] and makes it difficult for a phage to escape targeting through random mutations due to a concomitant loss of fitness (Extended Data Fig. 3).

Yet, if a phage-carried defence system recognizes conserved proteins from closely related phages, the systems must also avoid autoimmunity, that is, not target its cognate phage during prophage induction. Here we find that a gene previously annotated as a phage excisionase (*xis*), renamed *ith* (*ith-1* for the 80α variant, *ith-2* for the ΦNM2 variant), does not promote excision but instead encodes an Ith protein that inhibits Tha. *ith/tha* are encoded in a noncontiguous operon[15] (Extended Data Fig. 1a), where the main *cI* operon (including *tha*) is disrupted by the antisense *ith* gene, driven by its own promoter, P*ith* (Fig. 1a). A similar principle was elucidated with the anti-phage system BstA, which is inhibited by a cognate DNA immunity element through an unknown mechanism[39]. We show that in 80α, this genetic architecture is essential for coupling Tha-1 activity to the phage life cycle (Fig. 6d), with Tha-1 protecting the cell during the lysogenic cycle and being turned off to avoid autoimmunity when 80α enters the lytic cycle (Fig. 6e).

Finally, we argue that accessory gene regulation is an intrinsic part of the phage life cycle. Prophages protect themselves and their host against external threats by encoding anti-phage systems, promoting their survival and dissemination in nature. At the same time, the lytic phage life cycle can only be successful in the absence of autoimmunity, and in our case, 80α[Δith-1] mutants result in severely reduced phage titres upon activation. Moreover, encoding *ith* can help phages circumvent Tha immunity when infecting Tha-positive cells, (Extended Data Fig. 6c,d). For these reasons, we believe that additional elements beyond those directly involved in excision/integration, replication and packaging are also important factors in a temperate phage life cycle.

## Methods

### Bacterial strains and growth conditions

*S. aureus* RN4220 or RN450 was grown at 37 °C in tryptic soy broth, unless stated otherwise. Where relevant, 10 μg ml⁻¹ chloramphenicol or 10 μg ml⁻¹ erythromycin was supplemented for plasmid maintenance. Plasmids were isolated by Miniprep (Qiagen) and transformed into *S. aureus* through electroporation. Strains used in this study are listed in Supplementary Data 1c.

### Molecular cloning

Plasmids used in this study are listed in Supplementary Data 1d. Oligonucleotides (oligos) are listed in Supplementary Data 1e. Cloning strategies are listed in Supplementary Data 1f. Sanger sequencing to confirm plasmids and mutants was performed by Eurofins Genomics. For cloning the recoded *ith-1*, an *ith-1* gene where 90/201 nucleotide positions were changed throughout the open reading frame was synthesized by ThermoFisher and cloned into a plasmid.

For phage editing, prophages were edited in the chromosome as previously described[40,41]. Briefly, the allelic exchange vector pMAD-βgal (β-galactosidase) was transformed into the recipient cells and plated on tryptic soy agar (TSA) with 2.5 µg ml⁻¹ erythromycin at 30 °C to allow for the first recombination event. A colony was then picked, grown at 30 °C and plated at 42 °C on TSA with 2.5 µg ml⁻¹ erythromycin and 80 µg ml⁻¹ X-gal, selecting light blue colonies (indicative of integration). These were grown at 30 °C, and white colonies (indicative of the second recombination event) were screened for the presence of the intended mutation by PCR and sequencing.

#### Phage infections

Phages used in this study are listed in Supplementary Data 1b. For phage spot assays, an overnight culture of recipient cells was diluted and grown to an optical density ($OD_{600}$) of about 0.34. Lawns were made by mixing 300 µl of these cells with phage top agar and 5 µM $CaCl_2$ and poured onto square plates containing phage base agar. Tenfold phage serial dilutions were made and spotted on the bacterial lawns. The plates were then incubated for 18 h. The fold change was estimated as the reduction in phage titre relative to a non-protective control (empty plasmid). In most cases with protection, plaques could not be seen, so the last dilution where a clearance in the lawn could be seen was used. For pTha-1$^{80α}$ and pTha-2$^{ΦNM2}$, expression was driven by a constitutive promoter (originally from the type III-A CRISPR–Cas system from *Staphylococcus epidermidis*[42]) on a middle-copy pC194 plasmid[43].

For phage titering, 50 µl of RN4220 cells were mixed with 100 µl of the intended phage dilution and incubated at room temperature for 10 min. This was then mixed with phage top agar and 5 µM $CaCl_2$, poured on a phage base agar plate and incubated at 37 °C for 24 h.

#### Phage inductions

For phage inductions, overnight RN4220 or RN450 containing respective prophages were diluted 1:50, grown to an $OD_{600}$ of about 0.25 and normalized to an $OD_{600}$ of 0.2. About 2 µg ml⁻¹ mitomycin C was added, and the cultures were incubated at 30 °C overnight with shaking at 80 rotations per minute (r.p.m.). Phage lysates were sterilized with 0.2 mm filters (Minisart syringe filter, Sartonius Stedim Biotech). In the case of incomplete lysis, the $OD_{600}$ values were measured to about 0.5 after the overnight induction. For induction in the growth reader, 0.1 µg ml⁻¹ mitomycin C was used.

#### Lysogenization assay

To test whether THA can block lysogenization of incoming phage, RN4220 cells with an empty plasmid or a plasmid expressing Tha-2$^{ΦNM2}$ were infected by 1,000,000 p.f.u. of 80α containing an erythromycin cassette after the lysis genes. About 1 ml of cells at $OD_{600}$ 1.4 was mixed with 100 µl phage lysate at the desired dilution in the presence of 5 µM $CaCl_2$ and incubated at 37 °C for 30 min. The cells were then spun down, resuspended in 100 µl, plated on TSA plates containing 10 µg ml⁻¹ erythromycin and 17 mM Na-citrate to prevent secondary infections, and incubated for 24 h.

#### Non-quantitative and quantitative PCRs for prophage excision

For non-quantitative analysis of 80α and Φ11, 1.5 ml samples of overnight cultures from strains lysogenic for the wild-type phage and the indicated mutants were taken, and DNA extraction was performed using GenElute Bacterial Genomic DNA kit (NA2120; Merck) following the manufacturer's instructions.

For qPCR to track 80α excision, 80α, 80α$^{int}$ and 80α$^{ith-1}$ were induced as above for 60 min. These samples, and 0 time point samples, were lysed with 1 mg ml⁻¹ lysozyme and 1 mg ml⁻¹ lysostaphin at 37 °C for 20 min before the addition of 25:24:1 of phenol:chloroform:isoamyl alcohol with mixing. After centrifuging ($16,000 × g$ for 10 min), the aqueous phase was isolated, mixed with chloroform and spun as before. The aqueous phase from this was mixed with isopropanol, mixed and

centrifuged, and the supernatant was removed. The pellet was washed with 75% ethanol, dried and resuspended in nuclease-free $H_2O$. For the qPCR, 20 ng of total DNA was mixed with 100 nM of each primer, and PowerSYBR Green PCR master mix (ThermoFisher) was added according to the manufacturer's protocol. The qPCR was performed on an Applied Biosystems StepOnePlus real-time PCR machine. The primers used span the empty chromosomal attachment (*att*B) site (empty chromosome, that is, prophage has excised), and the internal control amplified the gyrase gene.

#### Escaper phage generation

To obtain evolved phages overcoming the Δ*ith* defect, plaque lawns from 80α$^{Δith-1}$ titrations using RN4220 as recipient strain were collected using a sterile spreader and added to 3 ml of phage buffer (1 mM NaCl, 0.05 M Tris pH 7.8, 1 mM $MgSO_4$, 4 mM $CaCl_2$) followed by centrifugation and filtration. The resulting lysate was used to infect fresh cultures of recipient bacteria, and the process was repeated until plaques showing wild-type morphology were observed as a majority in the phage population. Individual plaques were isolated by picking and resuspending the plaque in 400 µl of phage buffer, followed by filtering to produce a pure phage stock that was used to lysogenize RN4220. The lysogenic strains carrying evolved phages were subsequently verified as 80α$^{Δith-1}$ mutants by PCR, followed by induction and titering to ensure that a lysis and titre recovery had also been achieved. DNA was extracted from an overnight culture using GenElute Bacterial Genomic DNA kit (NA2120; Merck) following manufacturer's instructions and sent for whole-genome sequencing with MicrobesNG (Birmingham, UK).

To obtain escape phages overcoming Tha-1$^{80α}$ and Tha-2$^{ΦNM2}$ blocking, we made use of strains JP16029 (RN4220 pCN51 Tha-1$^{80α}$) and JP16112 (RN4220 pCN51 Tha-2$^{ΦNM2}$) where the corresponding Tha has been cloned under an inducible cadmium promoter. Titration experiments were done using phage 80α and Φ11 lysates (with phages where the *ith-1* and *tha-1* genes were mutated) and adding 5 µM $CdCl_2$ to induce expression of Tha. Although several lines of evolution experiments were carried out in parallel, we were only able to obtain one individual plaque per evolution where the morphology of the plaque was restored to the wild-type phenotype. These plaques were isolated, as previously described, and lysogenized in RN4220, and DNA was extracted and sent for whole-genome sequencing with MicrobesNG (Birmingham, UK)

#### Growth curves

For all growth curves with phage infections, overnight cultures in triplicate were diluted, grown to an $OD_{600}$ of 0.25, normalized to an $OD_{600}$ of 0.2 and seeded in a 96-well plate (Costar) in the presence of 5 µM ml⁻¹ $CaCl_2$. Phage was then added where relevant at the indicated MOIs, and the plate was incubated at 37 °C for 24 h with shaking at 500 r.p.m. in a FLUOstar Omega (BMG Labtech). $OD_{600}$ readings were taken every 10 min. To obtain the phage titres from the previous infections, the contents of the 96-well plates were resuspended at the end of the run, centrifuged at $10,000 × g$ for 2 min to remove cells and debris, and the lysate was titred as previously described.

#### Protein structure analyses

To identify the HEPN domain within Tha, the HHpred web-server was used (https://toolkit.tuebingen.mpg.de/tools/hhpred, version 57c8707149031cc9f8edceba362c71a3762bdbf8). To analyse the structure of Tha-1, AlphaFold2[26] and ColabFold v1.5.2[27] were used to generate a model for the structure of the Tha-1–Tha-1 homodimer. The structure was analysed, and images were generated by PyMOL v2.5.4 (Schrödinger). The DALI web server (accessed 27 March 2023)[28] searches against PDB90 (a collection of representative structures with less than 90% sequence identity from the PDB database) were performed for the truncated dimer (164–301).

## Protein alignments

Protein alignments were performed using Clustal Omega v1.2.4[44], and sequence similarity and identity were calculated by EMBOSS Needle v6.6.0[45].

## Western blot

For western blots to compare expression levels of Tha-1 and Tha-1[H270A], 1 ml of the respective overnight cultures (expressing either Tha-1 variant with an N-terminal 3xFLAG tag) was spun down, resuspended in PBS with 1 µg ml$^{-1}$ lysostaphin, 2 µg ml$^{-1}$ lysozyme and 2 µl RQ1 DNase (Promega) and incubated at 37 °C for 20 min. Samples were then run on a 15% SDS–PAGE gel with a PageRuler Plus prestained protein ladder (10–250 kDa) (ThermoFisher), transferred to a nylon membrane, blocked with 0.5% milk and human immunoglobulin G and labelled with antibodies. For Tha-1 and Tha-1[H270A], direct detection was done with a primary antibody fused to horseradish peroxidase (HRP) (at 1/1,000 dilution) against 3xFLAG. For GlmM (phosphoglucosamine mutase), a primary antibody was used first (at 1/2,500 dilution), followed by a secondary, HRP-fused anti-rabbit antibody (at 1/10,000 dilution). C-terminally His-tagged *S. aureus* GlmM protein was expressed and purified from *Escherichia coli* as previously described[46] and used for the production of rabbit polyclonal antibodies at Covalabs under project number 1846011. The HRP substrate used was Clarity Western ECL Substrate (Bio-Rad) with a 5 min incubation time before imaging.

## Tha-1-ORF61 co-expression toxicity test

To assess whether Tha-1 and ORF61 co-expression is toxic in the presence of aTc, overnight cultures containing the relevant plasmids were serial diluted and plated on TSA with 10 µg ml$^{-1}$ chloramphenicol and 10 µg ml$^{-1}$ erythromycin, and with 12.5 ng ml$^{-1}$ aTc where relevant, and incubated at 37 °C for 18 h.

## Detecting in vivo RNase activity by Tha-1

To test the RNase activity of Tha-1, sJR302 (Tha-1/ORF61) and sJR303 (Tha-1/ORF61[H270A]) overnight cultures were diluted and grown until they reached an OD$_{600}$ of 0.2. ORF61 expression was then induced by the addition of 125 ng ml$^{-1}$ anhydrotetracycline (Sigma-Aldrich). Non-induced cultures were kept to use as controls. Cultures were incubated at 37 °C, at 200 r.p.m. for 30 min, and the cells were collected to proceed to RNA extraction. Total RNA was extracted from the bacterial pellets using the TRIzol reagent method. Extracted RNA was treated with 5 µl TURBO DNase I in the presence of 1 µl of SUPERase-In RNase Inhibitor (Invitrogen) at 37 °C for 30 min.

To assess the effect of Tha-1 activity on RNA degradation, RNA integrity was analysed using Agilent RNA Nano LabChips (Agilent Technologies). All RNA samples loaded on the chip were adjusted to 200 ng µl$^{-1}$. The chip reading was performed on a 2100 Agilent Bioanalyzer.

## Time course to determine reversibility of Tha-1 toxicity

To investigate whether, once activated by ORF61, toxicity from Tha-1[80α] led to dormancy (reversible) or cell death (irreversible), *S. aureus* RN4220 cells with a plasmid expressing Tha-1/Tha-1[H270A] and a plasmid with aTc-inducible ORF61 were diluted to an OD$_{600}$ of 0.25. A sample was taken (for pre-induction c.f.u. calculations), and ORF61 expression was induced with 125 ng ml$^{-1}$ aTc. Samples were taken at 10 min, 20 min, 30 min, 1 h, 2 h and 3 h. Each sample was quickly spun down at 10,000 × $g$ for 2 min, supernatant was carefully removed and cells were resuspended. Cells were plated on TSA plates with 10 µg ml$^{-1}$ chloramphenicol and 10 µg ml$^{-1}$ erythromycin for plasmid maintenance, and on the same plates also supplemented with 125 ng ml$^{-1}$ aTc, to determine the number of escaper cells (that is, cells with random mutations in *tha-1* or *orf61*).

## Southern blot

Strains lysogenic for phages wild-type 80α and Δ(P*ith-1+ith-1*) were mitomycin C induced (1 µg ml$^{-1}$), and 1 ml of each culture at different time points after induction was collected and used to prepare standard mini lysates, which were resolved on a 0.7% agarose gel, southern blotted and probed for phage DNA. The labelling of the probes and DNA hybridization were performed according to the protocol supplied with the PCR-DIG DNA-labelling and Chemiluminescent Detection Kit (Roche). Primers are listed in Supplementary Data 1e.

## β-lactamase reporter assay

To detect the *ith-1* promoter (P*ith-1*) activity, cells harbouring plasmid pJP2138, a pCN41-plasmid with the 215 nucleotides upstream of *ith-1* (including the *ith-1* promoter, P*ith-1*) followed by a β-lactamase gene (*blaZ*), and prophages where relevant, were diluted from overnight culture and grown to an OD$_{600}$ of 0.25. Cells were then normalized to an OD$_{600}$ of 0.2, and a 0 time point sample was taken. Mitomycin C was added to the cultures where relevant, and the culture flasks were incubated at 30 °C for 1 h with shaking at 80 r.p.m., with a 60 min time point being taken. For all samples, 10 mM sodium azide was added to stop bacterial growth. Samples were then diluted in 50 mM KPO$_4$ buffer, seeded into a 96-well plate and 50 µl nitrocefin working solution (6 µl of nitrocefin stock (23.8 mg ml$^{-1}$ anhydrous nitrocefin in DMSO) diluted in 10 ml 50 mM KPO$_4$ buffer, pH 5.9) was added to start the reaction. Nitrocefin is a chromogenic substrate that produces colour upon cleavage by *blaZ*. The OD$_{490}$ was measured every 40 s in the plate reader (FLUOstar Omega, BMG Labtech). The slopes of each sample over time were used to calculate the relative β-lactamase activity, which reflects the activity of P*ith-1*. An empty plasmid (pCN41) was used as a negative control. The activity was calculated as the slope (Δabsorbance per second) divided by OD$_{600}$.

## Transcriptomic analysis of 80α during infection

RNA-seq and analysis was performed like previously described[30]. Briefly, *S. aureus* RN450 cells at an OD of 0.2 were infected with phage 80α at a MOI of 1. Samples were taken immediately after and at 30 min and 60 min after infection. Cells were lysed using a FastPrep-24 homogenizer (MP Biomedicals), and total RNA was extracted with an Ambion PureLink Trizol Plus kit according to the manufacturer's protocol. To remove genomic DNA, Quiagen RNase-free DNase kit was used, followed by a second treatment with RQ1 DNase (Promega). Enrichment of messenger RNA was done with MICROBexpress mRNA enrichment kit (Ambion). Complementary DNA (cDNA) synthesis and sequencing were performed by the University of Glasgow Polyomics facility, with Illumina 75 base pairs (bp) single-end reads produced with a NextSeq 500. Reads were mapped to 80α (NC_009526.1) with READemption (v0.4.3) RNA-seq pipeline[47].

## Phylogenetic analysis of phage proteins

To detect ORF61/ORF54 homologues in the phages used, the ORF61 sequence of 80α and the ORF54 sequence of Φ11 were used to find homologous proteins in the phage genomes using pBLAST v2.14.0. Of the temperate phages, 15 of 17 contained homologues of both (apart from Φ12 and ΦSLT), while none of the lytic phages did. To generate the phylogenetic trees, the relevant protein sequences were first aligned using MAFFT[48] (Galaxy version 7.508) through the Galaxy[49] bioinformatics platform. Trees were then constructed using FASTTREE[50] (Galaxy version 2.1.10), also through Galaxy.

## RNA extraction for Oxford Nanopore sequencing

To prepare the bacterial samples for RNA extraction, a culture of *S. aureus* RN10359 (RN450 with an 80α prophage) was grown overnight in 5 ml tryptic soy broth at 37 °C with shaking at 200 r.p.m. The culture was then diluted 1:100 in 20 ml media and incubated at 37 °C with shaking at 200 r.p.m. in a 100 ml flask until it reached an OD$_{600}$ of 0.5. At this point, 2 µg ml$^{-1}$ mitomycin C was added to the media, and the culture was incubated at 32 °C with shaking at 80 r.p.m. for 1 h. A separate culture without mitomycin C was used as a control and grown under the

same conditions. The bacterial cultures were then centrifuged, and the pellets were frozen in liquid nitrogen and stored at −80 °C until needed. Total RNA was extracted from the bacterial pellets using the TRIzol reagent method. Extracted RNAs were treated with 5 µl TURBO DNase I in the presence of 1 µl SUPERase-In RNase Inhibitor (Invitrogen) at 37 °C for 30 min. After removing DNA from the solution, the RNA concentration and RIN (RNA integrity number) were measured using Agilent RNA Nano LabChips (Agilent Technologies). Only RNA samples with at least RIN ≤ 9 were used in further analyses. The RNAs were stored at −80 °C until needed. The sequencing libraries yielded a total of 1,935,275 and 931,725 reads without and with MC induction, respectively. Of these, 8,859 and 192,016 reads from RNA obtained before and after the addition of mitomycin C mapped the 80α genome. The average length of the mapped reads was 441 bp and 1,025 bp, respectively.

## Oxford Nanopore RNA-seq

To prepare the RNA for sequencing, ribosomal RNAs were first removed from 10 µg of total RNA using the Illumina RiboZero Bacteria Kit. The resulting mRNA was then treated with *E. coli* poly(A) polymerase in the presence of ATP to add a polyA tail. The polyadenylated mRNA was then converted into a nanopore-compatible RNA library using the Direct RNA Sequencing Kit from Oxford Nanopore. This library was sequenced for 24 h on an Oxford Nanopore GridION instrument in an R9.4 flow cell, with base calling performed in real time using Guppy v3.5.1. The resulting data were saved to Bio Project PRJNA922758. The resulting reads were mapped using the Geneious Prime 2023 (Biomatters) using the phage 80α genome as a reference (accession number NC_009526).

## Statistics and reproducibility

Statistical analyses were performed on GraphPad Prism v10.1. For details on statistics and reproducibility, see the Reporting Summary.

## Reporting summary

Further information on research design is available in the Nature Portfolio Reporting Summary linked to this article.

## Data availability

The Oxford Nanopore RNA-seq data are available under BioProject PRJNA922758, and the 80α infection RNA-seq data are available at Sequence Read Archive through accession code PRJNA1080480. The phage accession codes are in Supplementary Data 1b. Source data are provided with this paper.

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

## Acknowledgements

We thank H. Ingmer (University of Copenhagen, Denmark) for the kind gift of phage Stab21 and M. Molendijk (Erasmus MC, the Netherlands) for the kind gift of phages RpcSa1, RpcSa2 and K (K was originally provided by S. Brouns (TU Delft, the Netherlands)). We also thank A. Gründling for the kind gift of the anti-GlmM antibody. This work was supported by grants MR/X020223/1, MR/M003876/1, MR/V000772/1 and MR/S00940X/1 from the Medical Research Council (UK); BB/V002376/1 and BB/V009583/1 from the Biotechnology and Biological Sciences Research Council (BBSRC, UK); and EP/X026671/1 from the Engineering and Physical Sciences Research Council (EPSRC, UK) to J.R.P. and grant PID2020-113494RB-I00 from the Spanish Ministry of Science and Innovation (Agencia Española de Investigación/Fondo Europeo de Desarrollo Regional, European Union) to I.L. J.T.R. was supported by EMBO Postdoctoral Fellowship ALTF 164-2021. P.I.-S. was supported by a F.P.I. (PRE2018-084479) contract from the Spanish Ministry of Science, Innovation and Universities.

## Author contributions

J.R.P. and J.T.R. conceived this study; J.T.R., N.Q.-P. and P.I.-S. performed the experiments; J.T.R., N.Q.-P., P.I.-S., I.L. and J.R.P. processed the data; J.T.R. wrote the manuscript with inputs from all the authors.

## Competing interests

The authors declare no competing interests.

## Additional information

**Extended data** is available for this paper at https://doi.org/10.1038/s41564-024-01661-6.

**Correspondence and requests for materials** should be addressed to Jakob T. Rostøl or José R. Penadés.

**Extended Data Fig. 1 | The 80α and Φ11 ith-1 (xis) genes are not excisionases. a**, A diagram showing a standard operon (top), where all the genes are transcribed from the same promoter, and a noncontiguous operon (bottom), where a gene with its own promoter runs in the opposite direction as the rest of the operon. **b**, A schematic depicting the lysogeny region of phage L54a. *orf2* was previously annotated as an excisionase (*xis*). **c**, Protein alignments of the Ith-1 proteins (top) and Int proteins (bottom) of 80α and Φ11, showing that these two phages encode the same Ith-1 (100% identity) but different Int proteins (13.2% identity, 20.4% similarity). **d**, Schematic showing the primers and PCR amplicons used to detect integrated (PCR^int), excised (PCR^exc), and circularised (PCR^circ) phage. Attachment (*att*) sites are highlighted, and primers are shown as coloured arrows. **e**, An

agarose gel showing PCR products from the reactions described in **d**, of WT 80α, 80α with a deleted integrase (80α^Δint), and 80α with a deleted *ith-1* (*xis*) (80α^Δith-1). DNA was isolated from overnight cultures, which will have detectable levels of spontaneous phage induction. The PCRs suggest that the integrase (*int*), but not *ith-1*, is involved in prophage excision. DNA ladders separate the different PCR reactions, with sizes indicated to the left. bp, base pairs. **f**, Representative plaques formed on *S. aureus* RN4220 from prophage induction of WT Φ11, Φ11^Δint, and Φ11^Δith-1 (*xis*). A zoomed in picture is provided for Φ11^Δith-1 to show the small plaque phenotype **g**, As in **e**, but with Φ11 phage variants, suggesting that the Φ11 *int*, but not *ith-1*, is involved in prophage excision. Experiments **e** and **g** were performed once.

**a**

1 kb

**80α**

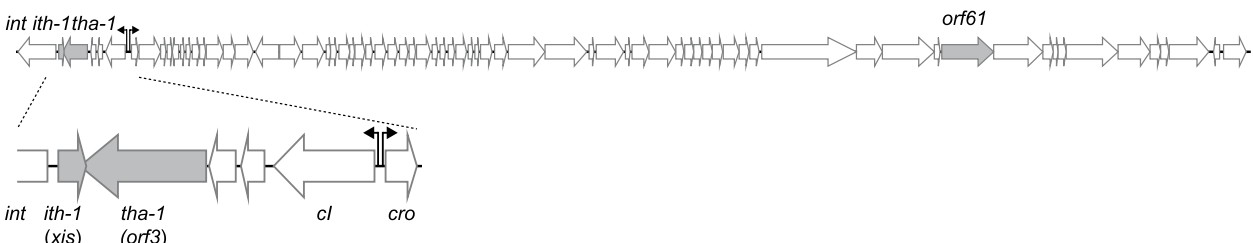

*int* *ith-1* *tha-1*　*orf61*

*int* *ith-1 (xis)* *tha-1 (orf3)* *cI* *cro*

**ΦNM2**

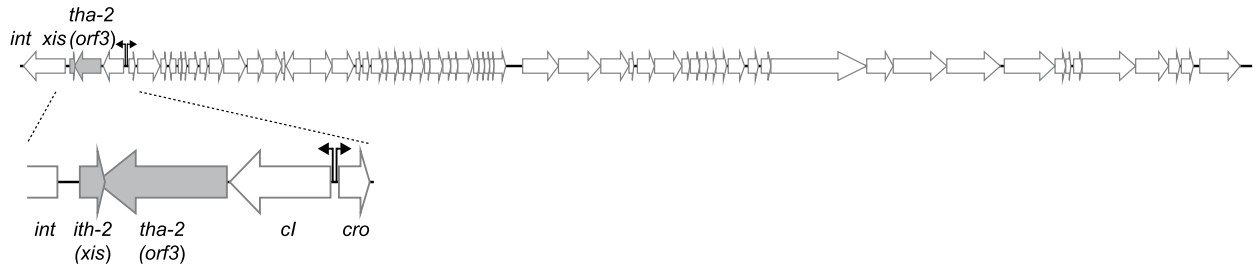

*tha-2* *int* *xis (orf3)*

*int* *ith-2 (xis)* *tha-2 (orf3)* *cI* *cro*

**b**

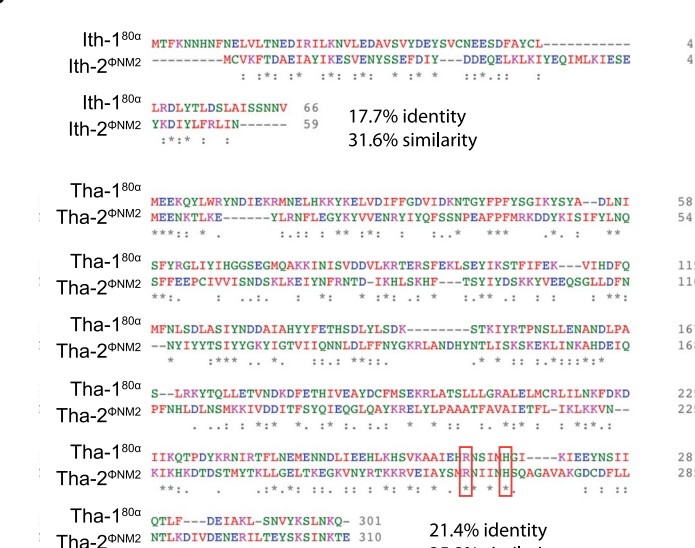

Ith-1⁸⁰ᵃ MTFKNNHNFNELVLTNEDIRILKNVLEDAVSVYDEYSVCNEESDFAYCL----------- 49
Ith-2ᶲᴺᴹ² ---------MCVKFTDAEIAYIKESVENYSSEFDIY---DDEQELKLKIYEQIMLKIESE 48

Ith-1⁸⁰ᵃ LRDLYTLDSLAISSNNV 66
Ith-2ᶲᴺᴹ² YKDIYLFRLIN----- 59

17.7% identity
31.6% similarity

Tha-1⁸⁰ᵃ MEEKQYLWRYNDIEKRMNELHKKYKELVDIFFGDVIDKNTGYFPFYSGIKYSYA--DLNI 58
Tha-2ᶲᴺᴹ² MEENKTLKE------YLRNFLEGYKVVENRYIYQFSSNPEAFPFMRKDDYKISIFYLNQ 54

Tha-1⁸⁰ᵃ SFYRGLIYIHGGSEGMQAKKINISVDDVLKRTERSFEKLSEYIKSTFIFEK---VIHDFQ 115
Tha-2ᶲᴺᴹ² SFFEEPCIVVISNDSKLKEIYNFRNTD-IKHLSKHF---TSYIYDSKKYVEEQSGLLDFN 110

Tha-1⁸⁰ᵃ MFNLSDLASIYNDDAIAHYYFETHSDLYLSDK---------STKIYRTPNSLLENANDLPA 167
Tha-2ᶲᴺᴹ² --NYIYYTSIYYGKYIGTVIIQNNLDLFFNYGKRLANDHYNTLISKSKEKLINKAHDEIQ 168

Tha-1⁸⁰ᵃ S--LRKYTQLLETVNDKDFETHIVEAYDCFMSEKRLATSLLLGRALELMCRLILNKFDKD 225
Tha-2ᶲᴺᴹ² PFNHLDLNSMKKIVDDITFSYQIEQGLQAYKRELYLPAAATFAVAIETFL-IKLKKVN-- 225

Tha-1⁸⁰ᵃ IIKQTPDYKRNIRTFLNEMENNDLIEEHLKHSVKAAIEHRNSIHGI----KIEEYNSII 281
Tha-2ᶲᴺᴹ² KIKHKDTDSTMYTKLLGELTKEGKVNYRTKKRVEIAYSNRNIINHSQAGAVAKGDCDFLL 285

Tha-1⁸⁰ᵃ QTLF---DEIAKL-SNVYKSLNKQ- 301
Tha-2ᶲᴺᴹ² NTLKDIVDENERILTEYSKSINKTE 310

21.4% identity
35.8% similarity

**c**

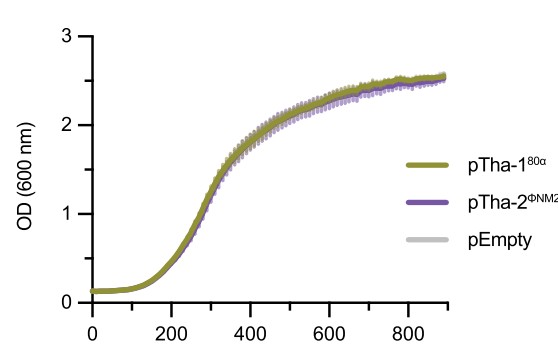

OD (600 nm)

0    200   400   600   800

— pTha-1⁸⁰ᵃ
— pTha-2ᶲᴺᴹ²
— pEmpty

**d**

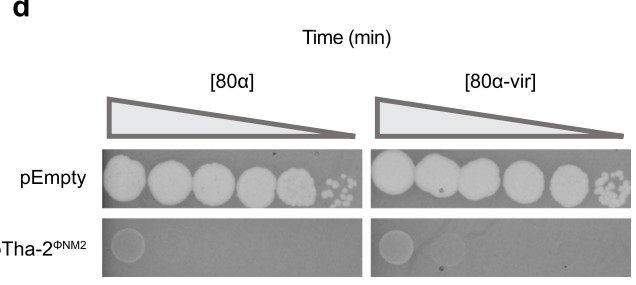

Time (min)

[80α]　　[80α-vir]

pEmpty

pTha-2ᶲᴺᴹ²

**e**

CFU/10⁶ PFU

1000

800

600

400

200

n.s.

pTha-2ᶲᴺᴹ²  pEmpty

**Extended Data Fig. 2 | See next page for caption.**

**Extended Data Fig. 2 | S. aureus phages have different gene configurations in the cI operon, and Tha does not block lysogenisation. a**, Maps showing the genomes of phages 80α (43.9 kilobases (kb)) (NC_009526) and ΦNM2 (43.1 kb) (DQ530360.1) with the noncontiguous operons highlighted. The black arrows indicate the lysogenic *cI* promoter and lytic *cro* promoters. 80α and ΦNM2 have similar noncontiguous operon configurations (highlighted). The scale bar represents 1 kb. **b**, Alignments of the 80α and ΦNM2 variants of Ith-1/2 (Xis) (17.7% identity, 31.6% similarity) and Tha-1/Tha-2 (ORF3) (21.4% identity, 35.8% similarity). The predicted catalytic residues of their respective HEPN domains are highlighted in red boxes (R265 and H270 for Tha-1[80α], R265 and H270 for Tha-2[ΦNM2]). **c**, Growth curves of *S. aureus* RN4220 cells harbouring plasmids expressing either Tha-1[80α] or Tha-2[ΦNM2], or an empty plasmid (pEmpty). No difference in growth is observed with the expression of these Tha variants from plasmids, suggesting the phage protection observed in Fig. 2a is not due to general toxicity. Each datapoint represents the mean of three biological replicates ±SD. The x-axis represents time (min). **d**, Staphylococci harbouring either an empty plasmid, or a plasmid expressing Tha-2[ΦNM2], are infected on soft agar by ten-fold dilutions of either WT 80α or a lytic mutant of 80α (80α-vir). This shows that Tha-2[ΦNM2] immunity is not caused by 80α lysogenisation. **e** *S. aureus* RN4220 harbouring a plasmid expressing Tha-2[ΦNM2] (pTha-2[ΦNM2]) or an empty plasmid (pEmpty) are infected by 10[6] 80α phage particles encoding an antibiotic resistance marker. Resulting colony forming units (CFUs), stemming from successful lysogeny by 80α, are shown. While pTha-2[ΦNM2] protected against 80α in **d** and in Fig. 2a, no significant change in the lysogenisation efficiency is seen, demonstrating that Tha immunity is activated during the lytic cycle. All bars represent the mean of three biological replicates, error bars represent SD. PFU, plaque-forming units; n.s., not significant (by unpaired, two-tailed t-test).

**a** 80α

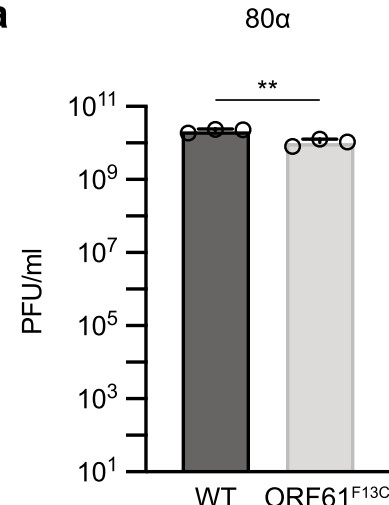

**b** Φ11

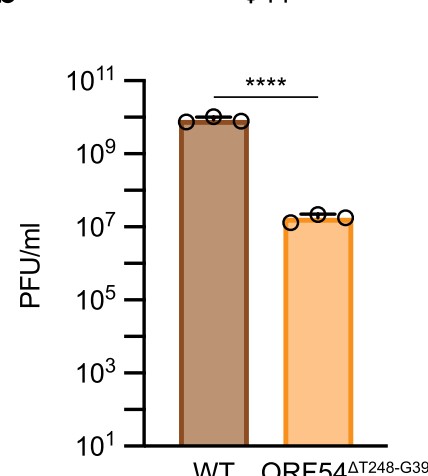

**Extended Data Fig. 3 | Tha escaper phages display reduced fitness. a**, Phage titres following prophage induction of 80α expressing WT ORF61 or the escaper ORF61$^{F13C}$. **b**, Same as **a**, but with Φ11 phages carrying either WT ORF54 or the escaper ORF54$^{ΔT248-G396}$. These phage mutants were difficult to obtain, and display reduced titres upon induction, demonstrating that it is difficult and costly for phages to generate mutations in conserved structural proteins to evade recognition by Tha-1/2. For **a-b**, all bars represent the mean of three biological replicates, error bars represent SD. Unpaired, two-tailed t-tests are used to determine significance. p values: **<0.01, ****<0.0001. p values are 0.0074 for **a**, <0.0001 for **b**. PFU, plaque-forming units.

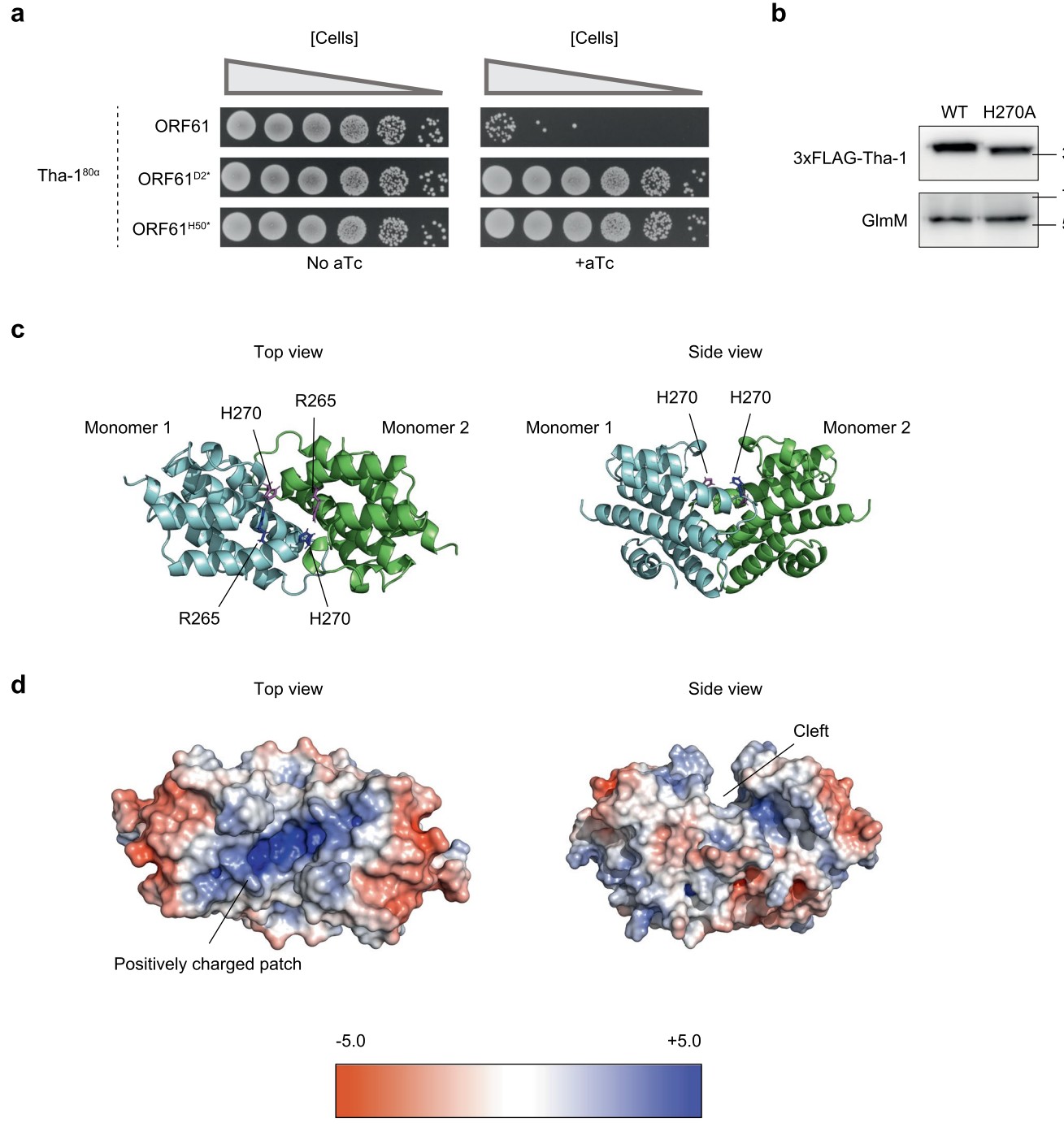

**Extended Data Fig. 4 | Tha-1 is activated by the ORF61 protein, not RNA, and the dimeric structure of the 80α Tha-1 HEPN domain dimer, as predicted by AlphaFold2. a**, Like in Fig. 2d, but with *orf61* genes containing stop codon in either position 2 (D2*) or position 50 (H50*). These mutants are unable to activate Tha-1, suggesting that the Tha-1 activator is a protein, not an RNA. **b** A western blot of either WT or the catalytic mutant (H270A) of Tha-1, both tagged with N-terminal 3xFLAG tags and constitutively expressed from plasmids, showing similar expression levels. GlmM is used as a loading control. Locations and sizes of standard protein marker are indicated. This experiment was performed once. **c**, A cartoon representation of the predicted homodimeric structure is shown of the HEPN region of Tha-1$^{80\alpha}$ (residues 164–301), which were confidently predicted by AlphaFold2 (pLDDT>90). Top views and side views are shown, coloured according to chain. The remaining Tha-1 regions (residues 1–163) would be below when looking from the side view. The side chains of the predicted catalytic residues arginine (R) 265 and histidine (H) 270 are highlighted. **d**, The same dimer as **c**, coloured according to surface electrostatic potential (-5 to +5, negatively charged to positively charged). The top view emphasises the positively charged patch near the putative active site, suitable for binding negatively charged substrate RNA. The side view shows the cleft between the subunits where catalysis likely takes place.

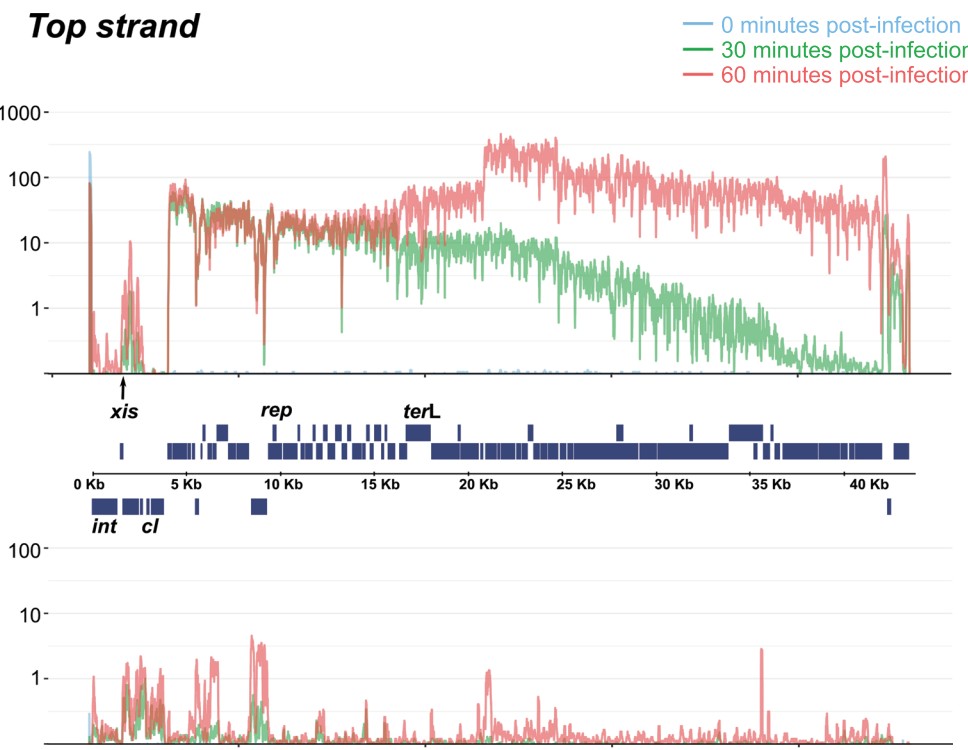

**Extended Data Fig. 5 | Transcriptomics of phage 80α during infection.**
Transcriptomics of 80α during infection by RNAseq, mapped to the top DNA strand (top) and bottom strand (bottom). RNA was harvested right after infection (0 minutes) (blue), 30 minutes (green), and 60 minutes after infection. Reads are normalised to the number of aligned reads. *xis* refers to the *ith-1* gene, and *ith-1* is transcribed shortly after infection, consistent with early blocking of Tha-1-mediated immunity (see Fig. 4b). *rep*, replicase; *terL*, large terminase subunit; *int*, integrase; Kb, kilobase.

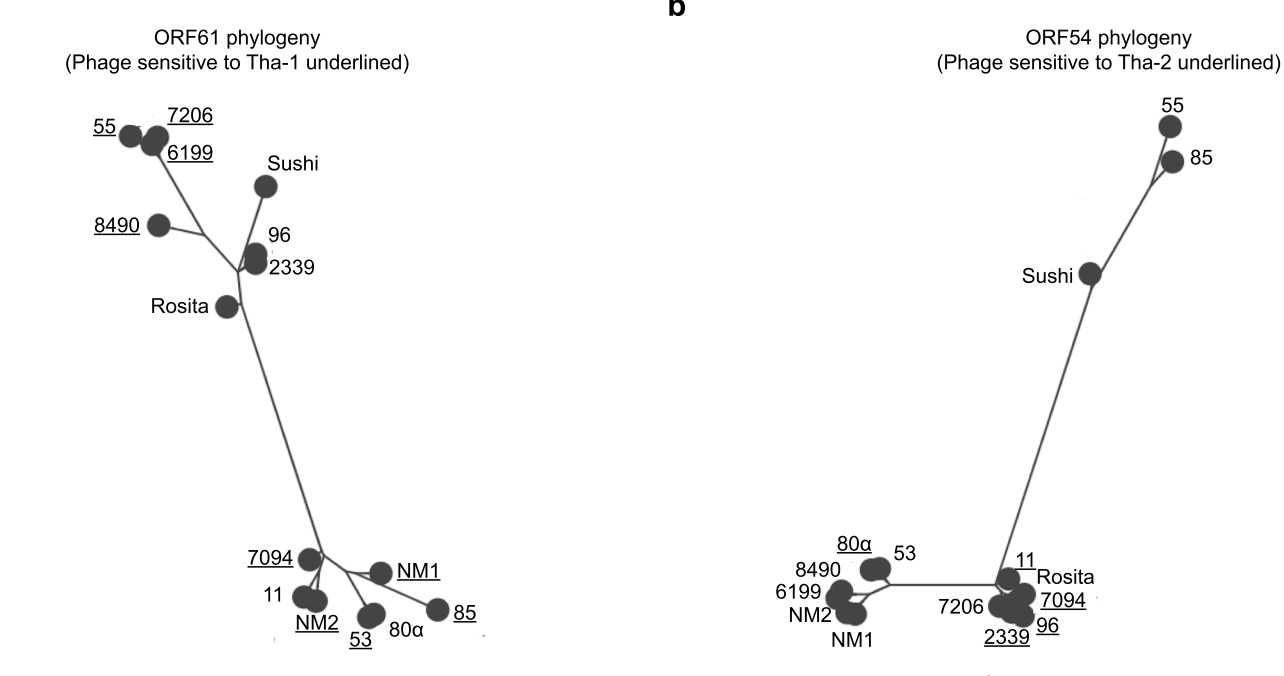

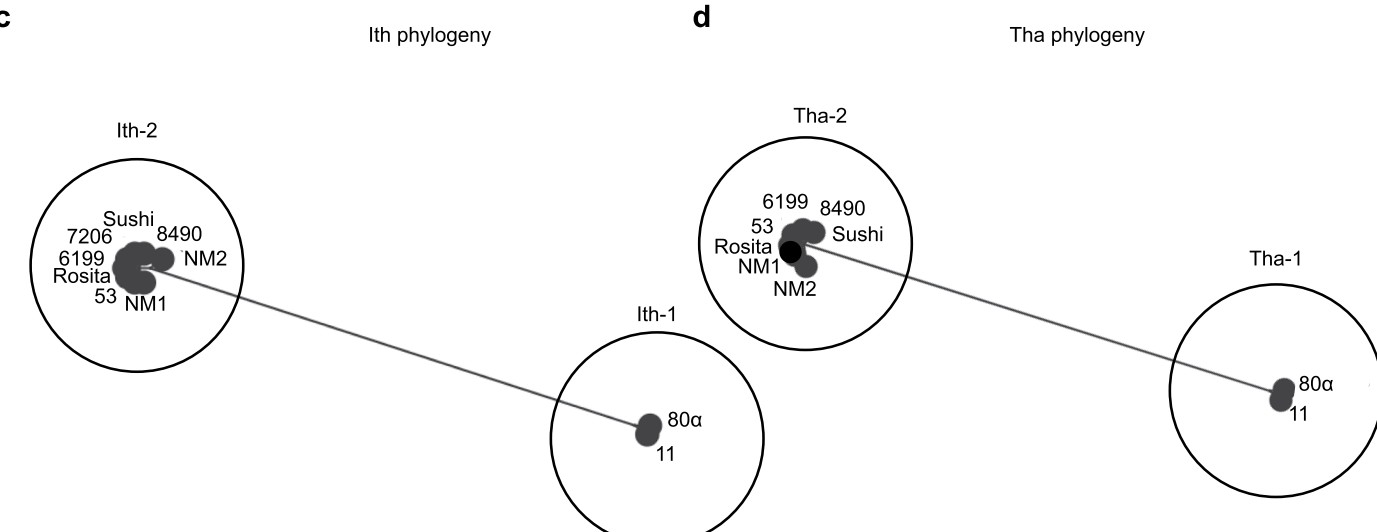

**Extended Data Fig. 6 | Phylogenetic analyses of salient proteins from the phages used in Fig. 2a.** FastTree was used to create maximum likelihood phylogenetic trees of protein homologues of ORF61$^{80\alpha}$ (Tha-1 activator) (**a**), homologues of ORF54$^{\Phi11}$ (Tha-2 activator) (**b**), Ith-1/2 (**c**), and Tha-1/2 (**d**). Phages not represented lack clear homologues to the respective proteins. In **a**, phages sensitive to Tha-1$^{80\alpha}$ in Fig. 2a are underlined, and in **b**, phages sensitive to Tha-2$^{\Phi NM2}$ in Fig. 2a are underlined. Phages Φ12, ΦSLT, Stab21, and K, which are not targeted by Tha-1/2 in Fig. 2a, lack ORF61/ORF54 homologues. One phage, Φ7206, encodes an Ith-2 protein but not a Tha-2 protein, *making ith-2* strictly an immune evasion factor used to infect *tha-2*-positive lysogens (**c-d**). Overall, these analyses suggest that the susceptibility of a phage to Tha-1/2 (Fig. 2a) depends less on the exact ORF61/ORF54 minor tail homologue sequences, and more on the presence of Ith-1/Ith-2, which can function as immune evasion proteins. See also Supplementary Data 1b.

**a**

**Lysogenic cycle**

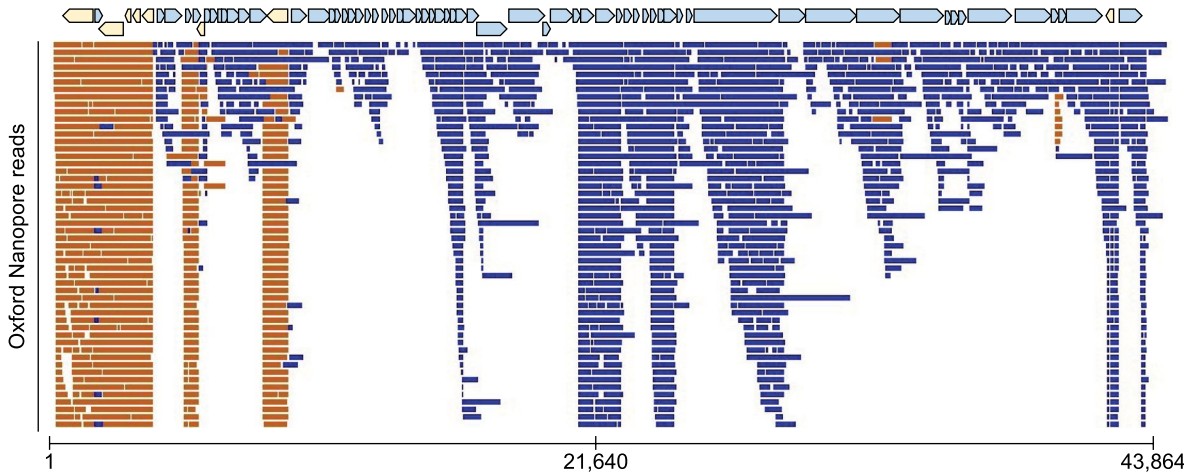

**b**

**Lytic cycle**

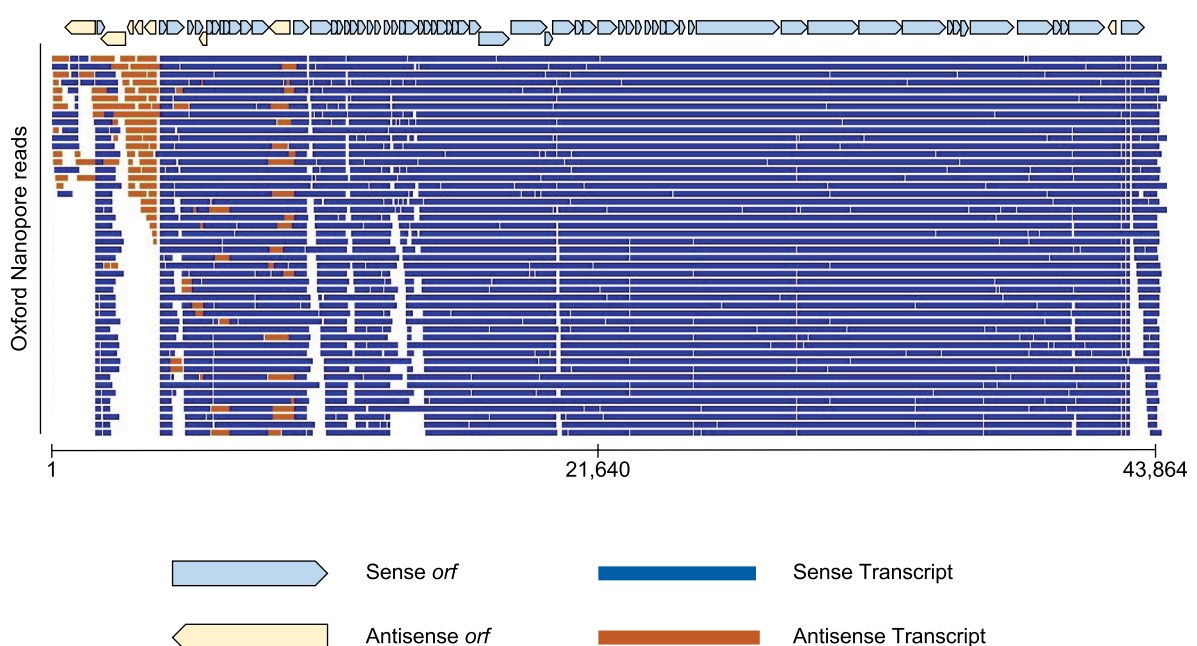

**Extended Data Fig. 7 | Transcription landscape of 80α during the lysogenic and lytic life cycle stages.** Long-read Oxford Nanopore Direct RNAseq of purified RNA from an *S. aureus* RN450 80α lysogen (**a**) or from 80α upon prophage induction by MC (**b**). Each read is coloured according to positive/negative strand mapping. During lysogeny (**a**), the *cI* promoter drives high expression of the operon, including *tha-1*, which is mostly turned off during the lytic cycle (**b**), when the lytic region of the phage is upregulated under the control of the *cro* promoter. *orf*, open reading frame.

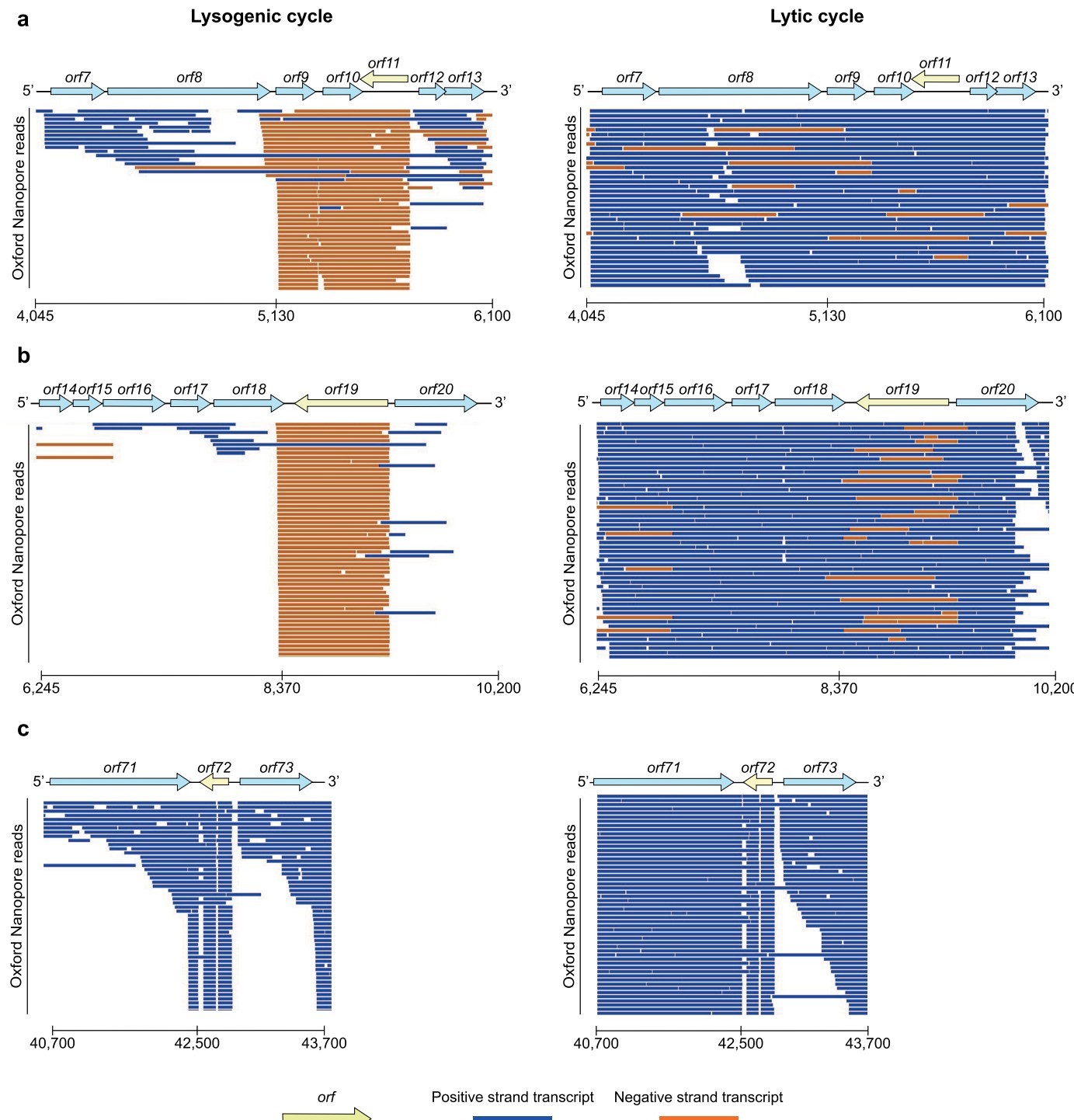

**Extended Data Fig. 8 | Snapshots of the transcriptional profiles of 80α around other noncontiguous operons.** Oxford Nanopore Direct RNAseq of RN450 containing 80α during lysogeny (left) and after induction by MC (right), with noncontiguous operon regions around *orf11* (**a**), *orf17* (**b**), and *orf72* (**c**) displayed, with each antisense gene being shown in yellow. For *orf11* and *orf19*, there is a clear switch in the transcription profiles after prophage induction

(right), likely helped by the noncontiguous genetic architecture. For *orf72*, we failed to observe negative sense transcripts during lysogeny and lysis, maybe due to low promoter activity, or the promoter only being active under certain circumstances. *orf*, open reading frame. Overall, these data highlight how bacteriophages can use noncontiguous operons with antisense genes to regulate gene expression in the lysogenic and lytic life cycles.

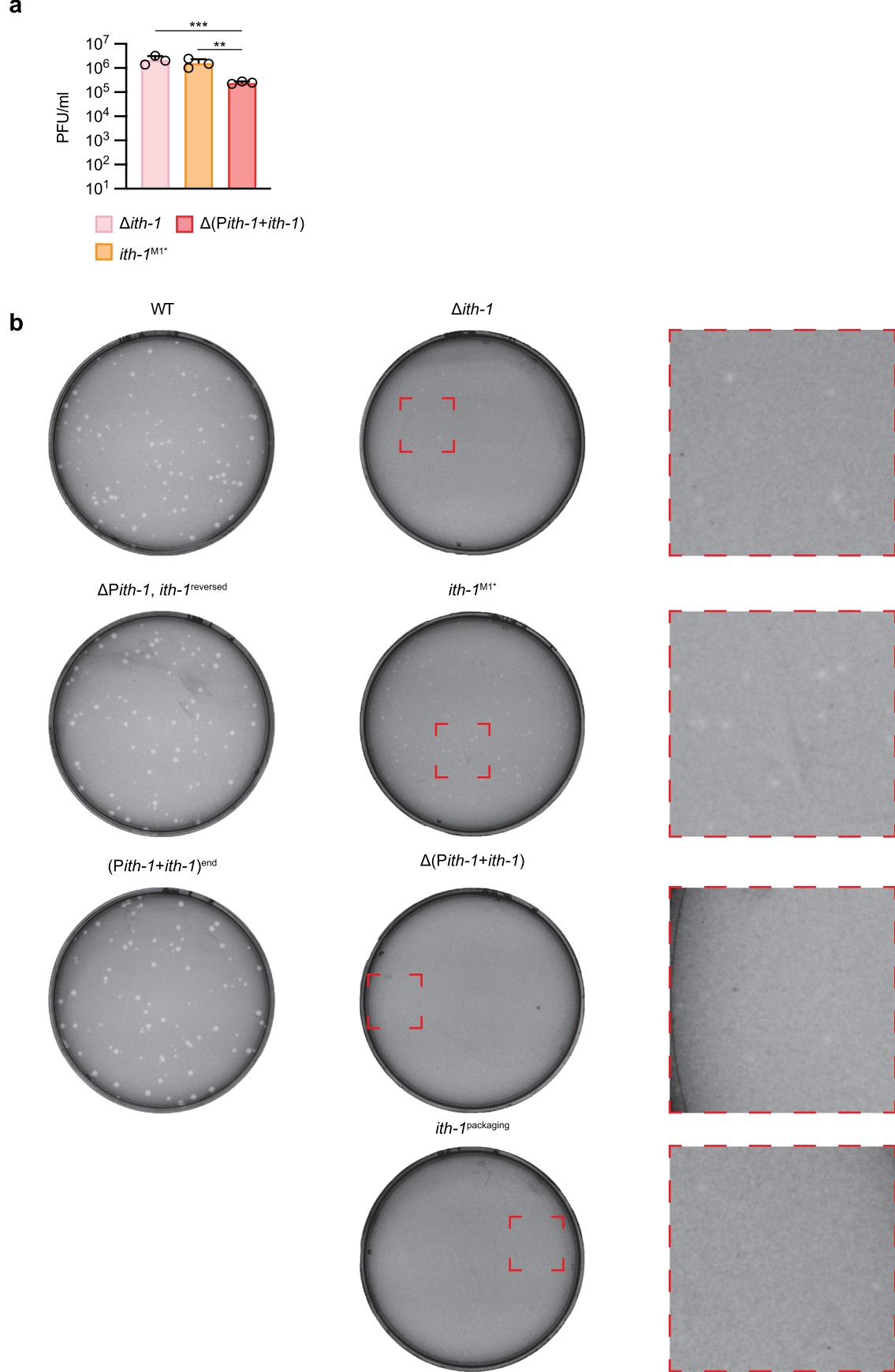

**Extended Data Fig. 9 | See next page for caption.**

**Extended Data Fig. 9 | Induction in an RNase III deletion background, and plaque sizes resulting from prophage induction of noncontiguous operon mutants. a**, Prophage induction of *S. aureus* RN4220 cells containing the 80α mutants indicated, in a strain where RNase III is deleted (Δ*rnc*). We hypothesised that with antisense transcriptional regulation, the regulation could stem from either dsRNA degradation by RNase III, or by RNA polymerase collisions. To distinguish between these possibilities, we tested mutant induction in a RNase III knockout strain. The difference between Δ*ith-1*/*ith-1*^MI* and Δ(P*ith-1*+*ith-1*) is similar to the difference in Fig. 6b, indicating that dsRNA degradation by RNase III is not the main mechanism of Tha-1 inhibition from the *ith-1* promoter (P*ith-1*). If dsRNA degradation was the main factor, this difference would be expected to

disappear in a Δ*rnc* background. Instead, the regulation is likely a result of RNA polymerase head-on collisions. All bars represent the mean of three biological replicates, error bars represent SD. One-way ANOVA with multiple comparisons was used to compare each mutant to Δ(P*ith-1*+*ith-1*). Adjusted p values: **<0.01, ***<0.001, 0.0006 for Δ*ith-1*, 0.0014 for *ith-1*^MI*. **b**, Representative images showing plaque sizes of the WT 80α or 80α variants with mutated operonic structures, relating to Fig. 6b. For the small plaques, a zoomed in image is displayed to highlight the small plaques. Δ*ith-1* and *ith-1*^MI* have reduced plaque sizes, since Ith-1 protein does not inhibit Tha-1. Δ(P*ith-1*+*ith-1*) and *ith-1*^packaging have even smaller plaque sizes, since Ith-1 protein is not made, and there is no antisense transcriptional interference from P*ith-1*.

# Reporting Summary

## Statistics

For all statistical analyses, confirm that the following items are present in the figure legend, table legend, main text, or Methods section.

| n/a | Confirmed | |
|---|---|---|
| ☐ | ☒ | The exact sample size (*n*) for each experimental group/condition, given as a discrete number and unit of measurement |
| ☐ | ☒ | A statement on whether measurements were taken from distinct samples or whether the same sample was measured repeatedly |
| ☐ | ☒ | The statistical test(s) used AND whether they are one- or two-sided<br>*Only common tests should be described solely by name; describe more complex techniques in the Methods section.* |
| ☐ | ☒ | A description of all covariates tested |
| ☐ | ☒ | A description of any assumptions or corrections, such as tests of normality and adjustment for multiple comparisons |
| ☐ | ☒ | A full description of the statistical parameters including central tendency (e.g. means) or other basic estimates (e.g. regression coefficient) AND variation (e.g. standard deviation) or associated estimates of uncertainty (e.g. confidence intervals) |
| ☐ | ☒ | For null hypothesis testing, the test statistic (e.g. *F*, *t*, *r*) with confidence intervals, effect sizes, degrees of freedom and *P* value noted<br>*Give P values as exact values whenever suitable.* |
| ☒ | ☐ | For Bayesian analysis, information on the choice of priors and Markov chain Monte Carlo settings |
| ☒ | ☐ | For hierarchical and complex designs, identification of the appropriate level for tests and full reporting of outcomes |
| ☒ | ☐ | Estimates of effect sizes (e.g. Cohen's *d*, Pearson's *r*), indicating how they were calculated |

*Our web collection on statistics for biologists contains articles on many of the points above.*

## Software and code

Policy information about availability of computer code

| Data collection | No software was used for data collection |
|---|---|
| Data analysis | No custom code or software was used.<br>Statistical analyses were performed on GraphPad Prism v10.1<br>Clustal Omega v1.2.4 for protein alignments<br>EMBOSS Needle v6.6.0 for calculating protein sequence identity and similarity<br>Alphafold2 was run through Google ColabFold v1.5.2<br>PyMol v2.5.4 for structure visualisation<br>DALI web server (http://ekhidna2.biocenter.helsinki.fi/dali/) (2023) was performed for looking for Tha-1 HEPN domain protein homologues<br>pBLAST 2.14.0 for finding 80α ORF61 homologues/Φ11 ORF54 homologues in phage genomes<br>READemption v0.4.3 RNAseq pipeline for 80α infection RNAseq analysis<br>MAFFT Galaxy v7.508 for protein alignments prior to phylogenetic analysis<br>FASTTREE Galaxy v2.1.10 for phylogenetic analysis<br>Guppy v3.5.1 for base calling during Oxford Nanopore RNA sequencing<br>Geneious Prime 2023 (Biomatters) for read mapping during Oxford RNA sequencing |

For manuscripts utilizing custom algorithms or software that are central to the research but not yet described in published literature, software must be made available to editors and reviewers. We strongly encourage code deposition in a community repository (e.g. GitHub). See the Nature Portfolio guidelines for submitting code & software for further information.

## Data

Policy information about availability of data

All manuscripts must include a data availability statement. This statement should provide the following information, where applicable:
- Accession codes, unique identifiers, or web links for publicly available datasets
- A description of any restrictions on data availability
- For clinical datasets or third party data, please ensure that the statement adheres to our policy

The Oxford Nanopore RNA sequencing data is available through BioProject PRJNA922758. The 80α infection RNAseq data is available from Sequence Read Archive accession code PRJNA1080480. The 80α genome is available through NC_009526, the ΦNM2 genome through DQ530360.1, and the other phage accession codes are listed in Extended Data Excel file 1b. The remaining data is available in the text/figures/extended materials, and raw data and materials/strains upon request from J.T. Rostøl (j.rostoel@imperial.ac.uk) or J.R. Penadés (j.penades@imperial.ac.uk).

## Research involving human participants, their data, or biological material

Policy information about studies with human participants or human data. See also policy information about sex, gender (identity/presentation), and sexual orientation and race, ethnicity and racism.

| | |
|---|---|
| Reporting on sex and gender | N/A |
| Reporting on race, ethnicity, or other socially relevant groupings | N/A |
| Population characteristics | N/A |
| Recruitment | N/A |
| Ethics oversight | N/A |

Note that full information on the approval of the study protocol must also be provided in the manuscript.

# Field-specific reporting

Please select the one below that is the best fit for your research. If you are not sure, read the appropriate sections before making your selection.

☒ Life sciences  ☐ Behavioural & social sciences  ☐ Ecological, evolutionary & environmental sciences

For a reference copy of the document with all sections, see nature.com/documents/nr-reporting-summary-flat.pdf

# Life sciences study design

All studies must disclose on these points even when the disclosure is negative.

| | |
|---|---|
| Sample size | Pre-determining sample size was not performed for this study. Quantitative experiments were performed three times to allow statistical analysis of significance, which provided robust differences between samples. |
| Data exclusions | We did not exclude any data |
| Replication | All experiments were performed with three biological replicates, unless stated otherwise. All attempts at replication were successful. Performed twice - Fig. 4a-c, Extended Data Fig. 2d. Performed once - Fig. 3c, d, Extended Data. Fig. 1e, g, Extended Data. Fig. 4a-b, and RNAseq experiments |
| Randomization | Samples were not randomised since we employed well-defined experimental strains with isogenic genetic backgrounds, comparing effects of single, known variables. No animals or patients were used in this study. |
| Blinding | Blinding was not performed since we used well-defined experimental strains, and results had strong effect sizes. The same researcher prepared samples and performed experiments. No animals or patients were used in this study. |

# Reporting for specific materials, systems and methods

We require information from authors about some types of materials, experimental systems and methods used in many studies. Here, indicate whether each material, system or method listed is relevant to your study. If you are not sure if a list item applies to your research, read the appropriate section before selecting a response.

## Materials & experimental systems

| n/a | Involved in the study |
|---|---|
| ☐ | ☒ Antibodies |
| ☒ | ☐ Eukaryotic cell lines |
| ☒ | ☐ Palaeontology and archaeology |
| ☒ | ☐ Animals and other organisms |
| ☒ | ☐ Clinical data |
| ☒ | ☐ Dual use research of concern |
| ☒ | ☐ Plants |

## Methods

| n/a | Involved in the study |
|---|---|
| ☒ | ☐ ChIP-seq |
| ☒ | ☐ Flow cytometry |
| ☒ | ☐ MRI-based neuroimaging |

## Antibodies

| | |
|---|---|
| Antibodies used | Monoclonal anti-FLAG M2-Perixodase (HRP) antibody from mouse, from Sigma-Aldrich. Catalogue number A8592, lot number SLCF0816 (1/1,000 dilution)<br>Custom anti-GlmM polyclonal antibodies from rabbit, from Covalabs (project number 1846011) (1/2,500 dilution)<br>Anti-rabbit IgG, HRP-linked antibody, from Cell Signal. Catalogue number 7074S, lot number 26 (1/10,000 dilution) |
| Validation | The anti-GlmM antibody was validated by Western blot in prof. Angelika Gründling's laboratory (Imperial College London), where native S. aureus GlmM and His-tagged S. aureus GlmM produced a clean signal at the right size, and E. coli GlmM expressed in S. aureus produced no signal. |

