## [Peer Review File · Nature Microbiology]

Peer Review Information

Journal: Nature Microbiology

Manuscript Title: Bacteriophages avoid autoimmunity from cognate immune systems as an intrinsic part of their life cycles

Corresponding author name(s): Professor José Penadés

Reviewer Comments & Decisions:Decision Letter, initial version:

Message: 24th May 2023

Dear Professor Penadés,

Thank you for your patience while your manuscript "Bacteriophages use noncontiguous operons to avoid autoimmunity from cognate immune systems" was under peer review at Nature Microbiology. It has now been seen by our referees, whose expertise and comments you will find at the end of this email. In the light of their advice, we have decided that we cannot offer to publish your manuscript in Nature Microbiology.

From the reports, you will see that the referees disagree about the potential advance and broad interest of your findings. From an editorial perspective, however, we tend to agree that your work falls into the exciting area of phage defense. Nevertheless, in this case we were concerned that Reviewers #1 and #2 agreed that the work was lacking biochemical characterization of your system which would be essential for ensuring the robustness of your conclusions and positing a solid mechanism, which we are in agreement would be necessary. From these reports, it is clear that a significant amount of additional experimental work will be necessary in order to support your key claims, and it is far from clear whether the results will continue to appear impressive in the light of this additional work. While we would not rule out consideration of a revised manuscript that makes a much stronger case in support of these claims, we feel that, at this stage, the present work is at too preliminary a stage to warrant publication in Nature Microbiology.

I am sorry that we cannot be more positive on this occasion, but hope that you find the referees' comments helpful when preparing your paper for resubmission elsewhere.

Yours sincerely,

Reviewer Expertise:

Referee #1: anti-phage defense systems
Referee #2: anti-phage defense systems, prophage
Referee #3: non-contiguous operons

Reviewers Comments:

Reviewer #1 (Remarks to the Author):

Rostøl et al. examine two uncharacterized genes within the cI/int operon of a lysogenic lambdoid phage in *S. aureus*, 80α. Tha-1 is a HEPN-containing protein that the authors claim is a generalized phage-defense system. The other gene, transcribed in the other direction within this region, does not encode an excisionase as annotated, but instead the region is involved in regulating Tha-1, possibly through an open-reading frame, xis. Growth arrest via Tha-1 can be activated by expression of ORF61, suggesting this is what is detected to activate the defense system in response to foreign phage infection. Surprisingly, this putative defense system targets itself during lytic growth and the overlapping antisense transcript allows the phage to avoid autoimmunity during its own lytic cycle.

Overall, the authors use many clever genetic experiments to test hypotheses, but there is a significant lack of biochemistry to support their model. Additionally, though the paper describes an interesting potential defense system/prophage exclusion mechanism, I am unclear how this paper has a broad impact. The authors advance the concept of "noncontiguous operon" as the major impact of the paper. Regarding this:

a) I am wary of how widely accepted the term "non-contiguous operon" is in the scientific community. For that reason, I personally don't want to endorse this terminology because it is antithetical to the definition of an operon as a single transcript.

b) It is not as surprising or novel as the authors communicate to the reader that there are transcripts on the opposite strand of operons, especially given the compaction of bacteriophage genomes. The notion that they have functions to the phage is also not as novel or surprising as the authors state. If they didn't, why would they be transcribed?

c) The authors do not thoroughly demonstrate experimentally (as they do in the referenced PNAS paper) that this is what they call a "noncontiguous operon". They do not concretely show biochemically that either the opposing transcript or an unconfirmed Xis gene product regulate the function of Tha-1. Though they provide several genetic experiments, by making deletions, etc. to the antisense region, they are inherently changing the sense transcript. To tease apart these functions I feel I need more biochemical evidence of what is taking place.

For one example, although an ORF is annotated (xis) the authors never provide concrete evidence that a Xis protein is translated. The authors mention in passing that they tried to express the protein and were unable to, but give no detail beyond that. Without such evidence, I am left wondering if this is just a case of anti-sense transcription and dsRNA degradation, downregulation of the cI/int promoter, some role for the Xis protein (direct protein-protein inhibition of Tha-1, stabilizing dsRNA transcripts for degradation, further down-regulating the cI/int promoter, activating degradation of Tha-1), a combination, or something else entirely. Without knowing if a protein is even produced, the model becomes unclear. Since this is a key message of the paper, I would ask for more biochemical data supporting the model, and more experiments to investigate the function of Xis.

Major comments:

1) As stated above, the authors do not convincingly demonstrate that the ORF xis is translated. First, the protein should be epitope tagged and shown to be expressed on a Western blot. Alternatively, they could fuse GFP or another reporter to xis or the N-terminus of xis. Second, co-expressing Tha-1 and ORF61 is sufficient to activate and block growth – so the authors should test whether over-expression of Xis in this context can block toxicity. Preferably this would be with xis synonymously recoded so there is minimal effect from any antisense transcripts. A control with a mutated start codon could be tested as well. Similarly, if you engineer 7206 to overexpress Pxis+xis (+/- rest of antisense transcript), does this overcome the protective effect of Tha-1 on 7206?

They do replace the predicted Xis start codon with a stop codon and this strain fails to lyse. However, this alone is not thoroughly convincing. Although minor, a cleaner experiment would be to mutate the start codon, as opposed to adding a stop codon in its place.

In figure 5b, Δ Pxis, xisreversed, it would be helpful to see what a start codon mutation looks like in this assay. Also, to be thorough, it would be nice to see that Tha-1 is still expressed in this context so that the result is due to expression of Xis rather than interfering with expression of Tha-1 in some way. This is a running issue throughout as the authors make many artificial genetic constructs but are unable to directly see what this is doing to expression of gene products. This leads to many assumptions being made about how the genetic constructs should probably operate but not how they do operate.

2) As the authors show, at least some of the effect of the xis transcript on Tha-1 is likely due to antisense transcripts and degradation of dsRNA by RNase III. If RNase III is not essential in Staph, the authors could test regulation of Tha-1 in an RNase III KO or knock-down. This may allow them to ascertain how much regulation of Tha-1 is through RNA degradation and how much is due to a potential Xis protein. Somewhat related, it may also be informative to knock-down the xis transcript during lytic growth and see if the 80a can still form particles/plaques. That would tease apart how much regulation of Tha-1 is due to lowering expression and would be a nice companion to the deletion mutants.

3) Though I agree that Tha-1 by all available evidence is likely a toxic RNase, it would be simple to co-express Tha-1 and ORF61 and look at what is happening to RNA on a gel. I understand that finding the specificity of the cleavage might be beyond the scope, but to see RNA being cleaved is important.

4) ORF61 activates the defense system, but according to the phage panel data, pTha-1 blocks many phages. Do all of these phages have a homologous gene to ORF61? If not, please explain.

5) In the phage panel data, as the authors note, Tha-1 naturally expressed within the genome offers little protection compared to the pTha-1 plasmid. As such, this is an artifact of overexpression and the broad anti-phage language should be toned down.

6) Regarding the phage plaquing data – since these phages are lysogenic, is it possible that

rather than providing protection the phages are integrating? Can lytic variants of these phages be tested?

7) The authors mention that this system could provide abortive infection. Typically, this is tested by infecting a strain containing the system with phages at MOIs both greater than and less than one. The experiments shown here are at MOI of 1 and 10. These MOIs should elicit the same general growth pattern if the system works by abortive infection as all bacterial cells would be infected. Please clarify.

8) Perhaps this is an artifact of the methodology or an issue with the figure (Fig. 4c), but the Nanopore RNA reads during lytic growth do not appear to contain the Xis start codon (and definitely not an RBS) as shown in figure 4C.

9) Typically, phage defense systems are conserved in many contexts, and those found in prophages are also present in "defense islands" to some extent. Tha-1 appears to be quite rare and only found in a few *S. aureus* prophages. Is it possible that this is an 80 α -specific super-infection exclusion mechanism, or perhaps a mechanism that blocks spurious lytic induction of 80 α ? The authors do not discuss this possibility. Relatedly, I also find it curious that the system targets itself, rather than recognizing some conserved region of foreign phages. Often, prophage encoded anti-phage systems do not target themselves.

10) It is unclear how diverse the phages in the tested phage panel are. Are they related to 80 α ? Something to indicate how related they are would be nice, e.g. a phylogenetic tree, whole genome alignment, etc. For all the reader knows these are all very similar phages.

Minor issues:

Lines 62-64: "However, no concrete examples have been described where the noncontiguous operon is advantageous for the regulation of a specific biological process." – The referenced PNAS paper shows that this organization regulates menaquinone biosynthesis.

Line 75: "labelled" should be "annotated as"

Line 83: "lamboid" should be "lambdoid"

Line 199: "data now shown" should be "data not shown"?

Line 480: "decreased fitness cost" should be either "decreased fitness" or "increased fitness cost"

Fig 3a: Please invert the colors to distinguish this from a plaque assay.

Fig 4a: Grid lines would be nice.

Fig. 4b and c: It would be nice to see total coverage in both directions rather than aligned reads (are these all the reads that mapped?).

Fig 5c: The caption says green and blue, I'm seeing green and orange.

Fig 5: A schematic similar to Fig. 1a, but with the position of Pxis, xis would be helpful to follow the several different constructs.

Perhaps the authors should rename xis since they show it is likely not an excisionase.

Reviewer #2 (Remarks to the Author):

In this study, Rostøl et al. identify a new prophage-encoded anti-phage defence system that they call THA (Tail-activated, HEPN domain-containing Anti-phage system). They show that this system is encoded together with a cognate "immunity" protein that allows phages expressing it to bypass the defence and replicate normally. The immunity protein for this system was previously mistakenly annotated as an excisionase protein. The authors provide strong evidence that it is not an excisionase and does in fact act as an immunity protein that inhibits the activity of the THA system. The THA systems encoded in *S. aureus* phages are predicted to function as RNAses due to the presence of the HEPN domain, and the activity of the characterized system is hypothesized to induce growth arrest, preventing phage replication and protecting the bacterial population. The study of anti-phage defence mechanisms is of high interest in the microbiology community at the present time, so this work will be of quite broad interest.

Overall, this is a well-presented story; importantly the authors examined the activity of the

system in the context of the prophage and have provided convincing data about how the system functions with respect to the inhibition provided by THA and the reversal of this inhibition by the Xis protein. However, there is a lack of insight into how this system is actually functioning to block phage replication. At which stage of the phage life cycle is the inhibition occurring? Does phage genome replication occur normally? Are phage virion intermediates produced? It appears that the cells are not lysed, so is there is some block to phage late gene expression (maybe only lysins)? It appears that a trigger for activity is located in the phage minor tail proteins, but it is not clear if this is a protein trigger, or an RNA sequence trigger. Do the authors have evidence that there is a cleavage site in this region of the phage genome (i.e. do the mutations provide evasion of the system at the protein or RNA level)? In addition, it's not clear if the growth arrest noted is reversible or irreversible, as might be expected for an Abi system. I would expect that more detailed insight into the specific mechanism of anti-phage activity be provided for publication in Nature Micro.

Major concerns:

I found the term "regulator" used in reference to the activity of the Xis protein (e.g., lines 29, 74-76) confusing as this brings to mind transcription factor type regulatory proteins. At the protein interaction level these types of biological interactors are usually referred to as inhibitors. The authors should consider changing this terminology.

The MOI of 10 used in Fig. 2b did not result in complete cell lysis or premature cell lysis. Did the authors determine if the CFUs/ml are changing throughout infection; are the cells lysing or is there just bacteriostatic activity that is relieved at some point? Of the cells that regrow, what are they? Resistant mutants, lysogens?

Quantification of changes in mRNA during phage infection of 7206 in a wild-type vs. Tha-1 prophage mutant should be done to determine if there is an arrest in all cellular activities, or if there is mainly a loss of transcription of phage products.

The authors must show that the H270 mutant is expressed and soluble to the same levels as wild-type THA. While the His residue may be involved in nucleic acid cleavage, mutation of this histidine residue may also lead to decreased protein expression/stability or solubility. A Western blot would provide evidence that it is the specific loss of enzymatic activity rather than instability/insolubility that causes the loss of activity.

Do phages that bypass THA defence encode a similar Xis, or is it strictly differences in the minor tail protein target sequence? Some additional alignments would help paint a clearer picture of the relationship between the Xis and Tha proteins. The escape mutants were done in a plasmid-based system and don't provide biological context if Xis is able to allow phage infection in the presence of Tha from a prophage. Can the repressor of another phage be exchanged with 80a and see if 80a(new repressor) can plate on a wild type 80a lysogen? Or can the Xis from 80a be put into 7206?

Are the Xis proteins broadly active or are they Tha sequence specific? Co-expressing the 80a and NM2 Xis/Tha-1/2 in the presence of bypassing phage would add knowledge for Tha functionality. Alternatively, can a chimera or truncated form of Tha be generated to determine if the N-terminal domain is essential for recognition of the phage tail proteins?

Minor concerns:

Figure 1a – Reference to the significance of the * and ** should be made in the figure legend.

Figure 2a - The blank spaces make this heatmap difficult to read. Do the clear/white boxes represent no inhibition at all and the clear boxes with * mean infection to wild-type levels but the plaques were smaller? It should be clearly stated if these data are PFU/mL or fold-inhibition.

Figure 3 – It would be easier for the reader if this figure was inverted; as presented these look like plaques, not colonies.

Figure S1b – There are two orf3's noted.

Line 32 – Noncontiguous is spelled incorrectly.

Line 155-162. This could be reworded to improve clarity. The movement between prophages and plasmid-expressed protein (I assume that is what Tha-1 alone means) is confusing. It would be useful to state how many phages were inhibited only by Tha-1 from the prophage.

Line 390 – There is an empty pair of brackets.

Supplemental Figure 1g – The labels say Φ 11 but the figure legend says 80a.

Reviewer #3 (Remarks to the Author):

I have reviewed the manuscript. I find the science presented in the paper fascinating. Although the authors emphasized that their findings represent a new mechanism of phage exclusion, e.g. Crispr-CAS, I believe the data reveals a fascinating mechanism of temperate phage biology. The experiments are performed logically with clear results. I recommend publication of the manuscript in Nature. But in my opinion the manuscript as presented is too long and wordy. I suggest it should be thoroughly edited for brevity and crispy reading. I also have some comments that the authors may respond to before the manuscript is accepted. My comments are not in order of chronology or in terms of importance although some are more important than others:

1. Of course, as I mentioned, it should be thoroughly edited.
2. I don't know why the gene referred to as xis was called so – may be because of homology or gene location – but since the authors have shown that it does not function as the 'classical' excisionase, I suggest to change it to one that more reflects its demonstrated function.
3. The triangular symbols at the top of, for example, figure 3 is misleading the way they are presented. If they reflect Dilution as noted, then it should be horizontally inverted reflecting more and more dilutions; as presented they reflect phage titers, not dilutions.
4. The manuscript misleadingly and frequently uses the word Tha1 when they really mean Tha+ wild type situations. Please carefully edit the entire manuscript about this point. Was the mutant Tha1 gene cloned in the plasmid? In line 166, the authors certainly mean, "... In the absence of WT Tha gene". Also, in line 143, same problem.
5. What kind of mutant is Tha1? Base-substitution? Deletion?
6. I strongly disagree with the use of the word non-contiguous operon. It is very misleading word to describe the case it refers to. I Insist that the authors use a different word to describe overlapping antisense operons.
7. Line 121. "... We sequenced five 80a mutant phage"
8. Line 422. Shouldn't the statement "Tha targets tail proteins" be "tail protein targets Tha?"
9. Are the authors using the word "moron" in the sense that Roger Hendrix defined it? Whatever the authors are talking about, please clarify and give a reference.
10. The Authors may point out that ORF61 has a dual role: tail protein as well as an activator of Tha gene.
11. Lines 317-322. These lines really belong to Experimental Procedures, not in the results section. This is one example. There are other situations in the manuscript where some of the statements belong to Experimental Procedures.
12. Line 417. The paragraph doesn't add much; may be deleted.
13. I also believe some of the discussions talk about evolutionary aspects of the current results. I feel that those are not quite important points in the context of the current findings; may be left out.

Sankr Adhya

** Although we cannot publish your paper, it may be appropriate for another journal in the Nature Portfolio. If you wish to explore the journals and transfer your manuscript please use our manuscript transfer portal. You will not have to re-supply manuscript metadata and files, but please note that this link can only be used once and remains active until used. For more information, please see our manuscript transfer FAQ page.

Note that any decision to opt in to In Review at the original journal is not sent to the receiving journal on transfer. You can opt in to In Review at receiving journals that support this service by choosing to modify your manuscript on transfer. In Review is available for primary research manuscript types only.

We would like to express our sincere appreciation to the reviewers for their valuable comments and suggestions. We believe that the new experiments conducted in response to their questions will provide fresh insights into the functioning of the system. We would also like to extend our gratitude to the reviewers, with special recognition for Reviewer 3, as their comments have highlighted a potential limitation in our previous presentation of the relevance of our findings. In *Staphylococcus aureus*, it is noteworthy that the majority of strains harbor at least one prophage, with an average of four prophages per cell. In this context, where a significant proportion of the prophages encode the immune systems characterised in our study and their corresponding anti-autoimmune mechanisms, it becomes evident that these immunity mechanisms (which protect cells against infection by other phages) and measures to prevent autoimmunity (when the prophages become activated) should be integral components in describing the various stages of the prophage life cycle. Now we have evidence that this is not just happening in *S. aureus* but also in other species, such *E. coli* and the prototypical phage 80 (paper in revision at Nature; reference 2023-08-13773).

Traditionally, life cycles have been analysed based on the impact of different processes (such as excision, replication, or packaging) on a phage's ability to reproduce. As demonstrated in our research, alterations in either the immune systems we have characterised or their corresponding mechanisms that regulate their function can profoundly affect the biology of prophages encoding these systems. Therefore, we propose that immunity and counter-immunity should be incorporated into the definition of prophage life cycles. We are grateful to the reviewers for guiding us toward this important perspective and we have now incorporated this idea into the manuscript.

Reviewers Comments

Reviewer #1

Rostøl et al. examine two uncharacterized genes within the *cl/int* operon of a lysogenic lambdoid phage in *S. aureus*, 80 α . Tha-1 is a HEPN-containing protein that the authors claim is a generalized phage-defense system. The other gene, transcribed in the other direction within this region, does not encode an excisionase as annotated, but instead the region is involved in regulating Tha-1, possibly through an open-reading frame, *xis*. Growth arrest via Tha-1 can be activated by expression of ORF61, suggesting this is what is detected to activate the defense system in response to foreign phage infection. Surprisingly, this putative defense system targets itself during lytic growth and the overlapping antisense transcript allows the phage to avoid auto-immunity during its own lytic cycle.

Overall, the authors use many clever genetic experiments to test hypotheses, but there is a significant lack of biochemistry to support their model. Additionally, though the paper describes an interesting potential defense system/prophage exclusion mechanism, I am unclear how this paper has a broad impact.

We thank the reviewer for constructively assessing our work. We have now performed additional experiments to confirm the proposed model.

In accordance with the reviewers' feedback, we have also made adjustments to highlight the importance of our work. As suggested by this reviewer, it may indeed seem surprising that the phage encodes a defense system that targets itself. However, our research has unveiled that this is not an isolated occurrence; rather, it is a prevalent characteristic observed in many Staphylococcal phages, as demonstrated here. Furthermore, we have obtained preliminary evidence of similar phenomena in other species, such as *E. coli*. Consequently, in alignment with reviewer 3's suggestion, we concur with the idea that the phage's immunity mechanisms, along with their mechanisms to counteract this immunity, should be integral components when defining the phage life cycle. We believe that this perspective is both unique and of significant importance, with the potential for broad-reaching impact.

The authors advance the concept of “noncontiguous operon” as the major impact of the paper.
Regarding this:

a) I am wary of how widely accepted the term “non-contiguous operon” is in the scientific community. For that reason, I personally don’t want to endorse this terminology because it is antithetical to the definition of an operon as a single transcript.

The term "noncontiguous operon" was initially employed in the original paper to describe a distinctive genetic organisation wherein within a classical operon, there exists a gene transcribed in the opposite direction. The authors asserted in that publication that this genetic arrangement would somehow enable the unique control of these genes. In our manuscript, we have opted to utilise the established terminology for this genetic configuration and, for the first time, have substantiated the claim regarding the regulatory advantages inherent to this genetic organisation. We appreciate, however, the concern the reviewer has with the heavy emphasis on the term noncontiguous operon in our initial submission. We understand that it is not yet a commonly used term in the scientific literature, and that it might distract from the underlying regulatory concepts we tried to convey with the use of the phrase. As a result, we have removed it from the title, and have used it less throughout the text and figures. Where it is used, it is often used together with “complex transcripts”, the term the eukaryotic virus field uses, or defined as overlapping, antisense transcripts. Yet, we still feel like the noncontiguous operon term has value in the context of our study, so we decided to leave it into the extent just described.

b) It is not as surprising or novel as the authors communicate to the reader that there are transcripts on the opposite strand of operons, especially given the compaction of bacteriophage genomes. The notion that they have functions to the phage is also not as novel or surprising as the authors state. If they didn’t, why would they be transcribed?

We concur with the reviewer that the presence of overlapping/antisense genes being transcribed is not unexpected; indeed, why wouldn’t they exist? Historically, when genomes were annotated and operons identified *in silico*, it was assumed that when a gene was oriented in the opposite direction to others, the flanking genes were associated with different operons. However, the original paper demonstrated that this assumption does not always hold true. In certain instances, the genes neighboring the gene in the opposite direction were transcribed from the same promoter, resembling the structure of a classical operon. This operon-like structure was disrupted by the presence of the antisense gene, thus aligning with the authors' definition of a "noncontiguous operon." This definition was also predicated on the notion, as mentioned earlier, that this genetic arrangement offered additional regulatory possibilities.

While we acknowledge that antisense transcripts are common in compact viral/phage genomes, to the best of our knowledge, there has not been a clearly defined case where a noncontiguous operon has been shown to regulate a significant *in vivo* phenotype. It could be argued that antisense genes often serve primarily to compact the genome and decouple the simultaneous expression of adjacent genes, rather than regulating a shared function. We have now provided a very clear and relevant example of their important regulatory function. We have made revisions in the text to elucidate this point.

c) The authors do not thoroughly demonstrate experimentally (as they do in the referenced PNAS paper) that this is what they call a “noncontiguous operon”. They do not concretely show biochemically that either the opposing transcript or an unconfirmed Xis gene product regulate the function of Tha-1. Though they provide several genetic experiments, by making deletions, etc. to the antisense region, they are inherently changing the sense transcript. To tease apart these functions I feel I need more biochemical evidence of what is taking place.

Regarding the opposing transcripts, they clearly exist based on the transcriptomics (Fig. 5b-c), and nuance is added with our characterisation of the *ith-1* (*xis* renamed) promoter with a reporter assay (Fig. 5a) combined with previous knowledge of the *cl* promoter. Moreover, we now have strong evidence for the translation of the *ith-1* (*xis*) gene. As for also changing the sense transcript when modulating the antisense transcript, this is the only way to study the antisense operonic structure and would also need to be done if characterising the region with biochemistry. Through our various

genetic prophage mutants, we dissect the structure by removing one possible contributing factor at the time, assessing the contribution of the *lth-1* (Xis) protein, the *ith-1* (*xis*) promoter, and the localisation of these elements in an antisense orientation to the *cl* (and *Tha-1*) operon. We have performed additional experiments, detailed below, in an RNase III knockout, and using *Tha-1* and *lth-1* (Xis) expressed from different plasmids, to bolster our previous results.

For one example, although an ORF is annotated (*xis*) the authors never provide concrete evidence that a Xis protein is translated. The authors mention in passing that they tried to express the protein and were unable to, but give no detail beyond that. Without such evidence, I am left wondering if this is just a case of anti-sense transcription and dsRNA degradation, downregulation of the *cl/int* promoter, some role for the Xis protein (direct protein-protein inhibition of *Tha-1*, stabilizing dsRNA transcripts for degradation, further down-regulating the *cl/int* promoter, activating degradation of *Tha-1*), a combination, or something else entirely. Without knowing if a protein is even produced, the model becomes unclear. Since this is a key message of the paper, I would ask for more biochemical data supporting the model, and more experiments to investigate the function of Xis.

As mentioned, we were unable to express and purify the proteins under study. To address the reviewer's comments, we have established a system where the anti-phage protein (referred to as *Tha-1*) is expressed from a plasmid. Cells carrying this plasmid were subjected to a challenge with a phage that is sensitive to *Tha-1*. As expected, the expression of *Tha-1* effectively blocked phage infection. We conducted a subsequent experiment by introducing an additional plasmid that expressed either the wild-type *xis* gene (now denoted as *ith-1*, signifying the inhibitor of *Tha-1*), the *ith-1* gene with a stop codon in the first position, or a recoded version of the *lth-1* protein (with an identical sequence but different codons). When we exposed the various strains to the phage, we observed that protection against phage infection was abolished in the presence of the WT *lth-1* protein, as well as the recoded *lth-1*. However, *lth-1*^{M1*} did not inhibit *Tha-1*, resulting in the phage being unable to infect the cells. This confirms that *lth-1* (formerly Xis) mainly functions as a protein and effectively inhibits its counterpart *Tha-1* protein (see new Figure 4 for details).

Major comments:

1) As stated above, the authors do not convincingly demonstrate that the ORF *xis* is translated. First, the protein should be epitope tagged and shown to be expressed on a Western blot. Alternatively, they could fuse GFP or another reporter to *xis* or the N-terminus of *xis*.

We strongly agree that providing evidence for *lth-1* (Xis) being translated is essential for our work. Our previous results, summarised in the new Fig. 4, demonstrate that *lth* (Xis) perform its role as a protein. Importantly, the manuscript also contains other data supporting *ith-1* being translated, including the *ith-1*^{M1*} mutation in Fig. 3e, and the *ith-1*^{M1*} mutation in Fig. 6a-c. Also, during our preliminary experiments (not included in the manuscript), we used an *ith-1*^{N6*} stop codon mutation, which behaved similarly to *ith-1*^{M1*}. In sum, we believe the *in vivo* and genetic data in our revised manuscript confirms that *ith* is translated into *lth* protein, and that the protein is the main regulatory

of Tha activity. The balance of *lth* expression during lysogeny and phage induction is what the antisense operonic structure contributes to, and why we have emphasised the antisense transcripts (noncontiguous operon) in our work.

Second, co-expressing Tha-1 and ORF61 is sufficient to activate and block growth – so the authors should test whether over-expression of Xis in this context can block toxicity. Preferably this would be with *xis* synonymously recoded so there is minimal effect from any antisense transcripts. A control with a mutated start codon could be tested as well.

We thank the reviewer for this suggestion, especially for the recoded *ith-1* (*xis*), which we think has been of great help proving this gene is translated without dsRNA interference. In Fig. 4a described above, we performed a very similar experiment, but instead of using the ORF61 overexpression plasmid, we used phage infection to trigger Tha-1 activity. We believe this is better and more natural since ORF61 is highly expressed with the plasmid system, and might be able to outcompete *lth-1* inhibition (even if this does not happen at physiological levels). This approach was also more convenient experimentally, since the Tha-1/ORF61 co-expression already uses two plasmids, and we do not have a reliable third plasmid in *S. aureus* compatible with these two.

Similarly, if you engineer 7206 to overexpress P_{xis}+*xis* (+/- rest of antisense transcript), does this overcome the protective effect of Tha-1 on 7206?

Again, a great suggestion. To avoid finicky phage engineering while still answering the reviewer's suggestion, we took advantage of a pre-existing 80α mutant that lacks *ith-1* (and a non-functional version of *tha-1* to allow the phage to obtain sufficient titres) (80α^{Δ*ith-1*, *tha-1* F32L}). We know that 80α is largely insensitive to plasmid-expressed Tha-1, producing similar numbers of smaller plaques (Fig. 2a). However, with the mutant 80α, infection is blocked by Tha-1. Therefore, *ith-1* of the wild-type 80α can normally block the Tha-1 of the host. This data is included as Fig. 4b.

Fig 4b

In addition, we hypothesised that the *ith* of an infecting phage can circumvent Tha immunity. From previous transcriptomics data, we know that the *ith-1* of 80α is expressed early upon infection (explaining Fig. 4b, above). We think immune evasion with *lth* proteins is quite common for our phages, and helps explain the infection patterns observed in Fig. 2a. An analysis of the *ith* genes of the phages in the panel (Fig. S6d) shows that many phages have *ith-2* genes, and all these phages are largely insensitive to Tha-2 inhibition (Fig. 2a). Only 80α and Φ11 have *ith-1* genes, and these phages are not very sensitive to pTha-1.

To conclude, the *ith* gene of an incoming phage can help circumvent Tha immunity, and this has added an extra layer to our work.

They do replace the predicted Xis start codon with a stop codon and this strain fails to lyse. However, this alone is not thoroughly convincing. Although minor, a cleaner experiment would be to mutate the start codon, as opposed to adding a stop codon in its place.

We are a bit confused by this comment. We do not add an additional codon, we change the ATG start codon to a TAA stop codon. Changing the first codon to a stop codon is the same as mutating the start codon, though it is slightly less likely to allow non-standard start codons upstream to be used and produce a longer, though functioning, *lth* protein. In preliminary experiments, we also used an N6* stop codon in *ith-1*, with the same results as the M1* version (not in the manuscript). We also observe similar, incomplete lysis when doing inductions for Fig. 6b, where the *ith* stop codon and *ith* deletion behave similarly. We have therefore left the figure as it is (old Fig. 3d, new Fig. 3e)

In figure 5b, ΔP_{xis} , *xis* reversed, it would be helpful to see what a start codon mutation looks like in this assay. Also, to be thorough, it would be nice to see that *Tha-1* is still expressed in this context so that the result is due to expression of *Xis* rather than interfering with expression of *Tha-1* in some way. This is a running issue throughout as the authors make many artificial genetic constructs but are unable to directly see what this is doing to expression of gene products. This leads to many assumptions being made about how the genetic constructs should probably operate but not how they do operate.

With the aforementioned results confirming the expression of *lth-1* and its ability to counteract *Tha-1* activity, all the experiments now possess a clear rationale, and their results are easily explainable. We have used multiple approaches to validate the proposed model, including the expression of various proteins, either individually or in conjunction with their respective partners, from plasmids, as well as modifications to the phage's genetic structure. These experiments have effectively dissected the roles of the different phage proteins involved in the process. We believe that there is no alternative hypothesis that could account for the results we have obtained. Therefore, introducing more mutants or constructs would only complicate the narrative of this study. For instance, the reviewer suggested that *Tha-1* expression might be negatively controlled by *lth-1*. However, in the experiments involving plasmids expressing *Tha-1* and *lth-1*, the defense system *Tha-1* is expressed from a distinct promoter, clearly independent of any control by *lth-1*. In these experiments, *lth-1* alleviates the function of *Tha-1*.

Another crucial point to consider is that *tha-1* is part of an operon that also includes the master repressor (*ci*) and the phage integrase (*int*). If *lth-1* controlled the expression of all these genes, one would expect that the evolved *ith-1* mutant phages, whether carrying mutations in *tha-1* or in the tail gene, would have impacted their functionality. However, when *tha-1* is removed (in the *ith-1* mutant), this mutant works like wild-type, with these mutations having no impact on the biology of the phage, suggesting that *lth-1* does not regulate other phage functions aside from its effect on controlling *Tha-1* activity.

2) As the authors show, at least some of the effect of the *xis* transcript on *Tha-1* is likely due to antisense transcripts and degradation of dsRNA by RNase III. If RNase III is not essential in *Staph*, the authors could test regulation of *Tha-1* in an RNase III KO or knock-down. This may allow them to ascertain how much regulation of *Tha-1* is through RNA degradation and how much is due to a potential *Xis* protein.

Following the reviewer's comment, we have conducted the suggested experiment. Our hypothesis was that genes transcribed in opposite directions might be reciprocally regulated, either through RNA polymerase collisions or double-stranded RNA (dsRNA) degradation by RNase III. Upon comparing the phenotypes of the various phage mutants, we observed that they were similar in both the wild-type and RNase III mutant strains (refer to Fig. S10a). Therefore, we can conclude that the observed regulatory effect most likely occurs through head-on RNA polymerase collisions.

Somewhat related, it may also be informative to knock-down the *xis* transcript during lytic growth and see if the 80 α can still form particles/plaques. That would tease apart how much regulation of *Tha-1* is due to lowering expression and would be a nice companion to the deletion mutants.

If straight-forward, this would be a nice experiment to do. We feel that it is unfeasible, however. Assuming the reviewer means knockdown of RNA transcripts, we are not aware of efficient RNA

knockdown platforms in *S. aureus*, and we feel like developing this for the current study would be excessive. If the reviewer refers to transcriptional knockdown by e.g. a dCas9 binding at the beginning of the *ith-1* (*xis*) gene, we think this would affect transcription of the downstream integrase gene (from the *cl* promoter), and possibly also of the *tha-1* transcript due to RNA polymerase pile-up at the DNA-bound dCas9, preventing full expression of the *tha-1* gene to be translated. Also, knockdown technologies are often somewhat inefficient, and it would be hard to ensure sufficient *ith-1* transcript knockdown to prevent all translation of lth-1 proteins.

Instead, we think the prophage induction of the 80 α variants in Fig. 6b addresses the concern raised. For *ith-1*^{M1*} and Δ *ith-1*, transcription from the *ith-1* promoter still occurs despite of the lth-1 protein not being made. To address the antisense transcript possibly leading to dsRNA degradation of the *tha-1* transcript, we performed the induction experiment in a Δ *rnc* background described above, ruling out that free *ith-1* transcripts significantly reduce *tha-1* transcripts through dsRNA degradation by RNase III.

3) Though I agree that Tha-1 by all available evidence is likely a toxic RNase, it would be simple to co-express Tha-1 and ORF61 and look at what is happening to RNA on a gel. I understand that finding the specificity of the cleavage might be beyond the scope, but to see RNA being cleaved is important.

Thanks to the reviewer's comment, we ran total RNA on an Agilent Bioanalyzer, visualising rRNA. This shows some RNA degradation of rRNAs upon Tha-1 activation by ORF61, and this data is now in Fig. 3b

Fig 3b

HEPN domains typically target ssRNA, and we suspect structural rRNAs are difficult substrates, most likely being cleaved at protruding rRNA loops or incompletely folded rRNA during ribosome biosynthesis. We think the main Tha-1 substrate is mRNA and possibly tRNA loops, leading to transcriptional shut-off.

4) ORF61 activates the defense system, but according to the phage panel data, pTha-1 blocks many phages. Do all of these phages have a homologous gene to ORF61? If not, please explain.

Yes, both ORF61 (Tha-1 trigger) and ORF54 (Tha-2 trigger) are highly conserved in *S. aureus* temperate phages. Of the 17 temperate phages, 15 phages have homologues of these two proteins. The two exceptions are Φ 12 and Φ SLT. This is now mentioned in the text (lines 339-340: "Of the 17 temperate phages in Fig. 2a, 15/17 have homologues of both 80 α ORF61 (recognised by Tha-1) and Φ 11 ORF54 (recognised by Tha-2).").

However, we believe the presence of the immune evasion proteins *ith-1/2* are more informative of how sensitive a phage is to Tha-1/2 inhibition. Many tested phages encode *ith-2*, explaining why they are resistant to Tha-2, and more phages are targeted by Tha-1. This is discussed in lines 350-353: "Of the phages in the panel, only two phages encode *tha-1/ith-1*, while six phages contain *tha-*

2/*ith-2* (Fig. S7c-d). Interestingly, none of the *ith-1* phages are strongly targeted by Tha-1, and none of the *ith-2* phages are strongly targeted by Tha-2 (Fig. 2a).”

We have made a new supplemental figure (Fig. S7) showing the phylogeny of ORF61, ORF54, Tha-1/2, and *lth-1/2*. This in large part helps explain the sensitivity to a given phage to Tha inhibition. In addition, the presence of *lth* proteins in phages adds a new layer to the *ith/tha* operon, where during lysogeny, it acts an immune system, but during phage infection, it can help an incoming phage evade Tha immunity.

5) In the phage panel data, as the authors note, Tha-1 naturally expressed within the genome offers little protection compared to the pTha-1 plasmid. As such, this is an artifact of overexpression and the broad anti-phage language should be toned down.

We notice two phages more sensitive to pTha-1 than prophage-encoded Tha-1, and one for pTha-2. In response, we have changed the wording in this part of the text removing the word broad, and broad no longer refers to Tha immunity in this context. We suspect overexpression of Tha from a plasmid can either allow recognition of a poorly recognised minor tail protein, or overcome *ith* inhibition by the incoming phage.

6) Regarding the phage plaquing data – since these phages are lysogenic, is it possible that rather than providing protection the phages are integrating? Can lytic variants of these phages be tested?

We do not think that Tha immunity promotes phage integration of the following reasons:

- i) In Fig. S5 (old Fig. S4), we show(ed) that the Tha-2 of Φ NM2 does not impact lysogenisation of an incoming 80 α phage.
- ii) The trigger of Tha, one of two minor tail proteins (ORF61/ORF54 homologues) are only expressed late in the lytic cycle, and should not be expressed when establishing lysogeny.
- iii) Fig. 2a shows that many phages are blocked by the prophage Tha-1/2 of 80 α / Φ NM2, and many of the incoming phages share integration sites with one of these two phages, making it difficult for them to integrate.

Yet, to be sure, we performed an infection of wild-type 80 α or 80 α -vir, a lytic version, into cells expressing Tha-2 ^{Φ NM2}, and immunity was observed in both cases. This is now Fig. S2d:

Fig S2d

7) The authors mention that this system could provide abortive infection. Typically, this is tested by infecting a strain containing the system with phages at MOIs both greater than and less than one. The experiments shown here are at MOI of 1 and 10. These MOIs should elicit the same general growth pattern if the system works by abortive infection as all bacterial cells would be infected. Please clarify.

Following the reviewer's comment, we have repeated the experiment at MOIs 0.1 and 5, fulfilling the criteria of higher and lower MOIs than 1. The results remain very similar, and have now replaced the old data in Fig. 2b-c. We believe the previous results at MOI of 1 still reflect most cells not being infected, due to i) the poor adsorption of our *S. aureus* phages (many phage particles do not infect, or take a long time to do so), and ii) general random error with measuring phage titres and CFUs needed to calculate MOIs.

8) Perhaps this is an artifact of the methodology or an issue with the figure (Fig. 4c), but the Nanopore RNA reads during lytic growth do not appear to contain the Xis start codon (and definitely not an RBS) as shown in figure 4C.

The reviewer is correct. Fig. 5c (previously 4c) has been adjusted to accurately reflect where the transcripts begin. Below is a zoomed-in picture (not part of the manuscript) of the reads in the relevant region, showing that the RBS and full *ith-1* gene are part of the transcripts (transcription starting 21 nucleotides upstream of the ATG).

9) Typically, phage defense systems are conserved in many contexts, and those found in prophages are also present in “defense islands” to some extent. *Tha-1* appears to be quite rare and only found in a few *S. aureus* prophages. Is it possible that this is an 80 α -specific super-infection exclusion mechanism, or perhaps a mechanism that blocks spurious lytic induction of 80 α ? The authors do not discuss this possibility. Relatedly, I also find it curious that the system targets itself, rather than recognizing some conserved region of foreign phages. Often, prophage encoded anti-phage systems do not target themselves.

Firstly, *Tha-1/2* seem common in *S. aureus*. A pBLAST search of *Tha-1*^{80 α} returns about 90 strong hits, almost all in *S. aureus*, while *Tha-2*^{9NM2} returns more than 100 strong hits. This is reflected by the phages we used in the phage panel in Fig. 2a, where out of 17 temperate phages, two have *Tha-1* and six have *Tha-2* (Fig. S6). In addition, two phages (ROSA and Φ 7206) encode *ith-2*, presumably to avoid immunity by *Tha-2*, without themselves encoding *tha-2* genes. This means that they have evolved to circumvent *Tha* immunity of lysogens they regularly infect. This is referred to in lines 354-356:

“Interestingly, two phages (ROSA and $\Phi 53$) only encode *ith-2*, and no *tha-2*, making *ith-2* a strict immune evasion factor with no role in regulating a cognate *tha-2* gene for these phages, used only to infect *Tha-2*-containing lysogens.”

We therefore anticipate that the *Tha* system is common in *S. aureus* phages, with possibly similar, more distantly related genes in other organisms. Therefore, we do state that the system seems quite common, e.g. in lines 505-507:

“It is commonly encoded by *S. aureus* phages under the lysogenic *cl* repressor promoter, and exemplified by *Tha-1* (found in e.g. 80 α) and *Tha-2* (found in e.g. Φ NM2).”

The idea that the antisense transcription can prevent spurious prophage induction or change lysogenisation dynamics upon phage infection is a very intriguing one. We performed several experiments to investigate this before our initial submission. It would make sense that the *ith-1* promoter might prevent transcription of the integrase after induction, helping to avoid reintegration (thus affecting induction rates). However, we found no evidence for this. For example, in the experiment below (not in the manuscript), we looked at spontaneous induction rates in wild-type 80 α and in an 80 α mutant lacking the *ith-1* promoter, the *ith-1* gene, and the *tha-1* gene (“80 α delta ORF2-3”). There was no significant change in spontaneous induction, suggesting the antisense transcription does not strongly regulate *cl* or *int* expression.

Not in the manuscript

Due to this negative data, and not wishing to complicate the paper further, we do not mention this possibility.

Thirdly, we do not find it that strange that the system would recognise a conserved protein shared by the cognate phage. As a lysogen, the prophage will be infected by both closely related and distantly related phages. We suspect that it would be hard to impossible to find a highly conserved gene that is shared between most closely related phages and not by the cognate phage (e.g. 80 α). This does not mean that the prophage should not evolve to also protect against closely related phages. Indeed, this is one of the reasons why such a diversity of anti-phage systems exists, and why there is often redundancy (e.g. 80 α also has an anti-lytic phage system (PMID: 36779718) and another system found by us).

Finally, we would like to add that the reviewer finding it curious to target a conserved gene shared with the cognate prophage adds to the novelty of our research, and shows it would be of interest to the scientific community.

10) It is unclear how diverse the phages in the tested phage panel are. Are they related to 80 α ? Something to indicate how related they are would be nice, e.g. a phylogenetic tree, whole genome alignment, etc. For all the reader knows these are all very similar phages.

This is a great point, and convinced us to perform phylogenetic analysis on the phages. However, the resulting figure below (not in the manuscript) does not tell the reader much about why phages are targeted. This is because the presence of *ith* genes in the phage, and small variations in minor tail proteins, are likely the only factors determining sensitivity to Tha. Also, in general, due to *S. aureus* temperate phages being quite closely related and being mosaic, phylogenetic comparisons often do not reflect the function of the phages (very similar phages can have different repressors or integrases, while distantly related phages can have many similar genes).

While we are happy to include a refined version of the above figure in the manuscript if requested, we think the phylogenetic analysis of ORF61/54 homologues, and of *lth* and *Tha*, are much more informative (new Fig. S7). From this, we can in large part explain why some phages are targeted by *Tha*-1/2 and not others (Fig. 2a).

Minor issues:

Lines 62-64: “However, no concrete examples have been described where the noncontiguous operon is advantageous for the regulation of a specific biological process.” – The referenced PNAS paper shows that this organization regulates menaquinone biosynthesis.

The referenced paper, by one of our co-authors, does describe the noncontiguous operon structure of the menaquinone biosynthesis operon. However, it does not provide a physiological explanation of why a noncontiguous operon is beneficial for the regulatory process, or whether a different configuration would also allow the regulation of this process. This is why we state that the *ith/tha*

noncontiguous operon is the first known case of where the genetic organisation is essential to the biological process in question.

Line 75: “labelled” should be “annotated as”

Changed

Line 83: “lamboid” should be “lambdoid”

Fixed

Line 199: “data now shown” should be “data not shown”?

Changed to “data not shown”

Line 480: “decreased fitness cost” should be either “decreased fitness” or “increased fitness cost”

Changed to “decreased fitness”

Fig 3a: Please invert the colors to distinguish this from a plaque assay.

Done

Fig 4a: Grid lines would be nice.

Horizontal (y axis) grid lines added to Fig. 5a (old 4a)

Fig. 4b and c: It would be nice to see total coverage in both directions rather than aligned reads (are these all the reads that mapped?).

For Fig. 5b-c (old Fig. 4b-c, read depth has been added on top of each panel):

Fig 5c: The caption says green and blue, I’m seeing green and orange.

The caption has been changed to say green and orange

Fig 5: A schematic similar to Fig. 1a, but with the position of P_{xis}, xis would be helpful to follow the several different constructs.

We agree. This section contains a lot of information, and we have added a schematic (Fig. 6a) to help the reader visualise the different constructs.

Perhaps the authors should rename xis since they show it is likely not an excisionase.

We have renamed Xis Ith (Inhibitor of Tha), with the 80 α version being Ith-1, and the Φ NM2 variant being Ith-2.

Reviewer #2

In this study, Rostøl et al. identify a new prophage-encoded anti-phage defence system that they call THA (Tail-activated, HEPN domain-containing Anti-phage system). They show that this system is encoded together with a cognate “immunity” protein that allows phages expressing it to bypass the defence and replicate normally. The immunity protein for this system was previously mistakenly annotated as an excisionase protein. The authors provide strong evidence that it is not an excisionase and does in fact act as an immunity protein that inhibits the activity of the THA system. The THA systems encoded in *S. aureus* phages are predicted to function as RNAses due to the presence of the HEPN domain, and the activity of the characterized system is hypothesized to induce growth arrest, preventing phage replication and protecting the bacterial population. The study of anti-phage defence mechanisms is of high interest in the microbiology community at the present time, so this work will be of quite broad interest.

We thank the reviewer for their appreciation of our work and for their kind comments.

Overall, this is a well-presented story; importantly the authors examined the activity of the system in the context of the prophage and have provided convincing data about how the system functions with respect to the inhibition provided by THA and the reversal of this inhibition by the Xis protein. However, there is a lack of insight into how this system is actually functioning to block phage replication. At which stage of the phage life cycle is the inhibition occurring? Does phage genome replication occur normally? Are phage virion intermediates produced? It appears that the cells are not lysed, so is there is some block to phage late gene expression (maybe only lysins)?

We acknowledge the need for additional mechanistic insights into the Tha mechanism, and as a result, we have incorporated supplementary data to offer a more comprehensive understanding of the system. In particular, our new Southern blot analysis of samples collected after prophage induction reveals that the *ith-1* (xis) mutant prophage is induced similarly to the wild-type and undergoes normal replication for the initial 60 minutes post-induction, after which replication is halted (see Fig. 3f). This timing corresponds to the expression of ORF61 (as indicated by our transcriptomic analyses, Fig. S6), which activates Tha-1. Furthermore, our new experiments demonstrate that when Tha-1 is active, it exhibits RNase activity (Fig. 3b), which is largely irreversible, as demonstrated in Fig. 3c. This explains why the prophage ceases replication. In sum, we believe that these recent findings, when combined with our earlier data, provide a detailed depiction of the Tha immunity mechanism.

It appears that a trigger for activity is located in the phage minor tail proteins, but it is not clear if this is a protein trigger, or an RNA sequence trigger.

We thank the reviewer for pointing this out. While we believed the Tha trigger was a minor tail protein, and not the RNA, we did not provide conclusive evidence of this, and the ORF61^{F13C} escaper mutation could also be an RNA escaper. Indeed, a recent report showed that the trigger for CBASS immunity in staphylococci is phage RNA (<https://www.biorxiv.org/content/10.1101/2023.03.07.531596v1>). While we have been unable to purify Tha-1 and ORF61 proteins to perform clean *in vitro* experiments, we performed an *in vivo* two-plasmid toxicity experiment with two stop codons in ORF61 (D2* and H50*), now in Fig. S3d:

Fig S3d

With these two mutations being in distant locations, we think it is highly unlikely that they could both fully disrupt the RNA from being recognised if RNA was the trigger. We also do not detect any predicted domains in the N-terminal of *Tha*, RNA-binding or otherwise. Instead, the ORF61 protein is the likely trigger.

Do the authors have evidence that there is a cleavage site in this region of the phage genome (i.e. do the mutations provide evasion of the system at the protein or RNA level)?

We do not think that *Tha* immunity is effected through specific cleavage of the *orf61* RNA, which is what a specific cleavage site *orf61* RNA would imply. The RNA could hypothetically be the *Tha* activator (discussed above), but not the *Tha* substrate. Firstly, we observe growth arrest in the *Tha-1*/ORF61 expression system, in the absence of any phage, suggesting host RNAs being targeted (Fig. 3a). In fact, using this system, phages that are insensitive to the system can be targeted if we activate *Tha-1* with ORF61 (Fig. 3d). We also see non-specific cleavage of rRNA (Fig. 3b). Thirdly, we see a lack of lysis in the prophage induction where *lth* is not translated in Fig. 3e (80α *lth-1^{M1}*). If the specific *orf61* (polycistronic) transcript was specifically cleaved, other late genes would be expressed, including the holin and lysin, which would result in normal lysis timing.

In addition, it's not clear if the growth arrest noted is reversible or irreversible, as might be expected for an *Abi* system. I would expect that more detailed insight into the specific mechanism of anti-phage activity be provided for publication in *Nature Micro*.

We have now performed a time course experiment with *Tha-1*/ORF61 expression, where we remove aliquots over time and see if the number of colony forming units (CFUs) decreases over time, in Fig. 3c:

Fig 3c

If *Tha-1* toxicity was fully reversible, we would expect the number of CFUs to remain stable over time in the *Tha-1*-ORF61 scenario. However, we see an early drop in CFUs already after 10 minutes, where only 10% of the original cells are still viable, and the number gradually decreases further. The number of escapers is negligible (the number of cells that can grow on +aTc plates). We were a bit surprised by the high toxicity of *Tha* activation, but it likely means that *Tha* is very potent once

activated. We must interpret this data carefully, since the expression levels of the proteins from the plasmids are likely higher than the physiological levels during immunity against a phage. However, we are unable to perform this experiment in a wild-type setting since i) we do not know exactly when Tha is triggered during phage infection, and ii) the phage will express a lot of toxic genes, which can lead to cell death regardless of Tha activity. Still, since ORF61 is expressed late in the phage life cycle, we think that at this point, the cell is doomed from the combination of Tha activity and toxic phage genes. So regardless of whether Tha causes cell death directly, the cell likely dies after Tha has been triggered under physiological conditions. This is mentioned in the discussion, which reads (lines 525-528):

“We observed rapid cell death upon activation of Tha-1 at non-physiological protein levels (Fig. 3c). Yet, we suspect that during phage infection, the minor tail protein trigger is expressed so late that the host cell is unlikely to survive, either through Tha-mediated cell death or toxic phage-expressed gene products.”

Major concerns

I found the term “regulator” used in reference to the activity of the Xis protein (e.g., lines 29, 74-76) confusing as this brings to mind transcription factor type regulatory proteins. At the protein interaction level these types of biological interactors are usually referred to as inhibitors. The authors should consider changing this terminology.

We thank the reviewer for pointing this out. We have changed the terminology in the lines mentioned. Lines 30-32:

“To avoid autoimmunity and allow the phage life cycle to complete, these systems are inhibited by a small overlapping gene previously thought to encode the phage excisionase.”

Lines 83-84:

“a gene previously annotated as excisionase (*xis*) and renamed *ith-1* (Inhibitor of Tha-1), inhibits Tha both through protein-protein interactions and antisense transcription.”

We have also refrained from the regulator terminology throughout the rest of the manuscript.

The MOI of 10 used in Fig. 2b did not result in complete cell lysis or premature cell lysis. Did the authors determine if the CFUs/ml are changing throughout infection; are the cells lysing or is there just bacteriostatic activity that is relieved at some point? Of the cells that regrow, what are they? Resistant mutants, lysogens?

In the infection at MOI 5 in Fig. 2b, and at MOI 10 in the old Fig. 2b, there is indeed no lysis observed, premature or otherwise. This is consistent with the lack of lysis during induction of the 80 α prophage with an *ith-1* stop codon (Fig. 3e), which does not lead to a delayed lysis (as it would for an eventual relief of the bacteriostatic state). To us it is also intuitive, since a non-specific RNase-induced growth arrest should not directly lyse the cell, and the phage will likely not produce sufficient holins and lysins in a growth arrested environment. We think that the cells that do grow at higher MOIs are the uninfected cells, and not survivors following phage clearance and resumption of growth. This is also supported by the toxicity observed in Fig. 3c (see above). Even with the majority of cells being infected and arrested, there will be many cells that are uninfected, and grow rapidly to a high OD. We therefore do not think the regrown cells during protection are resistant mutants or lysogens.

In response to the reviewer’s suggestion, we attempted an experiment where we infect cells and measure CFUs and PFUs over time. In this context, we want to see a drop in CFUs shortly after infection, proportional to the drop in PFUs, and indicating that the infected cells are unviable (due to Tha immunity or phage toxicity). If, unexpectedly, there was a drop in PFUs but not in CFUs, this would show that the bacteriostatic state could be relieved. However, this experiment did not work

since we do not see a marked drop of CFUs or PFUs in up to 20 minutes post-infection (not shown here, and not included in the manuscript). The lack of PFUs going down shows that the adsorption rates for our phage are low during a short period of time, and it makes impossible to detect CFU drops and help prove/disprove an Abi mechanism. We decided not to attempt incubating for longer than 20 minutes, since then the cells would divide so much that the CFU drop from infection will be eclipsed by cell division and with an increase in CFUs.

Of the cells that regrow in the condition without protection ($80\alpha^{\Delta(i\text{th-1}+\text{tha-1})}$), we suspect that these survive due to lysogenisation and superinfection exclusion. However, we are unable to show this directly, since $\Phi 7206$ integrates in the same chromosomal location as 80α , and we could not detect $\Phi 7206$ in cells after the experiment. We think that even without integrating, $\Phi 7206$ can enter an episomal, transient prophage state that protect against external $\Phi 7206$ infection during the duration of the experiment, or at least until reaching stationary phase. We still believe the lysogenisation is the reason for this, since we also infected RN4220 cells (without an 80α prophage) in the same experiment at MOI 5 (not in the manuscript, but shown below), and these recover and grow at exactly the same rate as the $80\alpha^{\Delta(i\text{th-1}+\text{tha-1})}$ cells. In these RN4220 cells ("RN4220 5"), we do detect integrated $\Phi 7206$ at the end of the experiment. Thus, overall, we believe that the $80\alpha^{\Delta(i\text{th-1}+\text{tha-1})}$ cells that recover, despite of no *Tha-1* protection, are due to transient superinfection exclusion from $\Phi 7206$ cells.

Figure not in the manuscript

So, in short, we suspect that infected cells with *Tha-1* stop to grow and cannot be rescued from the bacteriostatic state, and lyse very slowly (with fewer phages released) or not at all. We are unable to prove this directly, and would need microscopy, which we think is beyond the scope of this paper. Recovering cells in the non-protected cells lyse, and the cells that recover do so due to superinfection exclusion by the invading phage.

Quantification of changes in mRNA during phage infection of 7206 in a wild-type vs. *Tha-1* prophage mutant should be done to determine if there is an arrest in all cellular activities, or if there is mainly a loss of transcription of phage products.

We did not perform this experiment because the results would be difficult to interpret. Quantification of phage mRNAs would not prove that mRNAs have been cleaved by *Tha-1*, since changes could also come from a decrease in transcription of phage genes due to the general growth arrest. To add to this, the growth arrest from *Tha-1*/ORF61 co-expression (Figs. 3a, c-d) shows that *Tha-1* does cause general growth arrest also in the absence of phage, confirming that host RNA is targeted. It is highly likely that phage transcripts are also targeted, but even in the absence of this, *Tha-1* would still effect immunity by shutting the host down.

The authors must show that the H270 mutant is expressed and soluble to the same levels as wild-type *THA*. While the His residue may be involved in nucleic acid cleavage, mutation of this histidine

residue may also lead to decreased protein expression/stability or solubility. A Western blot would

provide evidence that it is the specific loss of enzymatic activity rather than instability/insolubility that causes the loss of activity.

Following the reviewer's suggestion, we performed a western blot of wild-type Tha-1 and the H270A mutant Tha-1, which shows similar expression levels. This data is included in Fig. S3a:

Fig S3a

Do phages that bypass THA defence encode a similar Xis, or is it strictly differences in the minor tail protein target sequence? Some additional alignments would help paint a clearer picture of the relationship between the Xis and Tha proteins.

We thank the reviewer for this question, and we have performed phylogenetic analyses on ORF61/ORF54, *Ith-1/2* (previously Xis), and *Tha-1/2*, which is Fig. S7. This shows that the sensitivity of phages to Tha immunity can mostly be explained by whether they encode their own *ith-1/2*. More phages have *ith-2*, which we think helps them to avoid immunity from Tha-2 from Φ NM2. The minor tail protein sequences can in part help explain Tha sensitivity as well (for example the middle cluster of phages for ORF61 does not activate Tha-1 immunity, and Φ 12/ Φ SLT and the lytic phages lack homologues to ORF61/ORF54, which is why they are not targeted). We have added lines to explain this in the figure legends for Fig. S7:

“Phages Φ 12, Φ SLT, Stab21, and K, which are not targeted by Tha-1/2 in Fig. 2a, lack ORF61/ORF54 homologues.

We are not including additional alignments, since we think the phylogenetic trees convey the message in a clearer way.

The escape mutants were done in a plasmid-based system and don't provide biological context if Xis is able to allow phage infection in the presence of Tha from a prophage. Can the repressor of another phage be exchanged with 80a and see if 80a (new repressor) can plate on a wild type 80a lysogen? Or can the Xis from 80a be put into 7206?

We think the *ith-1/2* presence on the phages (discussed above) combined with the interference data in Fig. 2a helps answer the question, and shows that *ith* genes can be used as immune evasion factors for the incoming phage. Indeed, two phages (ROSA and Φ 7206) encode only *ith-2*, not *tha-2*. These two phages are insensitive to Tha-2 interference in Fig. 2a, which means that their *ith-2* genes are only used to circumvent the Tha-2 system in lysogens, and not for regulating their own cognate Tha-2 system. From transcriptomic data (Fig. S6), we know that the *ith-1* gene of 80 α is expressed early during phage infection, presumably to avoid targeting by Tha-1, and we think this is also true for other phages.

In addition, from Fig. 2a, we know that 80 α is largely insensitive to plasmid-expressed Tha-1. However, with a mutant 80 α (which lacks *ith-1* and has a non-functional version of *tha-1* to allow the phage to obtain sufficient titres) (80 α^{Δ *ith-1*, *tha-1* F32L), infection is blocked by Tha-1. Therefore, *ith-1* of the wild-type 80 α can normally block the Tha-1 of the host. This data is included in Fig. 4b:

Fig 4b

While the reviewer is questioning whether this would also hold true when Tha is expressed from a lysogen instead of a plasmid, we think this assumption is sensible to make. Fig. 2a shows that if anything, protection by Tha is sometimes stronger when from a plasmid (presumably due to higher expression), so if the *ith-1* of 80α can block plasmid-encoded Tha-1 immunity, it will even more easily be able to block prophage-encoded Tha-1. While the suggestions of the reviewer (80α with a different repressor, or moving the 80α *ith-1* to Φ7206) could work and would be the most natural scenario, we think that the *ith-1/2* correlation with phage sensitivity in combination with the data in Fig. 4b clearly shows that the *ith* gene of an incoming phage can circumvent Tha inhibition. Also, the phage engineering required would be time-consuming, and it might be hard to make 80α function with a new repressor.

Are the Xis proteins broadly active or are they Tha sequence specific? Co-expressing the 80a and NM2 Xis/Tha-1/2 in the presence of bypassing phage would add knowledge for Tha functionality.

We have now addressed this question, and we have several lines of evidence for the specificity of Tha-1/2 and lth-1/2. Firstly, as discussed above, the phylogenetics of Tha and lth shows they are classified into two groups, Tha-1 and lth-1, exemplified by 80α, and Tha-2 and lth-2, exemplified by ΦNM2. In the interference data in Fig. 2a, no phage with lth-1 is strongly targeted by Tha-1, and no phage with lth-2 is strongly targeted by Tha-2.

Secondly, we performed an experiment to demonstrate that there is no cross-talk between the two types of lth/Tha. We used a two-plasmid system, where one plasmid expressed Tha-1 or Tha-2, and the second plasmid encoding lth-1 or lth-2. Using Φ7094, a phage sensitive to both Tha-1 and Tha-2, we showed that plaqueing could only occur when Tha-1 and lth-1 were together, and when Tha-2 and lth-2 were together. This is because the cognate lth inhibited the Tha immunity. The lth-2 of ΦNM2 could not inhibit the Tha-1 of 80α, or vice versa. This is included in Fig. 4c:

Fig 4c

Based on this, we believe that there are two types of lth/Tha systems, which do not cross-react.

Alternatively, can a chimera or truncated form of Tha be generated to determine if the N-terminal domain is essential for recognition of the phage tail proteins?

While this would be a nice proof of concept, we think this experiment is beyond the scope of the current manuscript. We do indeed believe that the N-terminal domain of Tha-1/2 is responsible for sensing ORF61/ORF54, while the C-terminal HEPN domain is the effector. We think it would be quite hard to get the chimera to work, since we would need to get the exact domain boundaries correct to allow the binding of the minor tail protein to relay the correct conformational change to the C-terminal domain. We think this would require a lot of trial and error, and it might not be possible. Instead, we hope that the lth/Tha non-cross-reactivity described above and the remaining phage sensitivity data is sufficient to understand that activates and inactivates Tha immunity.

Minor concerns

Figure 1a – Reference to the significance of the * and ** should be made in the figure legend.

Correct. However, with the introduction of the schematic showing the different 80 α mutants in Fig. 6a, the * and ** are no longer needed, and have been removed.

Figure 2a - The blank spaces make this heatmap difficult to read. Do the clear/white boxes represent no inhibition at all and the clear boxes with * mean infection to wild-type levels but the plaques were smaller? It should be clearly stated if these data are PFU/mL or fold-inhibition.

White spaces have been added, improving the figure. White means no inhibition, and * means small plaque phenotype, which gives similar numbers of plaques that are smaller than on the control strain. This is now stated in the Fig. 2a legend:

“White squares signify no immunity. Prophage repressor indicates repression of the incoming phage by its cognate phage repressor. SP, small (but similar number of) plaques.”

With regards to the type of inhibition, the legend states:

“The 10-fold reduction in plaqueing efficiency is shown from at least three independent experiments”

This means fold-change, and not PFU/ml.

Figure 3 – It would be easier for the reader if this figure was inverted; as presented these look like plaques, not colonies.

Done; Fig. 3a colours have been inverted.

Figure S1b – There are two orf3's noted.

Addressed, the first *orf3* has been changed to *orf2*.

Line 32 – Noncontiguous is spelled incorrectly.

Fixed

Line 155-162. This could be reworded to improve clarity. The movement between prophages and plasmid-expressed protein (I assume that is what Tha-1 alone means) is confusing. It would be useful to state how many phages were inhibited only by Tha-1 from the prophage.

Reworded. The relevant section has more details for clarity.

Line 390 – There is an empty pair of brackets.

A missing reference has been added

Supplemental Figure 1g – The labels say Φ 11 but the figure legend says 80a.

The legends of S1f-g now say Φ 11.

Reviewer #3

I have reviewed the manuscript. I find the science presented in the paper fascinating. Although the authors emphasized that their findings represent a new mechanism of phage exclusion, e.g. Crispr-CAS, I believe the data reveals a fascinating mechanism of temperate phage biology. The experiments are performed logically with clear results. I recommend publication of the manuscript in Nature.

We extend our gratitude to the reviewer for their constructive criticism and their appreciation of our research. Furthermore, upon careful consideration of their comments, we have recognised the significance of incorporating immunity and counter-immunity as integral components of the temperate phage life cycle. As a result, we have made the necessary modifications to our manuscript to introduce and substantiate this innovative concept.

But in my opinion the manuscript as presented is too long and wordy. I suggest it should be thoroughly edited for brevity and crispy reading. I also have some comments that the authors may respond to before the manuscript is accepted. My comments are not in order of chronology or in terms of importance although some are more important than others:

1. Of course, as I mentioned, it should be thoroughly edited.

We have thoroughly edited the manuscript for brevity and readability, and we hope this makes it easier to read and digest. This includes changes suggested by the reviewer below. However, with the inclusion of additional data due to concerns raised by the reviewers, we have also added new sections. Depending on the editor's point of view, we might also be able to shorten it further, but we wonder if this might also decrease readability for the broader readership of Nature Microbiology, since it would assume more prior knowledge of the field to understand without a detailed explanation.

2. I don't know why the gene referred to as *xis* was called so – may be because of homology or gene location – but since the authors have shown that it does not function as the 'classical' excisionase, I suggest to change it to one that more reflects its demonstrated function.

The gene was originally labelled *xis* due to a publication where they conclude a similar gene in phage L54a functioned as an excisionase (PMID: 2526804). We think this is untrue, also in the case of L54a.

We agree that it is sensible to rename the gene, and we have chosen *lth*, standing for Inhibitor of *Tha*. Both *Tha* and *lth* come in two flavours, *Tha*-1 and *lth*-1, exemplified by 80 α , and *Tha*-2 and *lth*-2, exemplified by Φ NM2 (Fig. S7). This naming convention makes it intuitive that type 1 and 2 recognise different minor tail proteins (ORF61 and ORF54, respectively), and that there is no cross-reaction (inhibition) between the components of each *Tha*/*lth* pair, e.g. *Tha*-1 is inhibited by *lth*-1, but not by *lth*-2 (see Fig. 4c)

3. The triangular symbols at the top of, for example, figure 3 is misleading the way they are presented. If they reflect Dilution as noted, then it should be horizontally inverted reflecting more and more dilutions; as presented they reflect phage titers, not dilutions.

This is true, and we have changed the figure text and figure legends to accurately describe the triangles.

4. The manuscript misleadingly and frequently uses the word *Tha1* when they really mean *Tha+* wild type situations. Please carefully edit the entire manuscript about this point. Was the mutant *Tha1* gene cloned in the plasmid? In line 166, the authors certainly mean, "... In the absence of WT *Tha* gene". Also, in line 143, same problem.

As explained above, we categorised *Tha* into two different types, *Tha-1*, exemplified by 80 α , and *Tha-2*, exemplified by Φ NM2. *Tha-1* and *Tha-2* recognise different minor tail proteins, and are inhibited by different *lth* proteins. Both *Tha-1/2* are *Tha* systems, however. We therefore believe the distinction is important, especially with the inclusion of the phylogenetic data in Fig. S6, which helps explain the interference data seen in Fig. 2a. From this, it seems that phages carrying an *ith-2* gene cannot be targeted by *Tha-2*, giving the *ith-1/2* genes an additional role, namely immune evasion during infection, as well as the previous role in regulating the cognate *Tha* system.

The above was not clear in our previous version of the manuscript, both because we did not explain thoroughly, and because we did not elaborate on the prevalence of the two types of *lth/Tha*. We have included wording to explicitly explain the classification in the new version. Lines 181-187 read:

"We observed that multiple phages have an identical genetic architecture and composition to 80 α in their lysogeny regions, including *ith-1/tha-1* pairs. Phages like Φ 11 and Φ 69 have almost identical *ith-1-tha-1* genes to 80 α . A second group of phages, including Φ NM2 (Fig. S2a) and Φ NM1, have similar genetic structures but the genes vary slightly in their sequences from the first group (Fig. S2b), containing *ith-2* and *tha-2*, which is even more abundant in the databases."

It is also mentioned in several other places, e.g. in lines 333-335:

"Since 80 α and Φ NM2 encode *tha-1* and *tha-2*, respectively, activated by different minor tail proteins, we wondered if their respective *ith-1* and *ith-2* genes have different immune suppression specificities."

Finally, to simplify, we have abandoned the use of THA to describe the system, referring to it only as *Tha*.

5. What kind of mutant is *Tha1*? Base-substitution? Deletion?

See above. *Tha-1* and *Tha-2* refer to different wild-type versions of the immune protein.

6. I strongly disagree with the use of the word non-contiguous operon. It is very misleading word to describe the case it refers to. I insist that the authors use a different word to describe overlapping antisense operons.

We appreciate that the use of the term noncontiguous operon has proven controversial. We have therefore toned down the emphasis of this term, removing it from the title, and discussing it less in the introduction and the discussion sections. Where we do use it, we usually define it in each case, and use it along with alternative phrases (like antisense operon), as well as often mentioning it along with the eukaryotic counterpart phrase, "complex transcripts".

For now, we have still decided to include passing references to the phrase. In our view, it is a more precise term than just saying overlapping antisense transcripts/operons, as explained in our rebuttal to reviewer 1. The phrase implies reciprocal regulation between the two antisense transcripts, and this co-regulation has not been described often in the field before, the *lth/Tha* example being an elegant example with a clear biological function for the genetic structure. In addition, the term noncontiguous operon is relatively recent, being coined only in 2019 (PMID: 30635413), and it will take longer before the term becomes ingrained in the scientific literature. We still do not think that this means we should entirely stop its use.

7. Line 121. "... We sequenced five 80a mutant phage" Fixed

8. Line 422. Shouldn't the statement "Tha targets tail proteins" be "tail protein targets Tha?"

Line rewritten, and no longer says "targets tail proteins"

9. Are the authors using the word "moron" in the sense that Roger Hendrix defined it? Whatever the authors are talking about, please clarify and give a reference.

The reviewer is correct. In one mention of it, we replaced the word moron to accessory gene, and this sentence reads (lines 205-206):

"broad immune potential of phages and their accessory genes,"

In the discussion section, where morons are mentioned, we have added a reference for Hendrix's publication (PMID: 10860721).

10. The Authors may point out that ORF61 has a dual role: tail protein as well as an activator of Tha gene.

The reviewer brings up a nice point. We have included this in our discussion, and lines 522-524 read:

"During phage infection, these late expressed proteins serve a dual function in addition to being structural phage proteins, namely directly or indirectly triggering the non-specific RNase activity of Tha."

11. Lines 317-322. These lines really belong to Experimental Procedures, not in the results section. This is one example. There are other situations in the manuscript where some of the statements belong to Experimental Procedures.

We have removed the lines in question, and included this information in the methods section instead. We have also rewritten throughout to decrease the amount of excessive experimental descriptions and things that could be in the experimental procedures section. One example is the description of how the Tha-1 escaper phage was generated (line 241)

12. Line 417. The paragraph doesn't add much; may be deleted.

We are unsure which the exact paragraph in question is; whether it is the paragraph that includes line 417 (and started at old line 410), or the section that started on line 419. If the former, we have shortened and rewritten the paragraph, though we think the previous section is rather complicated, and warrants a short summary. If referring to the section coming after (previous title "The THA phage escapers have reduced fitness"), we have removed this section entirely. We added a sentence mentioning it in passing in the part after the isolation of escaper phage, lines 249-252:

“These escaper mutants display reduced fitness, with 80α ORF61^{F13C} producing below 50% of WT titres (Fig. S4a), and Φ11 ORF54^{ΔT248-G396} 100-1000-fold lower titres compared to WT upon prophage induction (Fig. S4b).”

The figure with these data is no longer a main figure, but is in new Fig. S4.

13. I also believe some of the discussions talk about evolutionary aspects of the current results. I feel that those are not quite important points in the context of the current findings; may be left out.

We have rewritten the discussion section, with fewer mentions of evolutionary aspects.

Decision Letter, first revision:

Message 21st November 2023

:

Dear Professor Penadés,

Thank you for your patience while your manuscript "Bacteriophages avoid autoimmunity from cognate immune systems as an intrinsic part of their life cycles" was under peer-review at Nature Microbiology. It has now been seen by 2 of the original referees, whose expertise and comments you will find at the of this email. You will see from their comments below that your manuscript is almost there, but there are some final important points raised. We continue to be very interested in the possibility of publishing your study in Nature Microbiology, but would like to consider your response to these concerns in the form of a revised manuscript before we get to our final decision on publication.

In particular, you will see that Reviewer #2 requests some modifications to data display, as well as a sequence alignment. The rest referees' reports are clear and the remaining issues should be straightforward to address. Hopefully this will be quick!

If you have not done so already please begin to revise your manuscript so that it conforms to our Article format instructions at <http://www.nature.com/nmicrobiol/info/final-submission/>

The usual length limit for a Nature Microbiology Article is six display items (figures or tables) and 3,000 words. We have some flexibility, and can allow a revised manuscript at 3,500 words, but please consider this a firm upper limit. There is a trade-off of ~250 words per display item, so if you need more space, you could move a Figure or Table to Supplementary Information.

Some reduction could be achieved by focusing any introductory material and moving it to the start of your opening 'bold' paragraph, whose function is to outline the background to your work, describe in a sentence your new observations, and explain your main conclusions. The discussion should also be limited. Methods should be described in a separate section following the discussion, we do not place a word limit on Methods.

Nature Microbiology titles should give a sense of the main new findings of a manuscript, and should not contain punctuation. Please keep in mind that we strongly discourage active verbs in titles, and that they should ideally fit within 90 characters each (including spaces).

Please include a data availability statement as a separate section after Methods but before references, under the heading "Data Availability". This section should inform readers about the availability of the data used to support the conclusions of your study. This information includes accession codes to public repositories (data banks for protein, DNA or RNA sequences, microarray, proteomics data etc...), references to source data published alongside the paper, unique identifiers such as URLs to data repository entries, or data set DOIs, and any other statement about data availability. At a minimum, you should include the following statement: "The data that support the findings of this study are available from the corresponding author upon request", mentioning any restrictions on availability. If DOIs are provided, we also strongly encourage including these in the Reference list (authors, title, publisher (repository name), identifier, year). For more guidance on how to write this section please see:

<http://www.nature.com/authors/policies/data/data-availability-statements-data-citations.pdf>

To improve the accessibility of your paper to readers from other research areas, please pay particular attention to the wording of the paper's opening bold paragraph, which serves both as an introduction and as a brief, non-technical summary in about 150 words. If, however, you require one or two extra sentences to explain your work clearly, please include them even if the paragraph is over-length as a result. The opening paragraph should not contain references. Because scientists from other sub-disciplines will be interested in your results and their implications, it is important to explain essential but specialised terms concisely. We suggest you show your summary paragraph to colleagues in other fields to uncover any problematic concepts.

If your paper is accepted for publication, we will edit your display items electronically so they conform to our house style and will reproduce clearly in print. If necessary, we will re-size figures to fit single or double column width. If your figures contain several parts, the parts should form a neat rectangle when assembled. Choosing the right electronic format at this stage will speed up the processing of your paper and give the best possible results in print. We would like the figures to be supplied as vector files - EPS, PDF, AI or postscript (PS) file formats (not raster or bitmap files), preferably generated with vector-graphics software (Adobe Illustrator for example). Please try to ensure that all figures are non-flattened and fully editable. All images should be at least 300 dpi resolution (when figures are scaled to approximately the size that they are to be printed at) and in RGB colour format. Please do not submit Jpeg or flattened TIFF files. Please see also 'Guidelines for Electronic Submission of Figures' at the end of this letter for further detail.

Figure legends must provide a brief description of the figure and the symbols used, within 350 words, including definitions of any error bars employed in the figures.

When submitting the revised version of your manuscript, please pay close attention to our

[href="https://www.nature.com/nature-research/editorial-policies/image-integrity">Digital Image Integrity Guidelines.](https://www.nature.com/nature-research/editorial-policies/image-integrity) and to the following points below:

Please include a statement before the acknowledgements naming the author to whom correspondence and requests for materials should be addressed.

Finally, we require authors to include a statement of their individual contributions to the paper -- such as experimental work, project planning, data analysis, etc. -- immediately after the acknowledgements. The statement should be short, and refer to authors by their initials. For details please see the Authorship section of our joint Editorial policies at http://www.nature.com/authors/editorial_policies/authorship.html

- * include a point-by-point response to any editorial suggestions and to our referees. Please include your response to the editorial suggestions in your cover letter, and please upload your response to the referees as a separate document.

- * ensure it complies with our format requirements for Letters as set out in our guide to authors at www.nature.com/nmicrobiol/info/gta/

- * state in a cover note the length of the text, methods and legends; the number of references; number and estimated final size of figures and tables

- *This url links to your confidential homepage and associated information about manuscripts you may have submitted or be reviewing for us. If you wish to forward this e-mail to co-authors, please delete this link to your homepage first.

Please ensure that all correspondence is marked with your Nature Microbiology reference number in the subject line.

Nature Microbiology is committed to improving transparency in authorship. As part of our efforts in this direction, we are now requesting that all authors identified as 'corresponding author' on published papers create and link their Open Researcher and Contributor Identifier (ORCID) with their account on the Manuscript Tracking System (MTS), prior to acceptance. This applies to primary research papers only. ORCID helps the scientific community achieve unambiguous attribution of all scholarly contributions. You can create and link your ORCID from the home page of the MTS by clicking on 'Modify my Springer Nature account'. For more information please visit please visit www.springernature.com/orcid.

We hope to receive your revised paper within three weeks. If you cannot send it within this time, please let us know.

Yours sincerely,

Reviewers Comments:

Reviewer #1 (Remarks to the Author):

The authors have thoughtfully addressed all of my concerns, and I recognize the work that was put in to the revision.

Reviewer #2 (Remarks to the Author):

The authors adequately addressed the major concerns outlined in my previous review. I have a few additional comments on the revised manuscript.

Tha-1 and Tha-2 are activated by different minor tail proteins, which the authors say might possibly explain the differences in interference they observed. This seems like a simple analysis to do for phages for which the genome sequences are known and would provide additional evidence that these proteins are the trigger of the defence activity.

*edited to add that I see this is addressed later in the manuscript – it would be helpful to have this information included above. Also, a phylogenetic tree of these proteins is not what is needed here – a sequence alignment is. As amino acid positions that make phages susceptible to inhibition by Tha-1 are known, it should be quite simple to predict if a phage protein (e.g. ORF61 with conserved F13) will be targeted.

In the Tha-1 mutant isolation section the authors say they selected for a mutant plaque for which titre and plaque size were restored, which implies wild type activity. However, farther down this paragraph they say that this escaper mutant displayed reduced fitness. This appears to contradict their earlier statement.

It would be useful to know more about the phages used to test the Tha1 system in Figure 2a. Perhaps a table could be included. What does “similar, temperate Siphoviridae” mean? And which of the phages are Siphoviridae versus Myoviridae? As the authors make the point that Tha-1 activity is “targeting similar, temperate Siphoviridae, but not more distantly related lytic Myoviridae”, more detailed information should be included.

The section where all the 80a mutants are listed and then tested was extremely difficult to follow. It would be better to separate the mutants into groups and present the data at the same time that each mutant is described, along with the explanation for why they were created and the implications of the results.

Minor comments:

Lines 62-64 – This is not a complete sentence.

Line 87 – should read “and is presumed to...”

Line 147 – The term immunity usually applies to repressor-mediated resistance when referring to prophages. It would be better to use the term resistance here and elsewhere in

the manuscript.

Line 200 – should be HHpred

Line 208 – should be H270A

Line 228 – did the authors mean to add in some detail where the (methods) is?

Line 460 – the references need to be separated by a comma

Author Rebuttal, first revision:

We would again like to thank the reviewers for assessing our work in a fair and constructive manner. We are pleased that the new version addressed the main concerns of the reviewers, and believe that our findings and conclusions are now more firmly justified.

Reviewer #1 (Remarks to the Author):

The authors have thoughtfully addressed all of my concerns, and I recognize the work that was put in to the revision.

Thank you very much for your kind words.

Reviewer #2 (Remarks to the Author):

The authors adequately addressed the major concerns outlined in my previous review.

Thanks for your support.

I have a few additional comments on the revised manuscript.

Tha-1 and Tha-2 are activated by different minor tail proteins, which the authors say might possibly explain the differences in interference they observed. This seems like a simple analysis to do for phages for which the genome sequences are known and would provide additional evidence that these proteins are the trigger of the defence activity. *edited to add that I see this is addressed later in the manuscript – it would be helpful to have this information included above. Also, a phylogenetic tree of these proteins is not what is needed here – a sequence alignment is. As amino acid positions that make phages susceptible to inhibition by Tha-1 are known, it should be quite simple to predict if a phage protein (e.g. ORF61 with conserved F13) will be targeted.

We have now generated sequence alignments of the minor tail proteins. These proteins are highly conserved, usually above 90% identity to the respective 80 α and Φ 11 versions. With regards to the F13 residue, 7/7 phages targeted by Tha-1 have F13 in this position, while 4/8 of phages not targeted have F13, 4/8 having T13. There is therefore some correlation between F13 and targeting, but we don't think this is the main factor explaining targeting. Even the ORF61^{F13C} mutant can partially activate Tha-1 (Fig. 2d), and it is likely that a larger binding surface (involving more residues) is responsible for the putative ORF61-Tha-1 interaction.

Instead, as we have argued, the presence of *ith-1/2* seems like the main explanation for phage susceptibility to Tha-1/2. Since the data is not conclusive and it is difficult to display all the alignments in an easy way, we have decided not to include these alignments in the manuscript.

In the Tha-1 mutant isolation section the authors say they selected for a mutant plaque for which titre and plaque size were restored, which implies wild type activity. However, farther down this paragraph they say that this escaper mutant displayed reduced fitness. This appears to contradict their earlier statement.

We agree that this was previously inaccurate. The phrasing has been changed to: “we obtained one plaque for which titre and plaque size were partially restored”.

It would be useful to know more about the phages used to test the Tha1 system in Figure 2a. Perhaps a table could be included. What does “similar, temperate Siphoviridae” mean? And which of the phages are Siphoviridae versus Myoviridae? As the authors make the point that Tha-1 activity is “targeting similar, temperate Siphoviridae, but not more distantly related lytic Myoviridae”, more detailed information should be included.

We previously had phage accession numbers and categorisation in Extended Data Table 1b. We have expanded this table to now also include ORF54/61 homologues, and presence of Tha/lth, allowing the reader to access this information more readily. In the legend for Fig. 2a, we have also specified which phages are siphoviridae and which are myoviridae.

The section where all the 80a mutants are listed and then tested was extremely difficult to follow. It would be better to separate the mutants into groups and present the data at the same time that each mutant is described, along with the explanation for why they were created and the implications of the results.

We appreciate that this section was hard to follow, even after the addition of a map with the respective mutants (Fig. 6a). We have therefore rewritten this section to be more comprehensible. Specifically, we have grouped the mutants into two types, one where *ith-1* is expected to be inactive, and one where *ith-1* is expected to be more highly transcribed/expressed independently of the noncontiguous operon.

Minor comments:

Lines 62-64 – This is not a complete sentence.

Line 87 – should read “and is presumed to...”

Line 147 – The term immunity usually applies to repressor-mediated resistance when referring to prophages. It would be better to use the term resistance here and elsewhere in the manuscript.

WLine 200 – should be HHpred

Line 208 – should be H270A

Line 228 – did the authors mean to add in some detail where the (methods) is?

Line 460 – the references need to be separated by a comma

All these points have been addressed, either by rewording or by being in a section that was rewritten.

Decision Letter, second revision:

Message: Our ref: NMICROBIOL-23030723C

19th December 2023

Dear Dr. Penadés,

Thank you for submitting your revised manuscript "Bacteriophages avoid autoimmunity from cognate immune systems as an intrinsic part of their life cycles" (NMICROBIOL-23030723C). It has now been seen by the original referees and their comments are below. The reviewers find that the paper has improved in revision, and therefore we'll be happy in principle to publish it in Nature Microbiology, pending minor revisions to satisfy the referees' final requests and to comply with our editorial and formatting guidelines.

Thank you again for your interest in Nature Microbiology Please do not hesitate to contact me if you have any questions.

Sincerely,

Final Decision Letter:

Message 4th March 2024

:

Dear Professor Penadés,

I am pleased to accept your Article "Bacteriophages avoid autoimmunity from cognate immune systems as an intrinsic part of their life cycles" for publication in Nature Microbiology. Thank you for having chosen to submit your work to us and many congratulations.

Please note that *Nature Microbiology* is a Transformative Journal (TJ). Authors may publish their research with us through the traditional subscription access route or make their paper immediately open access through payment of an article-processing charge (APC). Authors will not be required to make a final decision about access to their article until it has been accepted. Find out more about Transformative Journals

We welcome the submission of potential cover material (including a short caption of around 40 words) related to your manuscript; suggestions should be sent to Nature Microbiology as electronic files (the image should be 300 dpi at 210 x 297 mm in either TIFF or JPEG format). Please note that such pictures should be selected more for their aesthetic appeal than for their scientific content, and that colour images work better than black and white or grayscale images. Please do not try to design a cover with the Nature Microbiology logo etc., and please do not submit composites of images related to your work. I am sure you will understand that we cannot make any promise as to whether any

nature portfolio

of your suggestions might be selected for the cover of the journal.

With kind regards,